# Projecting end of century climate extremes and their impacts on the hydrology of a representative California watershed

Fadji Z. Maina[1,3*], Alan Rhoades[2], Erica R. Siirila-Woodburn[1], Peter-James Dennedy-Frank[1]

[1] Energy Geosciences Division, Lawrence Berkeley National Laboratory 1 Cyclotron Road, M.S. 74R-316C, Berkeley, CA 94704, USA

[2] Climate and Ecosystem Sciences Division, Lawrence Berkeley National Laboratory 1 Cyclotron Road, M.S. 74R-316C, Berkeley, CA 94704, USA

[3] now at NASA Goddard Space Flight Center, Hydrological Sciences Laboratory, Greenbelt, MD, USA

*Corresponding Author: fadjizaouna.maina@nasa.gov

**Abstract**

In California, it is essential to understand the evolution of water resources in response to a
changing climate to sustain its economy and agriculture and to build resilient communities.
Although extreme conditions have characterized the historical hydroclimate of California, climate
change will likely intensify hydroclimatic extremes by the End of Century (EoC). However, few
studies have investigated the impacts of EoC extremes on watershed hydrology. We use cutting-
edge global climate and integrated hydrologic models to simulate EoC extremes and their effects
on the water-energy balance. We assess the impacts of projected driest, median, and wettest water
years under a Representative Concentration Pathway (RCP) 8.5 on the hydrodynamics of the
Cosumnes river basin. Substantial changes to annual average temperature ($>+2.5°C$) and
precipitation ($>+38\%$) will characterize the EoC extreme water years compared to their historical
counterparts. A shift in the dominant form of precipitation, mostly in the form of rain, is projected
to fall earlier. These changes reduce snowpack by more than 90%, increase peak surface water and
groundwater storages up to 75% and 23%, respectively, and drive the timing of peak storage to
occur earlier in the year. Because EoC temperatures and soil moisture are high, both potential and
actual evapotranspiration (*ET*) increase. The latter, along with the lack of snowmelt in the warm
EoC, cause surface water and groundwater storages to significantly decrease in summer, with
groundwater showing the highest rates of decrease. These changes result in more ephemeral EoC
streams with more focused flow and increased storage in the mainstem of the river network during
the summer.
**Keywords:** future climate extremes, integrated hydrologic model, global climate model, end of
century hydrology, watershed hydrology, water management

**Introduction**

California, the fifth*largest economy in the world, hosts one of the largest agricultural regions in the United States and is home to over 39 million people. Because of its geographic location, Mediterranean climate, geology, and landscape, the state of California is sensitive to climate change (Hayhoe et al. 2004). Understanding how water resources will evolve under a changing climate is crucial for sustaining the state's economy and agricultural productivity. The region is especially susceptible to climate change given its reliance on the Sierra Nevada Mountain snowpack as a source of water supply (e.g., Dettinger & Anderson, 2015). Studies show that temperatures may warm by as much as 4.5°C by the End of Century (hereafter, EoC) (Cayan et al., 2008), that snowpack is expected to decrease as most precipitation will fall as rain instead of snow (Siirila-Woodburn, et al., 2021), and that rain on snow events will exacerbate melt (Cayan et al., 2008; Gleick, 1987; Maurer, 2007; Mote et al., 2005; Musselman, Clark, et al., 2017; Musselman, Molotch, et al., 2017; Rhoades, Ullrich, & Zarzycki, 2018a). Given that precipitation falls predominantly in winter months and the summers are hot and dry, the snow accumulated during the winter provides important water storage for the dry season and is crucial to meet urban demand, sustain ecosystem function, and maintain agricultural productivity (Bales et al., 2006; Dierauer et al., 2018). As such, any significant reduction in the snowpack has the potential to drastically affect the hydrology of the state (Barnett et al., 2005; Harpold & Molotch, 2015; Milly et al., 2005; Rhoades et al., 2018 a,b).

Over the past several decades, researchers have worked to understand how changes in Sierra Nevada snowpack will affect important hydrologic fluxes such as evapotranspiration (Tague & Peng, 2013) and streamflow (Berghuijs et al., 2014; Gleick, 1987; He et al., 2019; Maurer, 2007; Safeeq et al., 2014; Son & Tague, 2019; Vicuna & Dracup, 2007; Vicuna et al., 2007). For

example, analyses of recent historical trends show that reductions in snowpack result in increases
in winter streamflow and decreases in the summer streamflow (e.g. Safeeq et al., 2012). However,
the sensitivity of a given area to these climatic changes depends on many factors including geology
and therefore drainage efficiency, topography, and land cover (Alo & Wang, 2008; Christensen et
al., 2008; Cristea et al., 2014; Ficklin et al., 2013; Mayer & Naman, 2011; Safeeq et al., 2015; Son
& Tague, 2019; Tang et al., 2019).

Climate change in California is also expected to lead to unprecedented extreme conditions,

which include both severe drought and intense deluge (Swain et al., 2018). In recent years, these
changes have already been observed in the forms of multi-year droughts (Cook et al., 2004; Griffin
& Anchukaitis, 2014; Shukla et al., 2015) and high-intensity precipitation events mainly caused
by atmospheric rivers (Dettinger et al., 2004; Dettinger, 2011; Dettinger, 2013; Ralph & Dettinger,
2011; Ralph et al., 2006). Periods without regular precipitation will require water management
strategies to adapt to ensure demands are met. Similarly, risk management plans and/or
infrastructure for floods, landslides, and other water surplus associated hazards (such as dam
failure) may also require reconsideration. This will be especially true if periods of precipitation,
including those associated with atmospheric rivers, become more extreme, variable, and occur
over a shorter window of time (Swain et al., 2018; Gershunov et al., 2019; Huang et al., 2020;
Rhoades et al., 2020b; Rhoades et al., 2021). Changes in water availability due to climate
"whiplash" will also have important ramifications for water resource management (Wang et al.,
2017; Swain et al., 2018) and significantly increase annual flood damages based on the level of
global warming that occurs (Rhoades et al., 2021). For example, in just the last two decades,
California has experienced the most severe drought in the last 1200 years (Griffin & Anchukaitis,
2014) followed by the wettest year on record (Di Liberto, 2017; SCRIPPS, 2017). These changes
in meteorological patterns may become the "new normal", raising several outstanding questions
related to how these changes in climate will impact the integrated hydrologic cycle, and
subsequently water resource availability for humans and ecosystems.

To project how changes in climate will impact watershed behavior, high-resolution,

physics-based models are one of the most promising ways to simulate system dynamics accurately,
particularly those that are non-linear, and constitute a better way to analyze a no-analog future than
the models used in the previous works. Previous studies analyzed future hydrologic conditions in
California but relied on models that do not 1) account for the interactions, feedbacks, and
movements of water from the lower atmosphere to the subsurface; 2) represent groundwater
dynamics and lateral flow; 3) incorporate physics-based high-resolution climate models and/or 4)
hydrologic models (e.g., Berghuijs et al., (2014); Gleick, (1987); He et al., (2019); Maurer, (2007);
Safeeq et al., (2014); Son & Tague, (2019); Vicuna & Dracup, (2007); Vicuna et al., (2007)).
Considerations of coupled interactions that explicitly account for groundwater connections are
important (Condon et al., 2020, 2013; Maxwell and Condon, 2016), especially given groundwater
is the largest reservoir in the terrestrial hydrologic budget and integral to water resource
availability. Also, previous studies have focused on the mid-century period (e.g. Maurer & Duffy,
2005; Son & Tague, 2019), which may indicate a more muted signal in hydrologic impacts than at
EoC. Understanding these impacts is essential because long-term climate projections show that
extremes will be more frequent and significant by the EoC (Cayan et al., 2008).

In this work, we assess the impacts of EoC extremely dry and intensely wet conditions on

the hydrodynamics of a Californian watershed that contains one of the last naturally flowing rivers
in the state. This allows us to investigate the impacts of climate change without the complexity of
active water management, and thus to set the context for water management decisions. We
specifically investigate how the water and energy balance respond to climate extremes under
climate change, and how those changes propagate to alter the spatiotemporal distribution of water
in different hydrologic compartments of the watershed. We focus our investigation on the changes
in groundwater and surface water storages. The balance of these two natural reservoirs, and their
relationship in response to changes in snowpack reservoir changes, is important for water
management decision making. We aim to 1) strengthen our physics-based understanding of the
main hydrologic processes controlling changes in water storages under a changing climate, 2)
quantify the magnitude and timing of these shifts in storage, and 3) identify the areas that are most
vulnerable to change.
To do so, we utilize a novel combination of cutting-edge climate and hydrologic model
simulations. We use an integrated hydrologic model (ParFlow-CLM; Maxwell & Miller, 2005),
which solves the water-energy balance across the Earth's critical zone. When projecting
hydrologic flows, ParFlow-CLM's explicit inclusion of three-dimensional groundwater flow is
important given its demonstrated role in impacting land surface processes like evapotranspiration
(Maxwell & Condon, 2016). We drive Parflow-CLM with climate forcing from a physics-based,
variable-resolution enabled global climate model (the Variable Resolution enabled Community
Earth System Model, VR-CESM; Zarzycki et al., 2014) that dynamically couples multi-scale
interactions within the atmosphere-ocean-land system. This novel pairing of models allows for
several key considerations not present in other methods. Our approach represents both dynamical
and thermodynamic atmospheric response to climate change across scales, different from "pseudo-
global warming" and "statistical delta" approaches used in many hydrologic modeling studies
(e.g., Foster et al., 2020; Rasmussen et al., 2011). While these approaches are useful to isolate the
impact of a given perturbation and/or variable, expected changes in climate will involve the co-
evolution of many processes, and may therefore not account for compensating factors. The
interaction between dynamical and thermodynamic responses has important, and sometimes,
offsetting effects on features such as atmospheric rivers. For example, Payne et al. (2020) show
that the thermodynamic response to climate change enhances atmospheric river characteristics
(e.g., Clausius-Clapeyron relationship), whereas the dynamical response diminishes atmospheric
river characteristics (e.g., changes in the jet stream and storm track landfall location). Therefore,
VR-CESM may simulate a more inclusive hydroclimatic response to climate change in the western
United States at a resolution that is at the cutting-edge of today's global climate modeling
capabilities for decadal-to-centennial length simulations (Haarsma et al., 2016).
We perform these couplings on spatial and temporal scales relevant for atmosphere-to-
land, and land-to-subsurface interactions, an important consideration, given the recent work
showing the importance of meteorological forcing resolution in representing the hydrologic cycle
(Kampenhout et al., 2019; Maina et al., 2020b; Rhoades et al., 2016; Rhoades, Ullrich, Zarzycki,
et al., 2018c; Wu et al., 2017). Climate conditions for EoC (2070-2100) and a 30-year historical
period (1985-2015) are simulated to identify the median, wettest, and driest water year (WY) in
each. We then simulate the subsequent watershed hydrology of each year using ParFlow-CLM
forced with those meteorological conditions.

**1. The Cosumnes watershed**
The Cosumnes River is one of the last rivers in the western United States without a major
dam, offering a rare opportunity to isolate the impacts of a changing climate on the hydrodynamics
without reservoir management consideration (Maina et al., 2020a; Maina and Siirila-Woodburn,
2020). The watershed spans the Central Valley-Sierra Nevada interface and therefore represents
important aspects of the large-scale hydrology patterns of the state, namely the assessment of
interactions between changes in precipitation, snowpack, streamflow, and groundwater across
elevation and geologic gradients. Located in Northern California, USA, the Cosumnes watershed
is approximately 7,000 km$^2$ in size (Figure 1) and is between the American and the Mokelumne
rivers. Its geology ranges from low-permeability rocks typical of the Sierra Nevada landscape
(volcanic and plutonic) to the porous and permeable alluvial depositions of the Central Valley
aquifers. These are separated by very low-permeability marine sediments. The watershed
topography includes a range of landscapes typical of the region (e.g. varying from flat agricultural
land, rolling foothills, and steep mountainous hillsides), and elevation varies from approximately
2500 m in the upper watershed to sea level in the Central Valley (Figure 1). The Sierra Nevada
mountains are characterized by evergreen forest while the Central Valley hosts an intensive
agricultural region including crops such as alfalfa, vineyards, as well as pastureland. Like other
Californian watersheds, the climate in the Cosumnes is Mediterranean consisting of wet and cold
winters (with a watershed average temperature equal to 0°C) and hot and dry summers (with
watershed average temperature reaching 25°C) (Cosgrove et al., 2003).

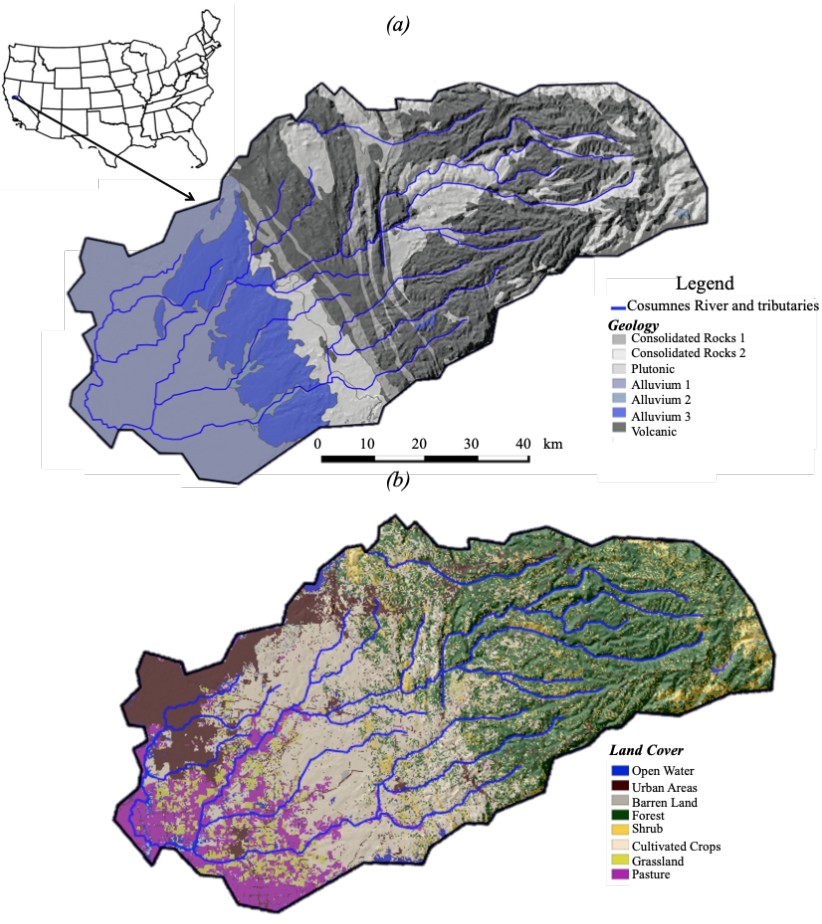


Figure 1: The Cosumnes Watershed (a) location and geology (Jennings et al., 1977), the alluvium

in blue corresponds to the Central Valley aquifers whereas the consolidated rocks in gray

correspond to the Sierra Nevada and cross-cutting marine sediments, and (b) land cover (Homer

et al., 2015).

## 2. Experimental Design

### 2.1. Variable Resolution Community Earth System Model (VR-CESM)

Historical and EoC meteorological forcings are obtained from a simulation using the VR-

CESM at a regionally refined resolution of 28 km over the Northern Pacific Ocean through the

western United States, including the Cosumnes watershed and a global resolution of 111 km

(Figure 2). CESM has been jointly developed by NCAR (National Center for Atmospheric
Research) and the DOE (U.S. Department of Energy) and simulates a continuum of Earth system
processes including the atmosphere, land surface, land ice, ocean, ocean waves, and sea ice and
the interactions between them (Collins et al., 2006; Gent et al., 2011; Hurrell et al., 2013). VR-
CESM is a novel tool to perform dynamical downscaling as it allows for the interactions between
the major components of the global climate system (e.g., atmosphere, cryosphere, land surface,
and ocean) while allowing for regional-scale phenomena to emerge where regional refinement is
applied, all within a single model (Huang et al., 2016; Rhoades et al., 2016; Rhoades, Ullrich, &
Zarzycki, 2018b; Rhoades, Ullrich, Zarzycki, et al., 2018c).

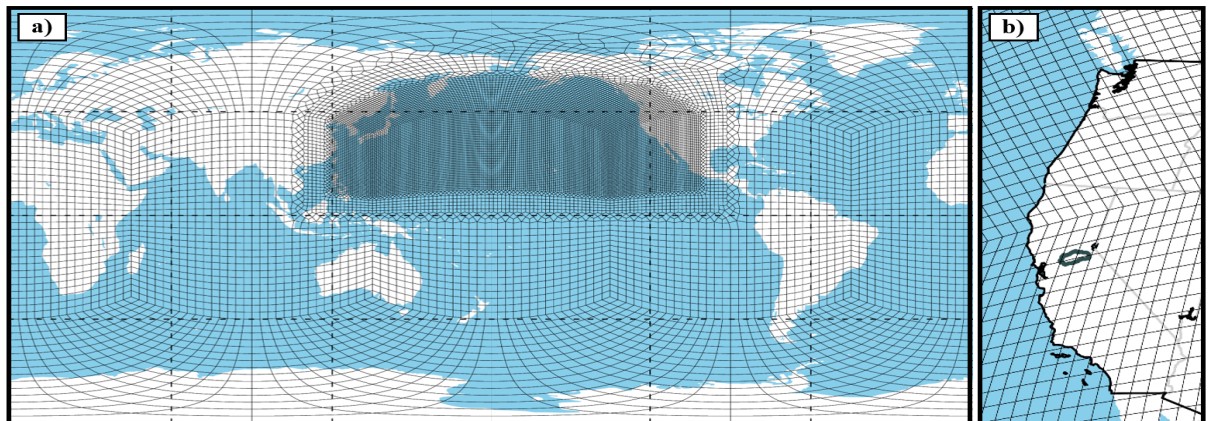

Figure 2: Variable Resolution Community Earth System Model (VR CESM) grid for (a) globe and
(b) coastal western US with the Cosumnes watershed overlaid in dark gray.

The atmospheric model used for these simulations is the Community Atmosphere Model

(CAM) version 5.4 with the spectral element dynamical core, with an atmospheric dynamics time
step of 75 seconds, an atmospheric physics time step of 450 seconds, a prognostic treatment of
rainfall and snowfall in the microphysics scheme (Gettelman and Morrison, 2015) and run under
Atmosphere Model Intercomparison Project (AMIP) protocols (Gates, 1992). Under the AMIP
protocols, the atmosphere and land-surface components of the Earth system model are coupled
and periodically bounded by monthly observed sea-surface temperatures and sea-ice extents.
Although this configuration does not exactly recreate historical water years and events, it is
expected to reasonably simulate the distribution of water year types. Also, it should be noted that
the model only projects future conditions, within the envelope of plausible future conditions of the
RCP8.5 scenario and its assumptions of greenhouse gas emissions, sea-surface temperatures, and
sea ice extents and would not be expected to exactly forecast individual water years. Simulations
with VR-CESM are performed for 30-year periods based on the climates from a historical period
(1985-2015) and an EoC period (2070-2100). EoC simulations, analogous to Rhoades, Ullrich, &
Zarzycki, 2018, are bounded by estimates of future changes in ocean conditions derived from a
fully-coupled bias-corrected CESM simulation (assuming historical ocean simulation biases will
be similar in the future simulation) and forced by greenhouse gases and aerosol concentrations
assumed in the RCP8.5 emissions scenario. Historical VR-CESM outputs have been compared
with reanalyses and future VR-CESM outputs have been analyzed for shifts in
hydrometeorological extremes in further detail in Rhoades et al., 2020 a,b. To couple the outputs
with ParFlow-CLM, we regrid the unstructured 28km VR-CESM data over the Cosumnes
watershed using bilinear interpolation in the Earth System Modeling Framework (Jones, 1999) to
a final resolution of approximately 11 km (i.e., 57 grids over the Cosumnes watershed). Notably,
each of the spectral elements in the VR-CESM grid, shown in Figure 1, has a 4x4 set of Gauss–
Lobatto–Legendre (GLL) quadrature nodes where equations of the atmospheric model are solved
(Herrington et al., 2019). Therefore, the actual resolution at which the atmospheric dynamics and
physics are solved in VR-CESM are at higher-resolution (~28km) than is shown in Figure 1,
making these some of the highest resolution global Earth system model simulations over California
to date (Haarsma et al., 2016).
To identify if VR-CESM is fit for purpose to simulate historical dry, median, and wet WYs,
and inform potential biases in future projections (over California and, more specifically, the
Cosumnes watershed), we first conduct a model comparison to a widely used observational
product, the Parameter-elevation Relationships on Independent Slopes Model (PRISM; Daly et al.,
2008) at 4 km resolution analogous to Rhoades et al., (2020a). However, in this study, we focus
our assessment of VR-CESM fidelity over California and the Cosumnes watershed. PRISM
provides daily precipitation, mean dewpoint temperature and maximum and minimum surface
temperature, and vapor pressure. PRISM precipitation and temperature data spanning 1981-2019
are compared with the VR-CESM 1985-2015 simulations.  We note that a mismatch in the time
period (1981-2019 versus 1985-2015) is deliberate. As stated previously, VR-CESM is simulated
under AMIP-protocols (bounded by monthly observed sea-surface temperatures and sea-ice
extents), and therefore we do not expect VR-CESM to exactly recreate past historical WYs.
However, we do expect that our 30-year simulation can reasonably recreate the range of WY types
over California and the Cosumnes, which is why we utilize the broader range of PRISM WYs that
are available.  For this comparison, we regrid the unstructured VR-CESM data to 4km resolution
(the native resolution of PRISM) using the Earth System Modeling Framework (ESMF) Offline
Re-gridding Weight Generator in the NCAR Command Language (NCL, 2021).
The comparison (discussed in appendix A) indicates that VR-CESM reasonably reproduces
the historical WY conditions (i.e., interannual range of PRISM precipitation largely overlaps with
the range of model bias simulated by VR-CESM). VR-CESM generally simulates a wetter
historical period over the Cosumnes (range of bias of 1330 mm) relative to PRISM (range of
interannual variability of 1320 mm). Basin-average minimum (421 mm) and maximum (1740 mm)
WY accumulated precipitation are slightly larger than those of PRISM. Of relevance to this study,
PRISM has shown notable uncertainties in the Sierra Nevada. Lundquist et al., 2015 showed that
an underrepresentation of the most extreme storm total precipitation in the Sierra Nevada can result
in an upper-bound uncertainty of 20% in WY accumulated precipitation in PRISM. Therefore, the
wettest WY simulated by VR-CESM is well within the 20% uncertainty range of PRISM's wettest
WY (1580 ± 316 mm). Further, differences in basin-average WY accumulated precipitation
between VR-CESM and PRISM are non-significant using a t-test and assuming a p-value < 0.05.
As discussed in further detail below, we posit that atmospheric river-related precipitation is likely
the driver of the wet bias mismatch with PRISM. However, we also note that the uncertainty
bounds of the PRISM product WY precipitation totals in the Sierra Nevada are estimated to be
upwards of ~20% too dry (e.g., Lundquist et al., 2015), particularly for extreme precipitation
events such as atmospheric rivers and in mountainous terrain.

**2.2. Integrated Hydrologic Model: ParFlow-CLM**
The integrated hydrologic model ParFlow-CLM (Kollet & Maxwell, 2006; Maxwell, 2013;
Maxwell & Miller, 2005) solves the transfer and interactions of water and energy from the
subsurface to the lower atmosphere including groundwater dynamics, streamflow, infiltration,
recharge, evapotranspiration, and snow dynamics. The model describes 3D groundwater flow in
variably saturated media with the Richards equation (equation 1, Richards, 1931) and 2D overland
flow with the kinematic wave equation (equation 2).
$$S_S S_W(\psi_P) \frac{\partial \psi_P}{\partial t} + \phi \frac{\partial S_W(\psi_P)}{\partial t} = \nabla . [K(x) k_r(\psi_P) \nabla(\psi_P - z)] + q_s \tag{1}$$

Where is $S_S$ the specific storage (L$^{-1}$), $S_W(\psi_P)$ is the degree of saturation (-) associated
with the subsurface pressure head $\psi_P$ (L), $t$ is the time (T), $\phi$ is the porosity (-), $k_r$ is the relative
permeability (-), z is the depth, $q_s$ is the source/sink term (T$^{-1}$) and $K(x)$ is the saturated hydraulic
conductivity (L T$^{-1}$).

ParFlow solves the mixed form of the Richards equation which has the advantage of

conserving the mass (Celia et al., 1990).

The kinematic wave equation is used to describe surface flow in two dimensions is defined

as:
$$-k(x)k_r(\psi_0)\nabla(\psi_0 - z) = \frac{\partial\|\psi_0,0\|}{\partial t} - \nabla.\vec{v}\|\psi_0,0\| - q_r(x) \quad (2)$$

Where $\psi_0$ is the ponding depth, $\|\psi_0,0\|$ indicates the greater term between $\psi_0$ and $0$, $\vec{v}$ is

the depth averaged velocity vector of surface runoff (L T$^{-1}$), $q_r$ is a source/sink term representing
rainfall and evaporative fluxes (L T$^{-1}$).

Surface water velocity at the surface in $x$ and $y$ directions, $(v_x)$ and $(v_y)$ respectively, is

computed using the following set of equations:
$$v_x = \frac{\sqrt{S_{f,x}}}{m}\psi_0^{\frac{2}{3}} \text{ and } v_y = \frac{\sqrt{S_{f,y}}}{m}\psi_0^{\frac{2}{3}} \qquad (3)$$
Where $S_{f,x}$ and $S_{f,y}$ friction slopes along $x$ and $y$ respectively and $m$ is the manning coefficient.
ParFlow employs a cell-centered finite difference scheme along with an implicit backward Euler
scheme and the Newton Krylow linearization method to solve these nonlinear equations. The
computational grid follows the terrain to mimic the slope of the domain (Maxwell, 2013).

ParFlow has many advantages in comparisons to other hydrologic models. Compared to

other hydrologic models (MODFLOW (Harbaugh, 2005), FELFOW (Trefry and Muffels, 2007),
SWAT (Soil and Water Assessment Tool) (Neitsch et al., 2000), SAC-MA (Sacramento Soil
Moisture Accounting Model)), ParFlow has the advantages of accounting for land surface
processes such as snow dynamics and evapotranspiration and their interactions with the subsurface
which are crucial for studying the hydrology of California. ParFlow also solved the subsurface
flow by accounting for variably saturated conditions, an important feature for calculating
groundwater recharge and the connection between the groundwater and the land surface processes,
which is not the case for the aforementioned models. While some hydrologic models have a better
representation of the land surface processes (Noah-MP (Niu et al., 2011), VIC (Variable
Infiltration Capacity Model Macroscale Hydrologic Model) (Liang et al., 1994)), these models do
not have a detailed representation of the subsurface flows. Because the surface flow is important
in the region and it establishes the connection between the headwaters and the valleys, its good
representation is essential for projecting changes in hydrology. Compared to other integrated
hydrologic models (CATHY (Catchment Hydrology) (Bixio et al., 2002), MIKE-SHE (Abbott et
al., 1986)), ParFlow has the advantages of solving a two-dimensional kinematic flow equation that
is fully coupled to the Richards equation.

ParFlow is coupled to the Community Land Model (CLM) to solve the surface energy and

water balance, which enables interactions between the land surface and the lower atmosphere and
the calculation of key land surface processes governing the system hydrodynamics such as
evapotranspiration, infiltration, and snow dynamics. CLM models the thermal processes by closing
the energy balance at the land surface given by:
$R_n(\theta) = LE(\theta) + H(\theta) + G(\theta)$ (4)

Where $\theta = \phi S_w$ is the soil moisture, $R_n$ is the net radiation at the land surface (E/LT) a

balance between the shortwave (also called solar) and longwave radiation, $LE$ is the latent heat
flux (E/LT) which captures the energy required to change the phase of water to or from vapor, $H$
is the sensible heat flux (E/LT) and $G$ is the ground heat flux (E/LT).

More information about the coupling between ParFlow and CLM can be found in Maxwell

& Miller, (2005). CLM uses the following outputs of the VR-CESM model at 3-hourly resolution
to solve the energy balance at the land surface: precipitation, air temperature, specific humidity,
atmospheric pressure, north/south and east/west wind speed, and shortwave and longwave wave
radiation.

We constructed a high-resolution model of the Cosumnes watershed with a horizontal

discretization of 200 m and vertical discretization that varies from 10 cm at the land surface to 30
m at the bottom of the domain. The model has 8 layers, the first 4 layers represent the soil layers
and the other four the deeper subsurface. The total thickness of the domain is 80 m to ensure
appropriate representation of water table dynamics. Observed water table depths (as measured at
several wells located in the Central Valley portion of the domain) vary between approximately 50
m and the land surface through a multi-year time period (Maina et al., 2020a). Therefore, to be
conservative for imposing the lower boundary layer, anything below 80 m is expected to remain
fully saturated. The resulting model comprises approximately 1.4 million active cells and was
solved using 320 cores in a high-performance computing environment. The Cosumnes watershed
is bounded by the American and Mokelumne rivers. We, therefore, impose weekly varying values
of Dirichlet boundary conditions along these borders to reflect the observed changes of river
stages. The eastern part of the watershed corresponding to the upper limit in the Sierra Nevada is
modeled as a no-flow (i.e., Neumann) boundary condition. Hydrodynamic parameters required to
solve the surface and subsurface flows (e.g., hydraulic conductivity, specific storage, porosity, and
van Genuchten parameters) are derived from a regional geological map (Geologic Map of
California, 2015; Jennings et al., 1977) and a literature review of previous studies (Faunt et al.,
2010; Faunt and Geological Survey (U.S.), 2009; Gilbert and Maxwell, 2017; Welch and Allen,
2014). We use the 2011 National Land Cover Database (NLCD) map (Homer et al., 2015) to
define land use and land cover required by CLM. We further delineate specific croplands (notably
alfalfa, vineyards, and pasture) in the Central Valley by using the agricultural maps provided by
the National Agricultural Statistics Service (NASS) of the US Department of
Agriculture's (USDA) Cropland Data Layer (CDL) (Boryan et al., 2011). Vegetation parameters
are defined by the International Geosphere-Biosphere Programme (IGBP) database (IGBP, 2018).
A complete description of the model parameterization can be found in appendix B and more details
in Maina et al. (2020a). The model has been extensively calibrated and validated using various
datasets, including remotely sensed data and ground measurements, which are however very sparse
in the area. Model validation which consists in comparing both surface and subsurface
hydrodynamics (groundwater and river stages) and land surface processes was performed over a
period of three years that includes extremely dry and wet water years (Appendix C). We
specifically compared simulated and measured river stages at three stations located in the Sierra
Nevada headwater, foothill, and the Central Valley. The annual averages absolute differences
between measurements and simulations were between 0.4 and 0.8 m. We selected four wells in the
Cosumnes watershed based on their availability of data to compare measured and simulated
groundwater levels. These wells are sparsely distributed in the Central Valley. The absolute
differences between observed and simulated groundwater levels vary between 0.47 to 3.73 m. The
highest absolute differences were attributed to the lack of best estimations of groundwater pumping
rates in the region. Nonetheless, the reasonable agreement between observations and simulated
variables over a period that includes both extremely dry and intensely wet conditions has allowed
us to conclude that the model can capture these extreme dynamics. We rely on remote sensing
data to assess the ability of our model to simulate key land surface processes (evapotranspiration
*ET*, soil moisture, and snow water equivalent *SWE*). We compared the simulated *SWE* to SNODAS
(The National Weather Service's Snow Data Assimilation, National Operational Hydrologic
Remote Sensing Center, 2004) and a *SWE* reanalysis by Bair et al., (2016). Our comparisons
indicated that the absolute differences between our *SWE* values and these data were equal to 3 mm
on average. Moreover, the simulated key parameters controlling the snow dynamics such as peak
snow and timing of snow ablation were also in agreement with remotely sensed data for both dry
and wet years (Appendix C). Absolute differences between the simulated *ET* and the remotely
sensed *ET* from METRIC (Mapping Evapotranspiration at High Resolution with Internalized
Calibration, Allen et al., 2007) were equal to 0.036 mm/s while the differences between the
simulated soil moisture and the SMAP (Soil Moisture Active Passive, SMAP, 2015) soil moisture
were 0.2. More details about model calibration and validation can be found in Appendix C and
previous publications (Maina et al., 2020a, Maina et al., 2020b; Maina and Siirila-Woodburn,
2020c). The model has also been successfully used in recent investigations of post-wildfire and
climate extremes hydrologic conditions and to assess the role of meteorological forcing scale on
simulated watershed dynamics (Maina et al., 2020a, b; Maina and Siirila-Woodburn, 2020c).
Initial conditions for pressure-head were obtained by a spin-up procedure using the forcing of the
historical median WY. We recursively simulated the historical median WY forcing until the
differences of storage at the end of the WY were less than 1%, indicating convergence. This
pressure head field is then used as the initial condition for each of the five WYs of interest (i.e.,
the EoC wet, EoC dry, historic wet, historic dry, EoC median). Though we acknowledge land
cover alterations are expected to occur by the EoC (either naturally or anthropogenically), in this
work we assume that the vegetation remains constant for both historical and EoC simulations for
simplicity. Although outside of the scope of this work, future studies will investigate the impacts
of an evolved land use/land cover, vegetation physiology, and resilience strategies to manage water
resources. Further, while the Central Valley of California hosts intensive agriculture that is reliant
on groundwater pumping for irrigation, we didn't incorporate pumping and irrigation in our model
configuration. We did this with the assumption that groundwater pumping rates may substantially
change in the future due to new demands, policies, regulations, and changes in land cover and land
use and aim to provide an estimate of the natural hydrologic system response to climate change.

**2.3. Analysis of EoC hydrodynamics**
To investigate how the EoC climate extremes affect water storages, we investigate five
hydrologic variables: *SWE*, *ET*, Pressure-head ($\psi$) distributions, and surface and subsurface water
storage. Total groundwater (GW) storage is given by:

$$Storage_{GW} = \sum_{i=1}^{n_{GW}} \quad \Delta x_i \times \Delta y_i \times \Delta z_i \times \psi_i \times \left(\frac{S_{s_i}}{\phi_i}\right) \qquad (5)$$

where $n_{GW}$ is the total number of subsurface saturated cells (-), $\Delta x_i$ and $\Delta y_i$ are cell discretizations
along the x and y directions (L), $\Delta z_i$ is the discretization along the vertical direction the cell (L),
$S_{s_i}$ is the specific storage associated with cell *i*, $\psi_i$ the pressure-head, and $\phi_i$ is the porosity.
Total surface water (SW) storage which accounts for any water located at the land surface
(i.e., any cell of the model with a pressure-head greater than 0) and includes river water or overland
flow is calculated via:

$$Storage_{SW} = \sum_{i=1}^{n_{SW}} \quad \Delta x_i \times \Delta y_i \times \psi_i \qquad (6)$$

where $n_{SW}$ is the total number of cells with surface water i.e., with surface $\psi$ greater than 0 (-),
and *i* indicates the cell.
We compare each EoC WY simulation to its corresponding historical WY counterpart and
both the historical and EoC medians. This allows us to assess how EoC extremes change relative
to what is currently considered an extreme condition as well as to "normal" in the relevant time.
Comparisons are shown as a percent change (*PC*) calculated using:

$$PC_{i,t} = \frac{X_{projection_{i,t}} - X_{baseline_{i,t}}}{X_{baseline_{i,t}}} \times 100 \qquad (3)$$

where $X$ is the model output ($ET$, $SWE$, or $\psi$) at a given point in space ($i$) at a time ($t$), *baseline* is
the selected simulation (historical median, EoC median, or historical extreme), and *projection*
represents the simulation obtained with the EoC extreme WYs (dry or wet).

**3.  Results**
In this section, we present a subset of the outputs from VR-CESM (precipitation and
temperature) to identify the extreme (dry and wet) and median WYs of interest. Changes in fluxes
and storages over the course of each WY, as well as the spatial variability of these changes in two
important periods of the WY (peak flow and baseflow) are also shown.

**3.1. Selection of the median, dry, and wet WYs**
From the historical and EoC 30-year VR-CESM simulations we select the median, wettest,
and driest WYs for comparison (see Figure 3a). Overall, the future WYs are ~30% wetter than the
historical WYs (p-value ~0.006 for two-tailed t-test of equal average annual precipitation) in
addition to being ~4.6°C warmer. Precipitation and temperature variances are mostly similar in the
historical and EoC simulations, though EoC minimum temperature may be more variable (p-value
~0.059 for two-tailed f-test of equal variance in minimum temperature). On average the timing for
the start, length, and end of precipitation is similar, though EoC precipitation may be less variable
in its start time (p-value ~0.053 for f-test of equal variance in days to reach 5th percentile of annual
precipitation). In the climate model, there are no clear trends between the precipitation timing
metrics and total amount of precipitation.

The EoC median WY is much wetter than its historical counterpart, with about ~250 mm

more precipitation that begins approximately 1 week earlier and ends approximately 2 weeks
earlier in the year. The EoC wettest WY is much wetter than the historical wettest WY and is
characterized by 42% more precipitation. This is consistent with Allan et al. (2020), who suggest
a wetter future. The EoC wettest WY is 3.8ºC warmer than the historical wettest WY and 4.6ºC
warmer than the historical median WY, as the historical median WY is one of the coolest years in
the series. Precipitation occurs earlier in the EoC wet WY compared to the historical wet or median
WYs, with the 5$^{th}$ percentile of precipitation reached 12 days earlier in the EoC wettest WY than
either the wettest or median historical WYs. The duration of the EoC wettest WY precipitation
season (146 days) is between the historical wettest WY (133 days) and the historical median WY
(155 days).

The EoC dry WY is also much wetter than its historic counterpart; in fact, the EoC dry WY

is wetter than the seven driest historical WYs of the 30-year historical ensemble. Simulation of 30
random draws from two identical normal distributions, repeated 100,000 times, finds that the
lowest value in one is higher than the seven lowest values in the other only ~1.1% of the time (p-
value ~0.011).  This statistical test reveals that this VR-CESM simulation suggests that future dry
years will be somewhat wetter than historical dry years. The EoC dry WY is only ~2.5ºC warmer
than the historical dry WY. The divergence in temperature is smaller for the comparison of EoC
and historical WYs of the dry extremes as opposed to the wet extremes because the historical dry
WY is the second-warmest WY in the historical simulations, while the EoC dry WY is the third
coolest in the EoC simulations. Precipitation in the EoC dry WY starts particularly early, with the
5$^{th}$ percentile of annual precipitation reached by mid-October. This is much earlier than either the
dry or median historical WYs, which don't reach that percentile of precipitation until mid-to-late
November. The historical dry WY also has a particularly short precipitation duration of only 97
days, while the EoC dry WY has a 163-day precipitation duration, more similar to the median
historical WY duration of 155 days.

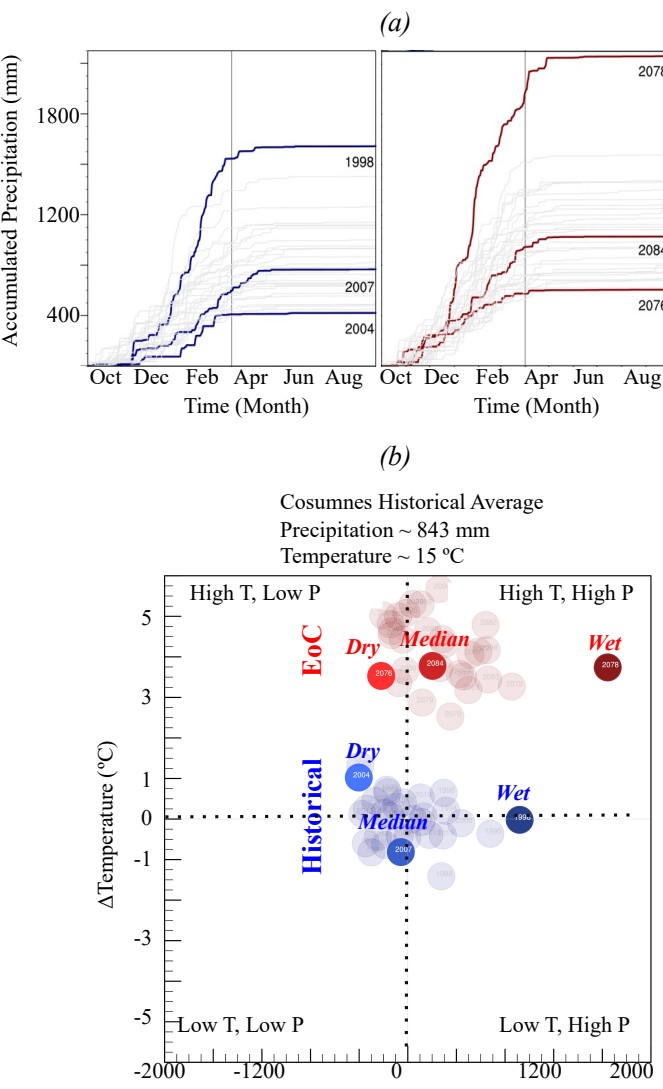

Figure 3: (a) VR-CESM accumulated total precipitation for the historical and End of Century
(EoC) simulations, and (b) quadrants for differences between each individual water year (WY)
and the historical average temperature and accumulated precipitation in the Cosumnes watershed.
The historical and EoC dry, median and wet WYs are indicated in blue and red, respectively.

Figure 4 shows the spatial distribution of accumulated precipitation anomalies across
California. These anomalies are computed for each of the six identified WYs relative to the
climatological average (the 30-year historical mean). These spatial plots provide context for the
changes modeled in the Cosumnes watershed relative to broader precipitation changes California-
wide. As in the Cosumnes, California-wide EoC dry, median, and wet WYs are all characterized
by higher precipitation totals than their historical counterparts. Importantly, the EoC wet WY is a
true outlier not only in the Cosumnes but across California too. California lies at an important
large-scale circulation transition, namely semi-permanent high-pressure systems associated with
the Hadley circulation. Therefore, how climate change alters the atmospheric dynamics over
California, or more specifically how far northward storm-tracks may shift, remains uncertain and
depends on climate model choice. This has led to papers that claim the future of California will be
wet across a range of climate models (e.g., Neelin et al, 2013; Swain et al., 2013; Gershunov et al.,
2019; Rhoades et al., 2020b; Persad et al., 2020) and, for select climate models, that it could be
drier.  Notably, these studies highlight an asymmetric response in the frequency of wet versus dry
WYs (i.e., anomalously wet WYs increase in frequency much more in the future than anomalously
dry WYs).  Many of the aforementioned studies also highlight that in anomalously wet WYs
extreme precipitation events (e.g., atmospheric rivers) will occur with greater intensity and
frequency and largely drive changes in WY precipitation totals (which is shown in our VR-CESM
simulations for California in more detail in Rhoades et al., 2020b).  Given these complexities and
others such as consideration for how dynamical and thermodynamical effects of climate change
may interact with one another to offset or amplify extreme precipitation events (Payne et al., 2020),
the hypothesis that global warming will result in a climate where the "wet gets wetter and dry gets
drier" may be too simplistic of an assumption for California.  Rhoades et al., (2020b) shows
quantitatively that the increases in precipitation observed in the VR-CESM outputs are due to a
greater number of intense atmospheric river events that occur more regularly back-to-back, which
was recently corroborated by Rhoades et al. (2021) using uniform-high-resolution CESM
simulations at different warming scenarios, and that atmospheric river precipitation totals increase
at a much larger rate (+53%/K) than non-AR precipitation totals (+1.4%/K), which agrees with
findings made in other studies such as Gershunov et al. (2019).

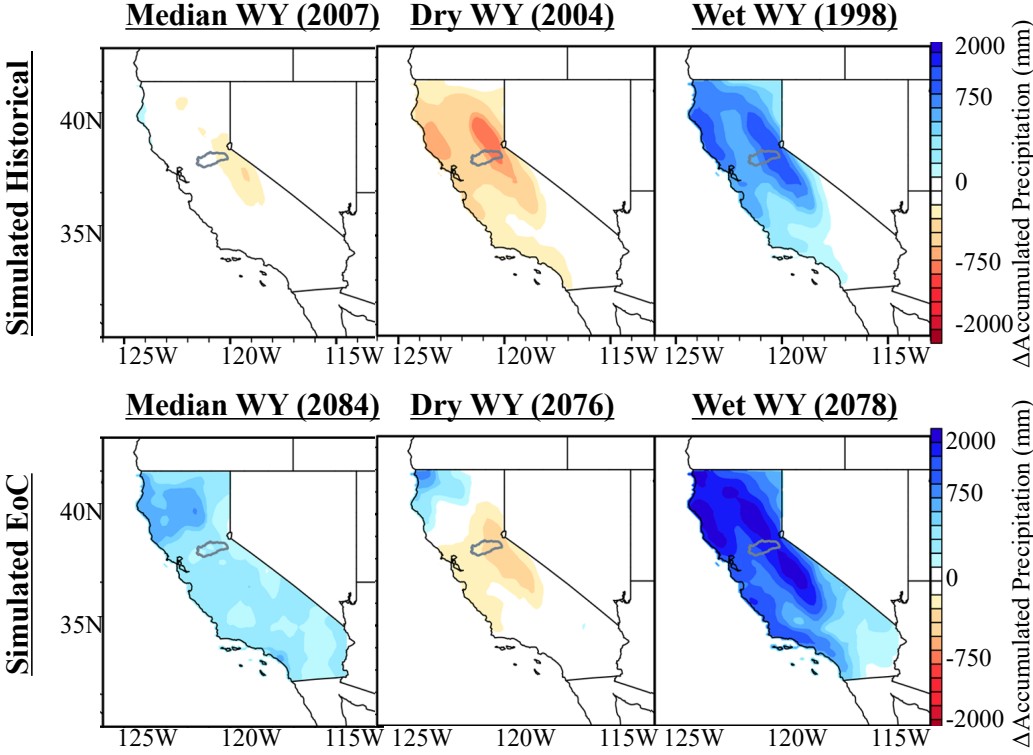

Figure 4: Precipitation spatial distributions of the dry, median, and wet water years (WY) for the
30-year historical and EoC simulations relative to the climatological average (derived from the 30-
year historical mean)

**3.2. Changes in annual watershed-integrated fluxes and storages**

Figure 5 illustrates the annual changes in the integrated hydrologic budget of the Cosumnes

watershed for the EoC WYs (i.e., median, dry, and wet) compared to the historical median WY.
The EoC median WY compared to the historical median WY has 38% more precipitation and the
temperature is 4.4°C higher. Further, the precipitation phase also shifts with an increase in rainfall
(54%) and a decrease in snowfall (-54%). This results in a significant decrease in *SWE* (-91%)
which is consistent with many other studies that have shown that increased temperatures due to
climate change will lead to low-to-no snow conditions (Berghuijs et al., 2014; Cayan et al., 2008;
Mote et al., 2005; Rhoades et al., 2018 a,b; Son & Tague, 2019). The increase in temperature and
precipitation results in an increase in *ET* (62%), consistent with the findings of other recent studies
(e.g. McEvoy et al., 2020). Nevertheless, the larger amount of precipitation associated with the
EoC is enough to offset higher *ET* demand and recharge groundwater and surface water, which
experience an increase of 4% and 19% respectively. The EoC wet WY has similar changes as the
EoC median WY when compared to the historical wet WY yet the magnitude of the increase in
surface (21%), and groundwater (11%) storages are higher due to more precipitation and higher
temperatures. The dry EoC WY is also characterized by higher precipitation (43%, the largest
increase) than its historical counterpart, this results in large increases in total groundwater (8%)
and surface water (38%) storages.
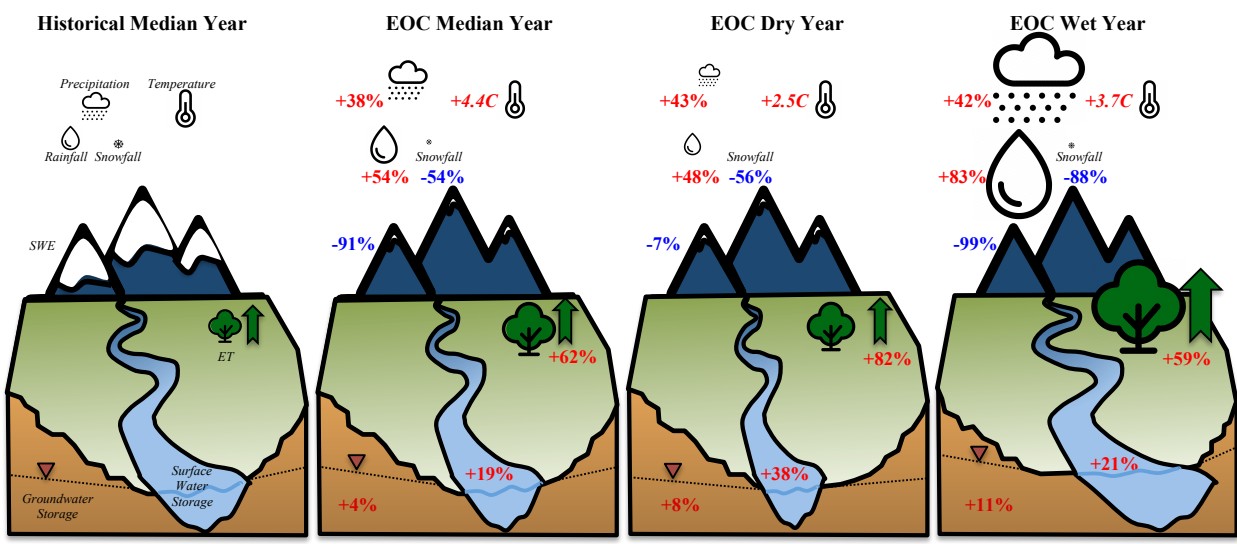

Figure 5: Annual percent changes in precipitation, rainfall, snowfall, temperature, *SWE*, *ET*, surface water, and groundwater storages in the EoC water years (WY) (i.e median, dry, and wet) at the watershed scale relative to their historical counterparts. Info-graphic size scaled to EoC conditions.

### 3.3. Temporal variation of watershed-integrated fluxes and storages

Understanding the annual changes at the watershed scale is important to broadly understand changes in the water budget in response to future climate extremes. However, a deeper understanding of the processes that drive these changes and the interactions from atmosphere-through-bedrock requires an analysis of their spatiotemporal variations as well. Figure 6 shows the temporal variations of each of the historical and EoC WY's integrated hydrologic budgets grouped by WY type (columns), with a top-down sequencing of hydrologic variables of interest in order from the atmosphere through subsurface (rows). This organization allows for the investigation of propagating impacts to be directly compared in time. In this section, we discuss historical vs EoC changes observed in each of the WY types (i.e., median, dry, and wet). Each WY shows unique hydrodynamic behaviors and changes compared to the historical conditions. The median WY sheds light on how changes in the precipitation phase and increases in temperature and precipitation in the EoC will impact the hydrodynamics. The dry WYs allow comparing EoC and historical low-to-no snow conditions whereas assessing the hydrodynamics of the EoC wet WY provides a better understanding of how intense EoC precipitation along with the warm EoC climate will shape the hydrology.

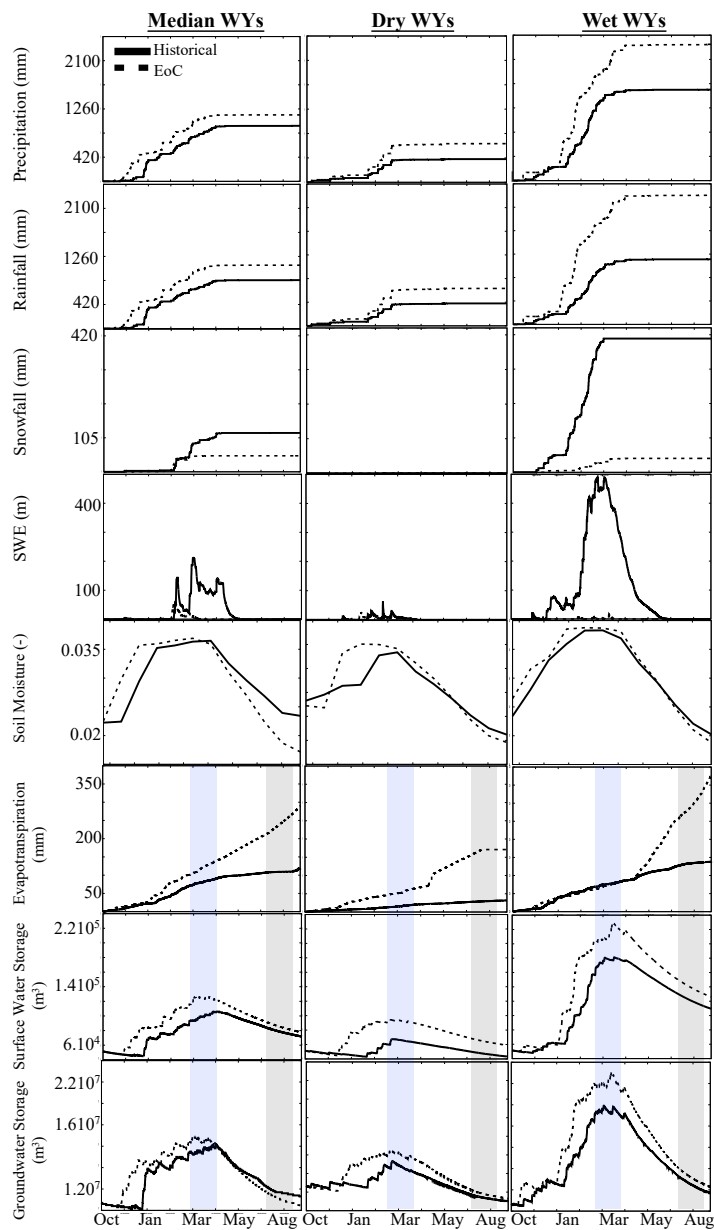

Figure 6: Temporal variations of the total cumulative precipitation, rainfall, and snowfall at the

watershed scale, total *SWE* at the watershed scale, the average watershed values of soil moisture,

the cumulative watershed *ET*, and the total surface water, and groundwater storages at the

watershed scale associated with the six historical and EoC Water Years (WY). The blue area

indicates the selected peak flow period while the gray area corresponds to the selected baseflow

conditions for the spatial distribution analyses.


### 3.3.1. Median water years

As indicated in section 3.1, the EoC median WY has more precipitation than the historical
median WY. The EoC precipitation comes mainly as rain due to the warmer temperatures of the
EoC and includes virtually no snowfall from late winter to early spring. This precipitation phase-
change combined with the earlier snowfall cessation date in the WY results in minimal and even
non-existent *SWE* in the Cosumnes watershed for much of the WY, a significant change compared
to historic conditions. EoC peak *SWE* occurs in February in contrast to the historical peak *SWE*,
which occurs in April. Due to the watershed's relatively low elevation, snow accumulates only in
the upper part of the Cosumnes watershed (~10% of the total watershed area). Only areas located
in the highest elevations (> 2000 m), such as the eastern limit of the watershed, show any *SWE* in
the EoC simulations whereas in the historical WYs we observed *SWE* as low as 1000 m.
The decrease in snow and the increase in rain along with an earlier onset of seasonal
precipitation directly impacts soil moisture, which sees an early increase with a slightly higher
peak than historical. As more water is available earlier in the EoC, the *ET* demand from increased
temperatures is met until substantially higher summer temperatures increase *ET* at a much faster
rate than the historical WY. The high EoC *ET* and the lack of snowmelt cause the soil to rapidly
dry from late-spring through late-summer.
Because of the marked increase in total precipitation and shift from snow to rain in the EoC
simulations, surface water storage generally increases throughout the WY. This is consistent with
previous studies (Gleick, 1987; He et al., 2019; Maurer, 2007; Safeeq et al., 2014; Son & Tague,
2019; Vicuna & Dracup, 2007; Vicuna et al., 2007). Surface water storage increases in early
November in the EoC simulations while in the historical simulations this increase occurs in
January. Similar to the earlier peak $SWE$ and soil moisture, the peak surface water storage in the
EoC is also earlier (January through February) compared to the historical period (March through
April). This late-season surface water storage remains larger because the accumulated precipitation
is large enough to overcome the increased $ET$ in a warmer climate. Similar to surface water storage,
groundwater storage increases earlier and peaks at a larger amount than the historical WY.
However, in contrast to the surface water storage, the groundwater storage during baseflow
conditions is lower in the median EoC compared to the median historical year. This decrease in
groundwater during baseflow conditions is due to the lack of snowmelt and higher EoC $ET$. In late
spring and summer in the EoC, groundwater keeps depleting through $ET$ and is not recharged by
snowmelt through surface and subsurface flows from the Sierra Nevada as in the historical period.
This may indicate that compared to surface water storages, groundwater storage may be more
sensitive to EoC hydroclimatic changes (which are multi-fold, and in this case include an increase
in precipitation, a transition from snow to rain, and higher $ET$). One way to quantitatively measure
this sensitivity is to compare the seasonal change in water storage between peak and baseflow
conditions. Historically, changes between peak and baseflow conditions (i.e., the amount of water
lost between peak and base flow) resulted in moderate seasonal changes in groundwater storage
(30%) and surface water storage (32%). The EoC simulations reveal larger seasonal variation for
groundwater and surface water storage (40% and 37% decreases, respectively). Groundwater in
the Cosumnes Watershed is mainly recharged in the headwaters and stored in the Central Valley.
Therefore, these Central Valley aquifers experience earlier and larger increases in storage which
lead to more water available to $ET$ and therefore aquifer depletion. A deeper understanding of this
phenomenon requires an analysis of the spatial patterns of these changes which is performed later
on in this study.

### 3.3.2. Dry water years


All EoC WYs are characterized by higher precipitation in the form of rainfall compared to
their historical counterparts. The historical dry WY has ~43% less total precipitation than the EoC
dry WY. However, we note that for the EoC dry WY the decrease in snowfall is less drastic than
the median or wet EoC years. This is because the historically driest WY is significantly warmer
than the historical average WY, and therefore already has a smaller snowpack, 94% lower than the
historical median WY. The EoC dry WY *SWE* also accumulates two months earlier than the
historical *SWE*. Because the differences in *SWE* between the dry WYs are smaller than the
differences in *SWE* between the median WYs (7% versus 91%), we can deduce that the early and
larger rise in soil moisture in the EoC dry WY is mostly due to an earlier and larger amount of
rainfall. The higher soil moisture and EoC temperatures result in higher *ET* throughout the WY
compared to the historical WY. This *ET* results in lower soil moisture by the end of the summer,
similar to the median WY. In addition, surface water storage peaks earlier and at a larger amount
compared to the historical WY. The surface water storage in the EoC remains higher throughout
the WY compared to its historical counterpart despite this higher *ET* due to the low precipitation
associated with the historical dry WY. We further note that the difference in surface water storage
during baseflow conditions between the two dry WYs is higher than the difference between the
two median WYs. The groundwater recharge starts two months earlier in the EoC driest WY
compared to the historical driest WY due to the changes in timing and magnitude of precipitation.
However, it is interesting to note that groundwater storage during baseflow conditions in the EoC
WY is nearly equal to the historical WY (within 3%). Thus, although more water enters the EoC
dry WY system through greater precipitation, it eventually exits by the end of the WY and no
considerable net gains to groundwater are observed. This significant reduction in groundwater
storage from late winter to end-of-summer is a result of the much larger EoC *ET* and highlights
the dynamic nature of the EoC dry year watershed interactions. Also similar to the median WY,
dry WY seasonal decreases in EoC storage are more pronounced in the groundwater signal (36%)
than in the surface water signal (33%). We further note that the decreases in groundwater and
surface water storages are, as in the median WY, larger (+8%) than the historical decreases.

### 613    3.3.3.  Wet water years

The EoC wet WY is significantly wetter than all other WYs. Yet, unlike the historical WY,

the precipitation largely comes as rain, as shown by the low-to-no snowfall and *SWE* totals (Figure
6). The difference in future versus contemporary wet WY *SWE* (99%) is larger than the differences
between the median and the dry WYs (91%). As in other WYs, soil moisture increases earlier
compared to the historical wet WY. A greater water availability enables the system to meet the
high EoC *ET* demand. Hence, *ET* in the EoC wettest year remains higher than the historical wettest
year *ET* throughout the WY. However, the increase in *ET*, combined with the lack of snowmelt
that can buffer and recharge soil moisture in spring, leads to less soil moisture at the end of the
WY compared with the historical WY. Further, surface water storage increases earlier and at a
much faster rate in the EoC WY compared to the historical WY. This is mirrored in the
groundwater storages. As in the other EoC simulations, when compared to the historical
counterpart the EoC wettest year shows a sharper decline in seasonal above and below groundwater
storage changes (occurring between peak flow and baseflow). Groundwater storage decreases 47%
in the EoC between peak flow and baseflow, whereas only a 41% decrease occurs in the historical
wet WY. Similarly, surface water storage decreases 44% in the EoC whereas only a 41% decrease
occurs in the historical wet WY.

**3.4. Spatial patterns of the changes in fluxes and pressure-heads**
**3.4.1.  Median water years**
To provide a deeper understanding of how the changes in precipitation timing, magnitude,
and phase affect the land surface processes and surface and subsurface hydrodynamic responses,
we assess the spatial patterns of these changes during two key periods in the WY, peak flow and
baseflow. Figure 7 shows the percent changes in $ET$, surface water pressure-heads, and subsurface
pressure-heads (i.e., pressure-heads of the model bottom layer) in the EoC median WY compared
to the historical median WY during peak flow and baseflow conditions (see the time frames in
Figure 6). Regions in red correspond to areas with smaller fluxes or pressure-heads in the EoC
compared to the historical ones, whereas regions in blue correspond to areas with larger fluxes or
pressure-heads in the EoC compared to the historical median WY. We study peak flow and
baseflow conditions because the analysis of the temporal variations of fluxes and storages has
shown that these two periods are characterized by different trends and represent the key periods in
understanding the hydrologic responses to the EoC extreme climate.
Relative to the historical median WY, during peak flow the EoC median WY is
characterized by an increased $ET$ across the majority of the watershed, especially in the Central
Valley, and larger surface water and subsurface pressure-heads (Figure 7a-c). $ET$ increases in the
EoC both because of the increase in water availability and increased evaporative demand, as
discussed in the previous section (3.3.1.). The increase in $ET$ is non-uniform across the watershed
because of the heterogeneity of the landscape's topographical gradients, land-surface cover, and
subsurface geological conditions. The Central Valley is characterized by a large increase in *ET*
compared to the Sierra Nevada, and the patterns of *ET* in the Central Valley are also more
homogeneous, a resultant of the geological characteristics of the area and the hydroclimate of the
watershed (i.e., where most of the precipitation falls over the Sierra Nevada but follows
topographic gradients downward into the valley where more recharge occurs). This leads to more
water available in the Central Valley compared to the Sierra Nevada characterized by less
permeable rocks. In addition, as most of the *ET* in the Central Valley comes from evaporation due
to the high temperatures of the EoC (not shown here), the increase in evaporation is higher in the
Central Valley due to its aquifers characterized by a high permeability (Maina and Siirila-
Woodburn, 2020) and the availability of water.

Surface and subsurface pressure heads both show general increases during the EoC peak

flow, yet these maps reveal that unlike *ET* the pressure head (and therefore storage) of water is
very heterogeneous in space. For example, in the Sierra Nevada, we observe an increase in
subsurface pressure-head (Figure 7c) only in some relatively permeable areas susceptible to
infiltration and recharge. Although the Central Valley aquifers are more permeable and
geologically less heterogeneous than the Sierra Nevada (as defined in the model), the changes in
subsurface pressure-head in the Central Valley are heterogeneous. This is because the recharge of
the Central Valley aquifers is dependent on the subsurface and surface flows from the headwater
(i.e., connectivity to the headwater).  In other words, only areas of the Central Valley that are
subject to stronger connectivity with the headwaters see an increase in subsurface pressure-head
in the EoC, likely because they are more regularly recharged by the headwaters through surface
and subsurface flows from these areas, a recharge that buffers the water depletion through *ET*.
These are mostly the areas located close to the streams where there is an exchange between the
subsurface and the surface and the Sierra Nevada foothills (in the alluvium 3 area, see Figure 1).

Relative to its historical counterpart, the EoC median WY is characterized by high *ET*

during baseflow conditions though less than during peak flow conditions. (Figure 7d). We observe
larger surface water pressure-heads in higher-order streams whereas surface water pressure-heads
decrease in the EoC in the majority of the low-order, ephemeral streams (Figure 7e). This
opposition of spatial pattern trends, resulting in more water in the main river channels, and less in
the smaller streams, occurs for several reasons. First, peak flow occurs earlier in the EoC and is
more rainfed, so that the ephemeral streams drain earlier in the EoC compared to in the historical
period. This sustained and longer duration of draining increases the surface water pressure-head
along the main river channels and is due to the contribution of the subsurface in the headwaters.
This contribution is also higher in the EoC due to larger amounts of precipitation. The trends along
the main river channel are also evident in the subsurface pressure-head maps (Figure 7f). Because
the surface water is larger along the main channels, the subsurface pressure-heads are also larger
here due to the interconnection between the subsurface and the surface (Figure 7f). However, in
general, subsurface pressure-heads decrease elsewhere in the EoC during baseflow because of the
lack of snowmelt and the higher *ET* demand. This result highlights the spatiotemporal complexity
of an expected watershed's response to changes in climate (shown here to be bi-directional), and
how factors such as river proximity may be crucial for consideration.

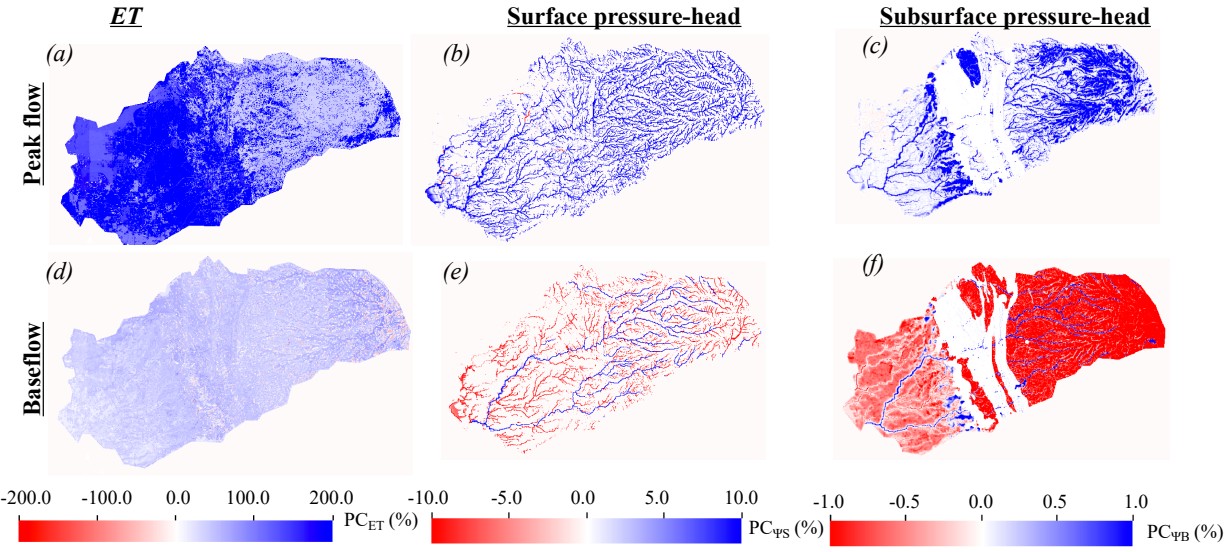

Figure 7: Comparisons between EoC median water year (WY) and the historical median WY peak flow and baseflow spatial distributions of percent changes in *ET* ($PC_{ET}$), surface water ($PC_{\Psi S}$) and subsurface ($PC_{\Psi B}$) pressure-heads. Regions in red correspond to areas with smaller fluxes or pressure-heads in the EoC compared to the historical ones, whereas regions in blue correspond to areas with larger fluxes or pressure-heads in the EoC compared to the historical WY.

### 3.4.2. Dry water years

Figure 8 illustrates the percent changes in *ET*, surface water, and subsurface pressure-heads in the EoC dry WY compared to the historical dry WY during peak flow and baseflow conditions. During peak flow conditions, the EoC dry WY has larger *ET*, surface, and subsurface pressure-heads than the historical dry WY (Figure 8a-c). *ET* is larger in this EoC dry WY not only because it is hotter, but also because there is more precipitation, as noted previously. Increases in surface pressure-heads are non-uniform across the domain. For example, surface water does not increase in high elevation areas (i.e., elevation > 2000m) in the EoC dry WY because the change in the precipitation phase is not significant. The main difference between the EoC and the historical dry

WY is the amount of the water flowing down gradient, which is higher in the EoC, hence the
surface water in the EoC becomes higher downstream. The increase in subsurface pressure-heads
in the EoC dry WY during peak flow conditions is heterogeneous with patterns similar to the
changes in subsurface pressure-heads associated with the EoC median WY.

During baseflow conditions, even though *ET* increases in the EoC driest WY relative to

the historical driest WY, surface, and subsurface pressure-heads also generally increase (Figure
8d-f). Given wetter conditions in the driest EoC WY, first-order streams are more pronounced. A
few low-order streams have less surface water in the EoC when compared to the historical dry
WY, similar to the results of the median WYs (see section 3.4.2). Subsurface pressure-head is
generally larger in areas subject to strong connectivity with the headwaters (i.e., receiving more
water from the headwaters through subsurface and surface flows) in the EoC dry WY relative to
the historical dry WY, with some regions experiencing no change from the historical conditions.
This suggests that the larger amount of precipitation associated with the EoC dry WY is sufficient
to supply enough water to account for high *ET* demands and recharge the groundwater.

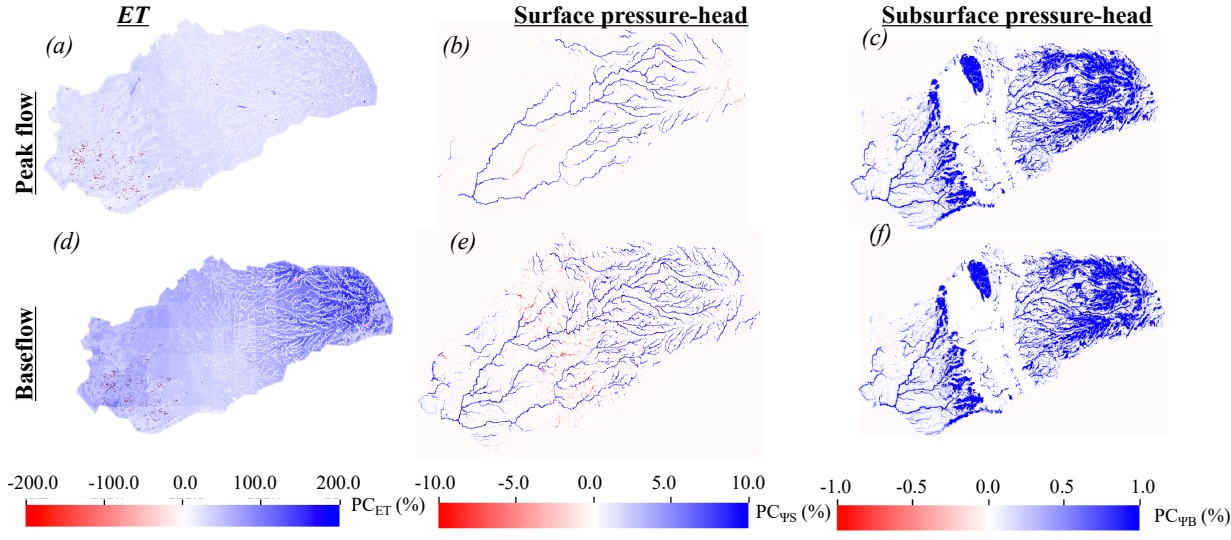


Figure 8: Comparisons between EoC dry water year (WY) and the historical dry WY peak flow
and baseflow spatial distributions of percent changes in *ET* ($PC_{ET}$), surface water ($PC_{\Psi S}$) and
subsurface ($PC_{\Psi B}$) pressure-heads. Regions in red correspond to areas with smaller fluxes or
pressure-heads in the EoC compared to the historical ones, whereas regions in blue correspond to
areas with larger fluxes or pressure-heads in the EoC compared to the historical WY.

### 3.4.3. Wet water years

Figure 9 shows the percent changes in *ET,* surface water, and subsurface pressure-heads in
the EoC wet WY compared to the historical wet WY during peak flow and baseflow conditions.
During peak flow, the EoC wet WY is characterized by larger *ET* and subsurface pressure-heads
relative to the historical wet WY and a more heterogeneous mixture of regions with both higher
and lower surface water conditions throughout the catchment (Figure 9 a-c). Analogous to other
WYs at EoC, the surface water pressure-head increases (decreases) are apparent in larger-order
(smaller order) streams, both in the Sierra Nevada and in the Central Valley. In the wettest WY,
this occurs for several reasons. First, the larger volume of precipitation, plus seasonal shifts in
precipitation timing result in the filling of the higher-order streams and depletion of the lower-
order streams during peak flow. Second, in the historical wet WY, a significantly greater amount
of snowpack is present in the Sierra Nevada in the upper elevation of the headwaters, allowing for
slower, steadier amounts of water that is released during the spring via snowmelt, and in turn,
supporting low-order streams over a longer period of time. The latter effect is immediately visible
in Figure 9e, where decreases in EoC surface pressure heads are visible in the headwaters, despite
the watershed-total showing an increase in EoC surface water storage during baseflow (see Figure
6). Similar to the two previous EoC WYs, the subsurface pressure-head increases are shown more
distinctly in the Central Valley during peak flow, under the main river channels, and in the foothills
during baseflow (see previous sections on the discussion of hydroclimatic and geologic impacts).

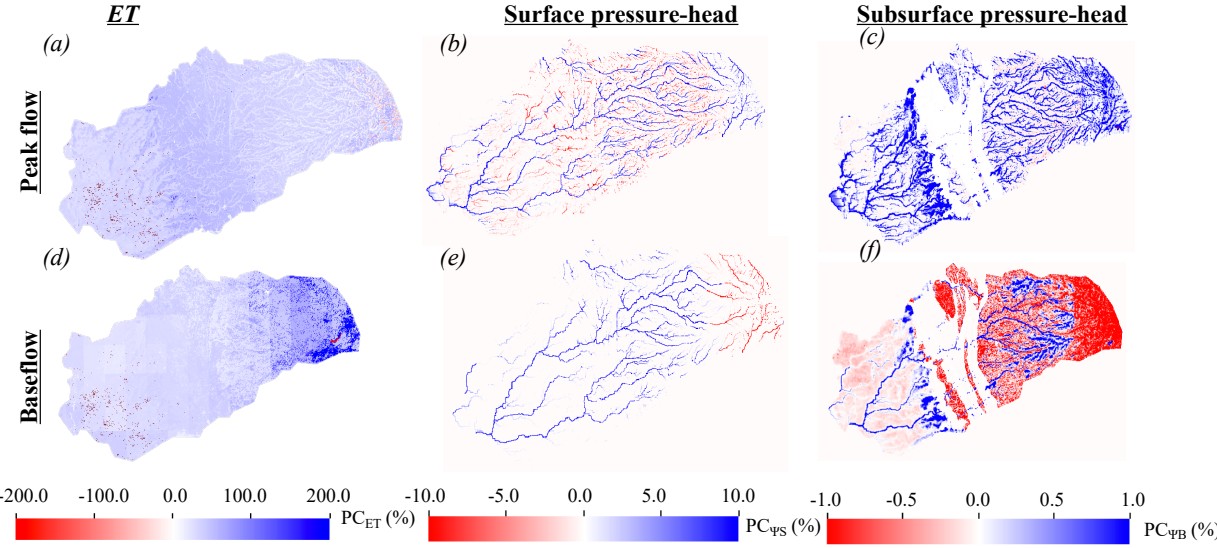

Figure 9: Comparisons between EoC wet water year (WY) and the historical wet WY peak flow and baseflow spatial distributions of percent changes in *ET* ($PC_{ET}$), surface water ($PC_{\Psi S}$) and subsurface ($PC_{\Psi B}$) pressure-heads. Regions in red correspond to areas with smaller fluxes or pressure-heads in the EoC compared to the historical ones, whereas regions in blue correspond to areas with larger fluxes or pressure-heads in the EoC compared to the historical WY.

## 4. Discussion

### 4.1 Comparison with previous studies

Some of the results presented in this study are qualitatively in agreement with previous studies yet provide important new insights. For example, Maurer & Duffy, (2005) used 10 global climate models to predict, as in this study, an increase in winter flows with an earlier peak flow timing in the WY and a decrease in summer flows. Maurer & Duffy show that mid-century projected annual precipitation and streamflow increases of 7% and 13% (respectively). Although our study focused on EoC projections, we found that compared to the historical median WY, annual surface water will increase by 19% in the EoC median WY. Compared to their findings,

our work sheds light on how these changes in runoff will occur across the watershed based on its
physical characteristics and highlights that while runoff will increase in the EoC lower-order
streams mainly located in the Sierra Nevada will see a decrease due to the change in the
precipitation phase. Mallakpour et al., (2018) also had a similar finding in a study that shows that
future California streamflow is altered similarly to Maurer & Duffy, (2005) under both the RCP4.5
and RCP8.5 emissions scenarios, with RCP8.5 showing the highest changes during peak flow.
However, contrary to our work the authors mentioned that the annual changes in streamflow will
not be significant probably due to the compensation between increases in peak flow and decreases
in baseflow. This was likely shaped by the differences in climate and hydrologic models used to
derive these conclusions. Similar changes in streamflow were obtained by He et al., (2019) who
drove the hydrologic model VIC with 10 global climate models to understand potential changes in
runoff in California due to climate change. Hydrologic changes computed from the 10 global
climate models were consistent and robust and showed an increase of around 10% in annual
streamflow by the late century, a percentage similar to what has been found in this study. The
authors mentioned that watershed characteristics such as geology, topography, and land cover
strongly impact the hydrologic response to climate change. Relationships between watershed
characteristics (e.g., physiographic parameters) and its responses to climate change were further
explored by Son & Tague, (2019) who highlighted that because vegetation and subsurface geology
control both water availability and energy demand, they in turn influence watershed sensitivity to
a changing climate as shown in this study.
The increases in groundwater storage shown in this study are also in agreement with
Niraula et al., (2017) who used the hydrologic model VIC to show that groundwater recharge will
likely increase in the northern portion of the western United States in a changing climate. However,
contrary to their work that estimates changes in groundwater recharge over a large domain (i.e.,
the western United States). In this work, we show that groundwater recharge decreases in the
summer in some areas due to the lack of snowmelt and high EoC *ET*. Increases in *ET* in response
to global warming were also documented by Pascolini-Campbell et al., (2021) who showed a 10%
increase in global *ET* from 2003 to 2019.

An advantage of our approach is a more explicit estimate of spatiotemporal changes in

groundwater-surface water feedbacks because Parflow-CLM physically solves the transfer and
movement of water from the bedrock to the canopy. Additionally, the aforementioned studies used
different emission scenarios and models to project changes in hydrology, nonetheless, their results
have shown that the directions of the observed changes are consistent across models and emission
scenarios and only the magnitude of these changes is uncertain. Hence, the trends observed in this
study using a single model and emission scenario likely represent the trends we would observe
using different models and scenarios. While our results show similar patterns and changes, our
study provides a much finer-grained perspective on the sensitivity of a watershed to changes in
climate extremes based on its subsurface geology, topography, and land cover. It also highlights
that the spatiotemporal analyses of these changes may reveal different trends than if only assessed
as annual changes. Understanding these localized changes and sensitivities is critical and has
practical implications for water management.

**4.2 Implications for water resources management**

Because our work provides a better understanding of the spatiotemporal changes in

hydrodynamics in response to future extremes, our findings also have important implications for
water resources in California. While previous work more broadly focused on how temperature
increases will alter the precipitation phase and reduce seasonal snowpack and increase winter
runoff, this work brings new physical and more granular insights into how watersheds may respond
to climate extremes. In particular, both wet and dry WYs in the future experience increased
precipitation. As such, even in future dry WYs, water managers and stakeholders may need to
prepare more for large precipitation events that may increase the possibility of flooding and require
new infrastructure management strategies. For example, in a future where WYs are generally
wetter, having alternatives for water supply during periods of sustained drought could be less
important. However, as we show in this paper, shifts in precipitation timing, phase, and magnitude
have cascading impacts on soil moisture profiles and *ET* withdrawals, which subsequently impact
discharge and groundwater dynamics. Future shifts in water availability earlier in the year, as well
as more dynamic transitions between peak and baseflow conditions (as quantified here), may
impose stresses on water distribution, especially those systems already under scrutiny (e.g. those
resources over-allocated or facing environmental degradation).

In addition, while these projections show increases in surface water and groundwater

storages at watershed-scale, our results also highlight important localized spatiotemporal changes
across a watershed, where the assumption of water storage increase does not necessarily hold in
all geographic locations (e.g., areas that are not close to the river in the Central Valley). Our study
also shows that the decreases in groundwater storage in the Central Valley aquifers are more
significant than the decreases in surface water storage during baseflow conditions. This may call
for new conveyance infrastructure that can move water from the relatively wetter areas to the drier
areas and/or where infiltration can more readily occur. The latter suggests solutions such as
Managed Aquifer Recharge (MAR) could become an increasingly important climate change
adaptation. Finally, our study also highlights that lower-order streams will likely become more
ephemeral in the EoC due to flashier runoff and higher evaporative demand, such conditions will
have important implications for fish spawning and ecosystem nutrient cycling. Although our
results are embedded with uncertainties and are based on a single projection and model, they do
highlight the need for a revisitation of current water management strategies. Further studies using
different climate and land-use scenarios and models of varying complexity and resolution could
help build more confidence and provide more information in defining how future water
management strategies would need to change to be more resilient to more extreme WYs in the
future.

**4.3 Study limitations**
This study combines novel climate and hydrologic simulations that provide both
advantages and disadvantages compared with previous work (He et al., 2019; Maurer & Duffy,
2005; Niraula et al., 2017; M. Safeeq et al., 2014; Son & Tague, 2019). We note several of these
disadvantages below. In the integrated hydrologic model, the subsurface geology and land cover
characterization has inherent and, in some cases, irreducible uncertainty. This study uses
hydrodynamic parameters as defined by Maina et al. (2020a), which assumes that the subsurface
hydrodynamics from the Sierra Nevada to the Central Valley is almost completely hydrologically
separated except through overland flow. However, it is not clear whether fractures or other
macrostructures may drive more surface and subsurface flows from the headwaters to the Central
Valley aquifers. In addition, we use the historical land surface cover map when simulating the
EoC. Since vegetation will dynamically respond to a changing climate, the land surface cover used
in the EoC simulations may be unrealistic and may influence, for example, *ET* and/or soil moisture.
For example, it has been shown that the stomatal resistance of plants will change due to rising $CO_2$
with important implications for both the water and energy balance (Lemordant et al., 2018; Milly
& Dunne, 2017). Yet, our use of historical land surface cover does have the advantage of isolating
changes in fluxes associated with climate change alone and could be compared in future work with
additional simulations that account for both changes in the land surface and climate. Future studies
will assess the impact of changes in vegetation physiology and land surface cover on watershed
hydrodynamics. In this study, we did not include the impacts of anthropogenic activities such as
pumping and irrigation due to the uncertainties in predicting these fluxes in EoC. While these
human interventions could substantially change the hydrologic system, our study isolates the
impacts of a changing climate on the natural system. Future studies can now estimate the impacts
of different pumping and irrigation scenarios at EoC that may further impact the hydrologic system
hydrodynamics in a changing climate and compare and contrast with this work. Although our VR-
CESM simulations represent a cutting-edge global climate model simulation (e.g., 28 km regional
grid-refinement, coupled atmosphere-land simulation with prescribed ocean conditions, etc.),
further work may be needed to evaluate how a more refined grid resolution impacts atmospheric
process representation over the Cosumnes watershed, particularly in the headwaters (Maina et al.,
2020b). We further acknowledge that the 30-year simulation may not be sufficient to capture
certain climate extremes (e.g., 1-in-50-year storm). Future studies, if computational resources are
available, will seek to explore how the use of a longer time period might influence the
identification of the most extreme dry and wet WYs from VR-CESM.
In this study, we relied on deterministic models to represent both the atmospheric (VR-
CESM) and hydrologic (ParFlow-CLM) dynamics. These models are very sensitive to the initial
conditions and input parameters (La Follette et al., 2021; Lehner et al., 2020; Song et al., 2015)
which are uncertain given the lack of data characterizing the above and below-ground
environment, including its hydrological response. Thus, while it is important to assess the
sensitivity of the model outputs to these uncertain parameters, these models are computationally
expensive and require many parameters. For example, a complete sensitivity analysis of the
hydrologic model requires running it thousands of times to explore the full parameter space (which
has a dimension of over 29). Such an approach is not feasible with the currently available
computational resources because it takes longer than one wall-clock day to simulate a single water
year for a single model parameterization, even in a high-performance computing environment.
Future work could employ reduced order models based on a subset of the physics-based model
runs to explore parameter space further (e.g. Maina et al., 2022). In addition, because of the
behavior of hydrological processes, the climate variability, and the uncertainties of deterministic
models, model validation should ideally be performed over a long period to account for different
changes and variabilities. In this study, model validation was limited to a period of 5 years due to
computational constraints. Although this period encompasses the wettest and driest years on record
in the region, we acknowledge that it may not be sufficient to capture the full range of hydrological
variability. Another limitation of using deterministic models is that the temporal variations of
hydrological processes tend to follow a stochastic behavior in accordance with the so-called Hurst
phenomenon (Hurst, 1951; Koutsoyiannis, 2003). As a result, the use of deterministic models such
as the ones employed in this study could intensify the impacts of hydrological extremes and climate
change. Finally, it has also been demonstrated that while the changes in water balance exhibit
greater variability on climatic scales, the most important changes in hydrologic processes remain
the overexploitation of groundwater (Ferguson and Maxwell, 2010) which has an impact on the
rise in sea level (Koutsoyiannis, 2020). In addition to projecting the use of groundwater by the end
of the century, future studies could compare the two approaches (deterministic and stochastic) to
better assess the limitations and the uncertainties associated with them.
**5    Summary and Conclusions**

The effects of climate change are increasingly felt across many regions of the world,

especially in hydrologically sensitive regions with Mediterranean climates such as California.
Many studies over the years have been conducted to better understand the hydroclimate of the EoC
and its impacts on the hydrologic cycle. Previous studies have used a multitude of different models
at varying complexity and climate scenarios to highlight that the future climate has multiple
plausible outcomes. Most of these studies indicate warmer temperatures and precipitation that
mostly falls as rain instead of snow. For example, the state of California is projected to experience
more punctuated climate extremes coupled with a marked decrease in the Sierra Nevada snowpack
(Cayan et al., 2008; Gleick, 1987; Musselman, Molotch, et al., 2017; Rhoades, Ullrich, &
Zarzycki, 2018). Such drastic transitions have already started to shape the hydroclimate of
California. Faced with this new normal, it is becoming increasingly important to assess how the
integrated hydrologic cycle may respond to these perturbations and connect these responses more
directly to water resource management, particularly with modeling frameworks that can better
represent the interactions between the changing atmosphere and the surface and subsurface
hydrology.

In this work, we used state-of-the-art physics-based models at high resolutions for their

respective communities to project changes in meteorological conditions at the EoC and assess how
their combined effects influence watershed hydrology from the land surface to the deeper
subsurface. Importantly, our approach to couple a variable resolution Earth System Model and an
integrated hydrologic model allow for us to simulate hydro-meteorological conditions which are
jointly driven by thermodynamical and dynamical shifts in climate. We model the Cosumnes
watershed, which spans the Sierra Nevada and Central Valley and hosts one of the last rivers in
the state without a large dam, as a testbed to understand how climate drivers will impact water
resources in the EoC. We performed climate simulations over 30-year periods historically (1985-
2015) and at EoC (2070-2100) and identified the driest, median, and wettest WYs from those
simulations, which were then used as meteorological forcing for the hydrologic model. Our
coupled simulations project that, for the Cosumnes watershed, temperature and precipitation will
both increase by the EoC across all WY types (wettest, median, and driest). In addition,
precipitation is projected to fall earlier compared to historical conditions and mainly in the form
of rain. For the median and wet WYs the precipitation season has earlier cessation dates, while the
dry EoC WY, which is wetter than its historical counterpart, persists significantly longer into the
spring. As a consequence of warmer temperatures, all WYs show a substantial decrease in *SWE*.
The shift of precipitation from snowfall to rainfall, as well as the increase in the amount of
precipitation and the early start of precipitation lead to an overall increase in soil moisture and
more water available to meet the higher EoC *ET* demand. Importantly, this increase in *ET* is
heterogeneous across the watershed and highlights one of the main advantages of using an
integrated hydrologic model such as the one we employed in this study to assess the spatiotemporal
patterns of change. Our results show that the sensitivity to the changes in *ET* at EoC depends on
the subsurface geology and topographical gradients.  More specifically:
● The geological and topographical complexities of the Sierra Nevada headwaters
lead to highly heterogeneous changes in *ET*. Changes in *ET* are higher in permeable
areas such as the plutonic rocks where water can be more easily extracted.

• *ET* changes in the Central Valley of the Cosumnes watershed are predominantly

uniform with the highest sensitivities in the vicinity of the Cosumnes River due to

the high availability of water.

Precipitation increases enough in the EoC to provide water for both increased *ET* and

increased surface water storage. Surface water storages also increase earlier in the WY and have
higher peak amounts. This earlier and larger increase is a direct consequence of an earlier start in
precipitation at EoC, a marked change in the precipitation phase, and an overall larger amount of
precipitation when compared with the historical WYs. However, our results also highlight that
during baseflow conditions surface water decreases, especially in lower-order streams, showing
that these areas are highly sensitive to the change in precipitation phase. Our simulations also show
that the seasonal variability of the EoC watershed behavior is also more dynamic. In general,
decreases in seasonal water storages occurring between peak flow and baseflow conditions are
more than 10% higher in the EoC compared to the historical conditions.

EoC groundwater storages are also projected to increase earlier in the WY with peaks

greater than those found historically. Yet these storages decrease significantly during baseflow
conditions due to the higher *ET* at EoC and the absence of recharge from snowmelt. Contrary to
the changes in surface water storages, groundwater storages show a larger decrease due to their
dependence on the surface water from the Sierra Nevada. Our results also show that changes in
subsurface pressure-heads are not uniform and are bi-directional throughout the Cosumnes
watershed. Because the connectivity between the Central Valley aquifers and the Sierra Nevada
headwaters (i.e., subsurface and surface flows from the headwater to the Central Valley aquifers)
plays an important role in the hydrodynamics of this watershed, only areas with a strong connection
with the headwaters, such as the foothills and the river channels, see an increase in subsurface
pressure-heads at EoC. However, the subsurface pressure-heads decrease elsewhere in the Central
Valley aquifers especially in baseflow conditions due to the high $ET$ and the lack of snowmelt. In
the river channels, this is due to the exchange between the subsurface and the surface whereas the
foothills characterized by the consolidated sediments serve as "spillover."

Our results provide novel understandings about possible changes in the integrated

hydrologic response to changes in EoC climate extremes. An important caveat is that our
simulation was a single set of climate realizations and may not properly bound internal variability
uncertainty like an ensemble of climate simulations could. However, beyond the widely agreed-
upon changes of decreased snowpack and shifts in runoff timing in the literature, we show that in
this simulation: 1) EoC precipitation increases even in the driest years; 2) despite an increased
temperature, and hence $ET$, both groundwater and surface water storage increase relative to
historical conditions because of increased precipitation; and 3) there is a distinct spatial pattern,
particularly in surface water storage, in which smaller-order streams see reduced flow while the
larger order streams see an increased flow. These changes will have strong implications on natural
resource management.

In this study, land cover changes are assumed to not occur, however, changes in land cover

are expected to occur in the future, either naturally or anthropogenically. Further vegetation
physiology will also change in response to an increase in $CO_2$. Thus, future studies should
investigate the impacts of these changes and how they may further alter the integrated hydrologic
budgets. Additionally, future studies could also assess the effects of anthropogenic activities such
as pumping and irrigation under a changing climate, other emissions scenarios, and/or the
sequencing of variable end-member WYs and the interannual memory of the hydrologic system.
Importantly, an understanding of this variability could be used to inform how water managers
might prepare for more intense and/or intermittent extremes in the future. Future research could
also use multiple emission scenarios to better assess the range in hydrodynamic responses
dependent on the severity of climate change, especially those related to the magnitude and spatial
location of the precipitation response since they are likely more uncertain and scenario-dependent
than the trends at the watershed-scale.

## Appendix A: Comparisons between VR-CESM and PRISM historical conditions


Figure A1 highlights differences in dry, median, and wet WY accumulated precipitation
relative to the 1981-2019 PRISM climatology.  VR-CESM generally recreates the spatial pattern
of anomalous dry and wet patterns across California for each WY type.  This is shown via the
common regions of minimum and maximum anomalies relative to the PRISM climatology.
Notably, there are regions where VR-CESM anomalies are not consistent with PRISM.  This is
primarily shown in the wettest water year in portions of the Central Valley, western slopes of the
Sierra Nevada, and southern California.  This is likely correlated with resolution and the lack of
orographic gradients (both valleys and peaks) in VR-CESM at 28km resolution.  Mismatches in
accumulated precipitation may also be due to representation of atmospheric rivers (ARs) in VR-
CESM that were found to be generally larger, slightly more long-lived and make landfall more
frequently over California (Rhoades et al., 2020b).  Figure A2 shows Cosumnes watershed WY
accumulated precipitation and surface temperature.  WY accumulated precipitation is shown in
Figure A 2a and 2b for PRISM and VR-CESM, respectively.  All WY accumulated precipitation
simulated by VR-CESM over 1985-2015 are within the range in PRISM, save for the wettest WY.
This is shown more explicitly in quadrant space in Figure A2c where the range of annual bias in
VR-CESM relative to the range of interannual variability in PRISM for accumulated precipitation
and temperature is shown.  VR-CESM generally simulates a wetter historical period over the
Cosumnes (range of bias of 1330 mm) relative to PRISM (range of interannual variability of 1320
mm).    Basin-average  minimum  (421  mm)  and  maximum  (1740  mm)  WY  accumulated
precipitation are slightly larger than is found in PRISM.  Of relevance to this study, PRISM has
shown  notable  uncertainties  in  the  Sierra  Nevada.    Lundquist  et  al.,  2015  showed  that  an
underrepresentation of the most extreme storm total precipitation in the Sierra Nevada can result
in an upper-bound uncertainty of 20% in WY accumulated precipitation. Therefore, the wettest
WY of VR-CESM is well within the 20% uncertainty range of PRISM's wettest WY (1580 ± 316
mm). Further, differences in basin-average WY accumulated precipitation between VR-CESM
and PRISM are non-significant using a t-test and assuming a p-value < 0.05. The range of
temperature bias in VR-CESM (2.74 °C) relative to the range of PRISM interannual variability
(2.93 °C) was also within the temperature uncertainties discussed in Strachan and Daly, 2017.
They showed that a general cool-bias in PRISM temperatures were found on the leeside of the
Sierra Nevada when compared with 16 out-of-sample in-situ observations across an elevation
gradient of 1950 to 3100 meters with an overall mean bias of −1.95 °C (maximum temperature)
and −0.75 °C (minimum temperature).

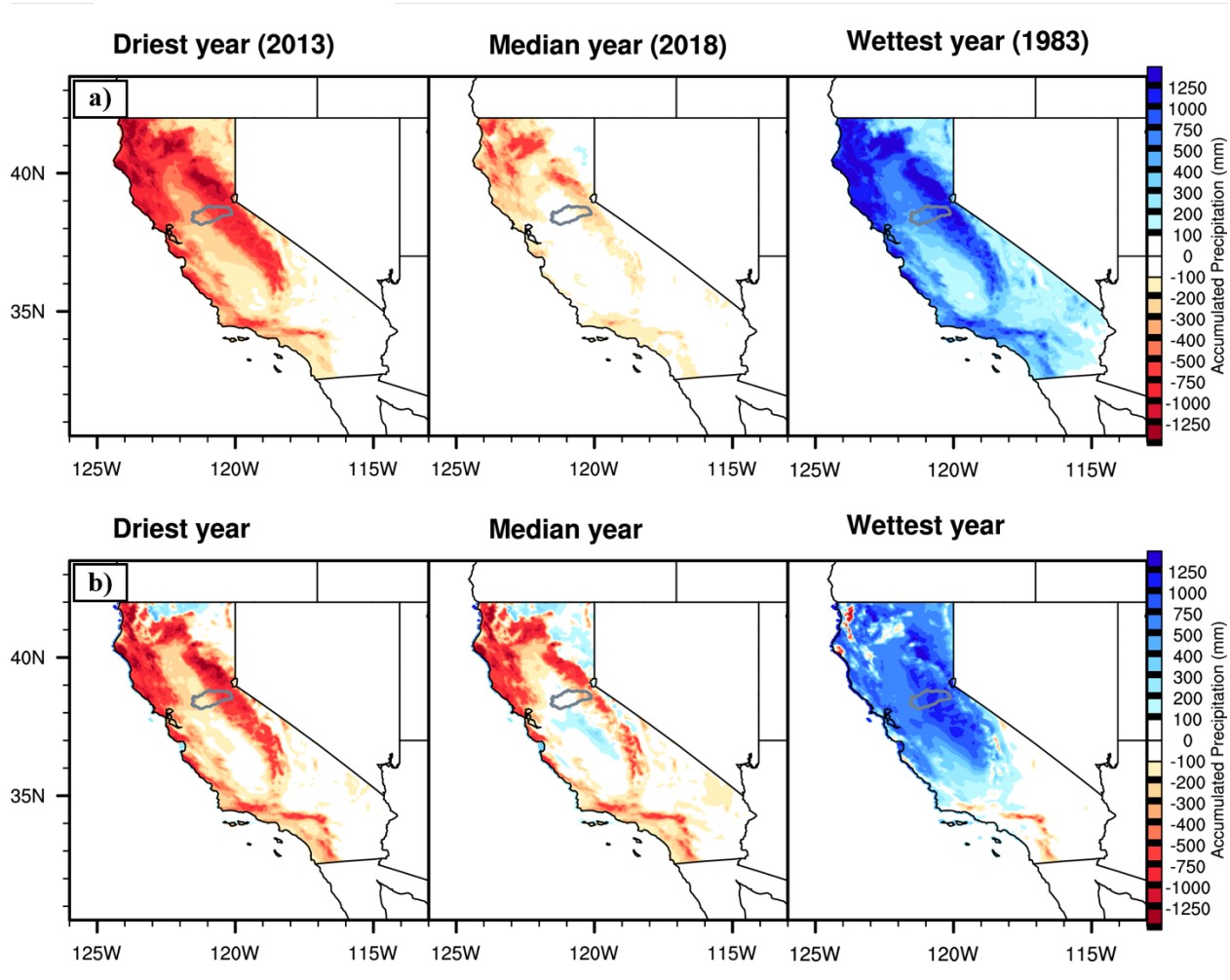


Figure A1: Differences in the driest, median, and wettest water year accumulated precipitation
over California in a) PRISM and b) VR-CESM relative to the 1981-2019 PRISM climatology.
The Cosumnes watershed boundary is outlined in gray.

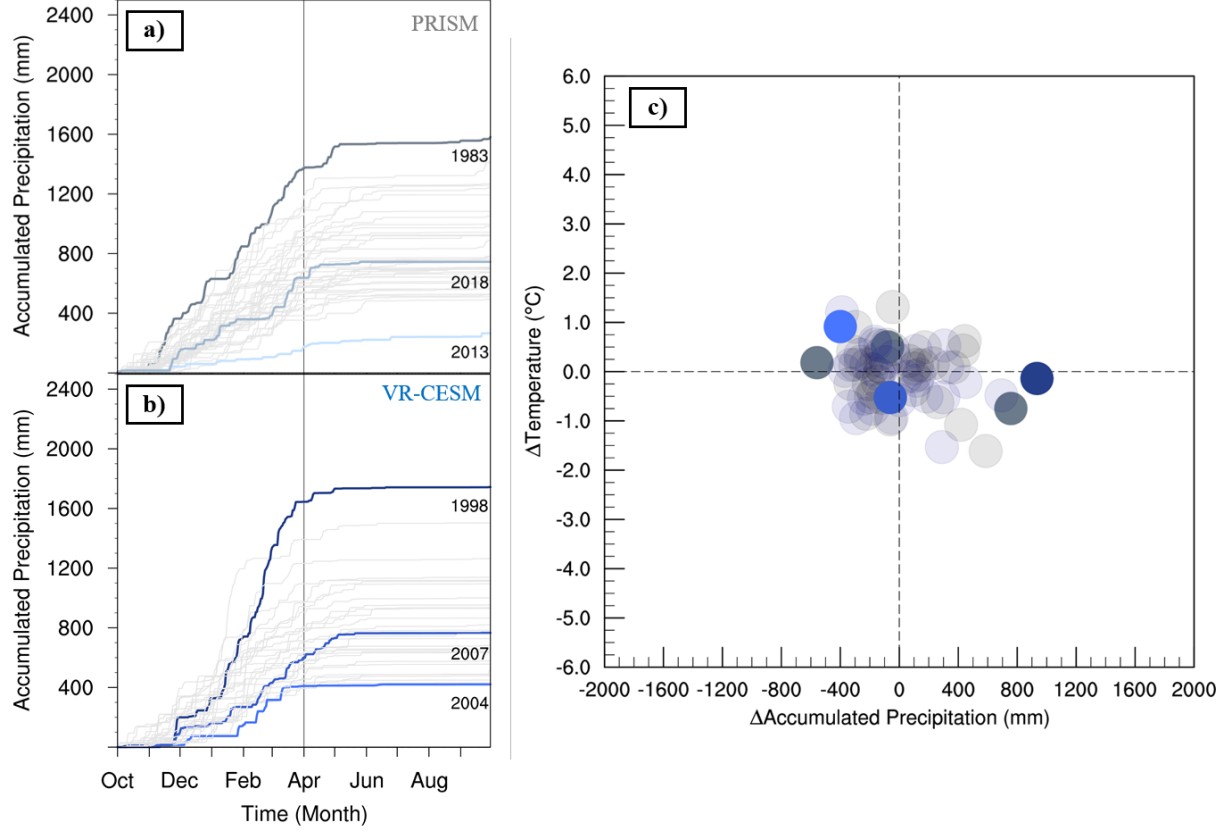


Figure A2: Cosumnes watershed accumulated precipitation totals in a) PRISM (gray; 1981-2019)
and b) VR-CESM (blue; 1985-2015) with dry, median, and wet years emboldened. c) shows
differences in PRISM (gray) and VR-CESM (blue) relative to the PRISM climatology (1981-2019)
in temperature and accumulated precipitation quadrant space.  Dry, median, and wet water years
are emboldened.


**Appendix B: Integrated Hydrologic Model Parameterization**
**1. Input Variables**

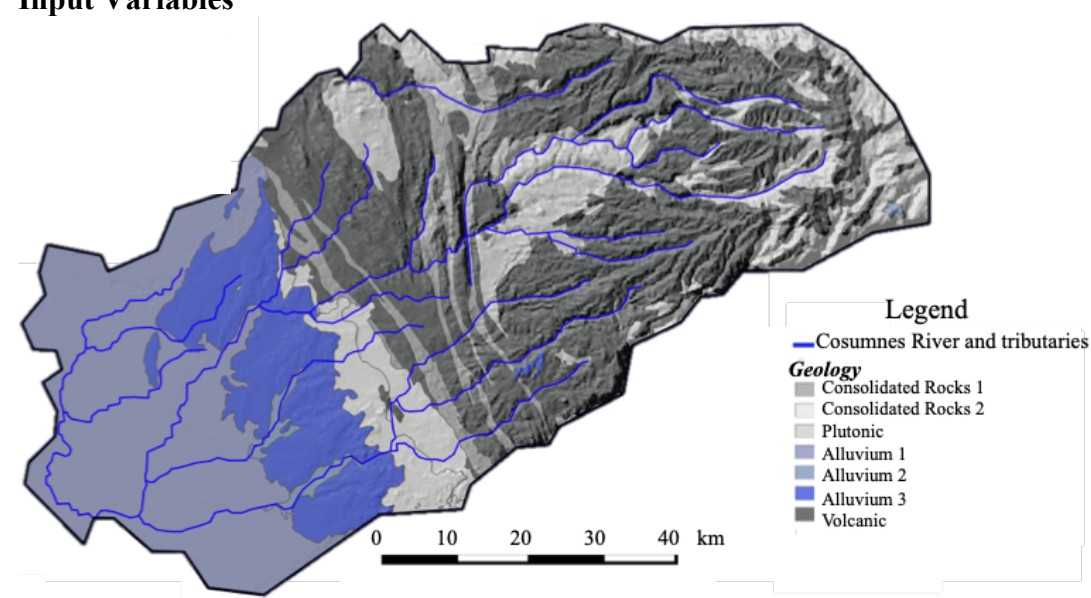

Figure B1: Geological map of the Cosumnes watershed (source: USGS, Jennings et al., 1977)

| Hydrodynamic properties based on the geology | | | | |
|---|---|---|---|---|
| Geological Formation | Porosity (-) | Specific Storage (m$^{-1}$) | Van Genuchten α (m$^{-1}$) | Van Genuchten n (-) |
| Bedrock (Consolidated, Plutonic and Volcanic Rocks) | 0.02 | $10^{-6}$ | 3.0 | 3.0 |
| Alluvial aquifers | 0.2 | $10^{-4}$ | 3.0 | 3.0 |

Table B1: Assigned values of hydrodynamic parameters (porosity, specific storage and Van
Genuchten parameters). Values are based on literature review (Faunt et al., 2010; Faunt and
Geological Survey (U.S.), 2009; Flint et al., 2013; Gilbert and Maxwell, 2017; Welch and Allen,
2014).

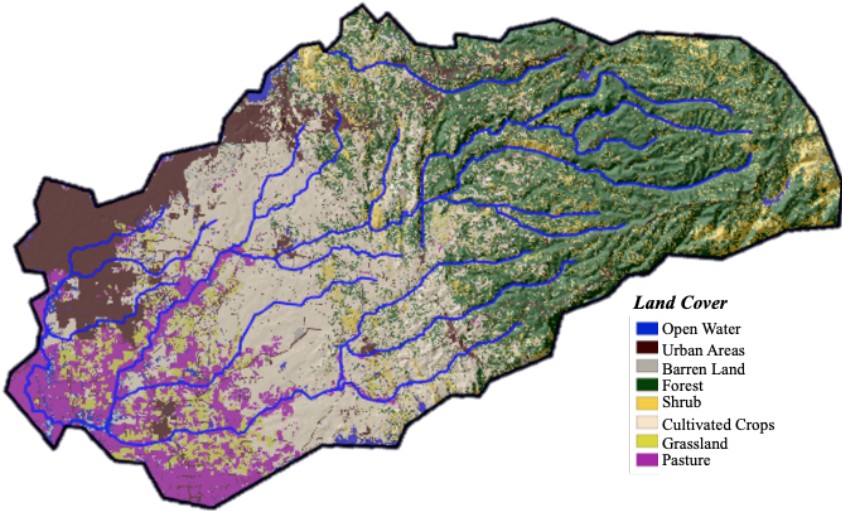

Figure B2: Cosumnes watershed characteristics: land use and land cover (source: Homer et al.,
2015), and model boundaries.

| Surface roughness based on land use | | | |
|---|---|---|---|
| **Land Use** | **Manning Coefficient (h.m$^{-1/3}$)** | | |
| Forest | $5 \times 10^{-2}$ | | |
| Shrub land and agricultural area | $5 \times 10^{-3}$ | | |
| Urban areas | $5 \times 10^{-5}$ | | |
| **Crop properties** | | | |
| **Crop Type and Reference** | **Height (m)** | **Maximum Leaf Area Index (-)** | **Minimum Leaf Area Index (-)** |
| Alfalfa (Evett et al., 2000; Orloff, 1995; Robison et al., 1969) | 0.6 | 6.0 | 2.0 |
| Pasture (Buermann et al., 2002; King et al., 1986; Rahman and Lamb, 2017) | 0.12 | 6.0 | 1.0 |
| Vineyards (Johnson and Pierce, 2004; Vanino et al., 2015) | 0.9 | 3.0 | 0.6 |

Table B2: Manning coefficients and crop properties

| Boundary conditions | Value |
|---|---|
| Mokelumne and American river | Weekly-varying Dirichlet boundary conditions. These values are based on the measured river stages. |
| Sierra Nevada limit | No flow Neumann boundary condition |
| Bottom of the model | No flow Neumann boundary condition |

Table B3: boundary conditions


**2.   Numerical model set-up**


| Domain size | ~7000 km$^2$ | | | | | | | |
|---|---|---|---|---|---|---|---|---|
| Spatial discretization | 200 m horizontal from 0.1 m to 30 m in the vertical direction | | | | | | | |
| | Vertical Resolution | | | | | | | |
| | Layer | 1 | 2 | 3 | 4 | 5 | 6 | 7 | 8 |
| | Δz(m) | 0.1 | 0.3 | 0.6 | 1.0 | 8.0 | 15.0 | 25.0 | 30.0 |
| Simulation time | Model validation (from water year 2012 to water year 2017), then future water years | | | | | | | |
| Temporal discretization | hourly | | | | | | | |

Table B4: Numerical model discretization
**3. Output variables**

| **Selected output variables** | **Temporal scale** | **Spatial scale** |
|---|---|---|
| Snow Water Equivalent | Yearly, monthly, and hourly | Domain-average and point scale |
| Evapotranspiration | Yearly, monthly, and hourly | Domain-average and point scale |
| Soil Moisture | Yearly, monthly, and hourly | Domain-average and point scale |
| River Stages (also surface water storages) | Yearly, monthly, and hourly | Domain-average and point scale |
| Groundwater levels variations (also subsurface storages) | Yearly, monthly, and hourly | Domain-average and point scale |

Table B5: Selected output variables



**Appendix C: Integrated Hydrologic Model Validation**
We compared temporal variations of streamflow at 3 stations located in the Sierra
(uplands), the intersection between the Sierra and the Central Valley, and the outskirts of
Sacramento (see Figure C1). Four wells in the watershed (see Figure C1) have reasonable, publicly
available records of groundwater levels and were used to check the ability of the model to
reproduce water table depth variations.

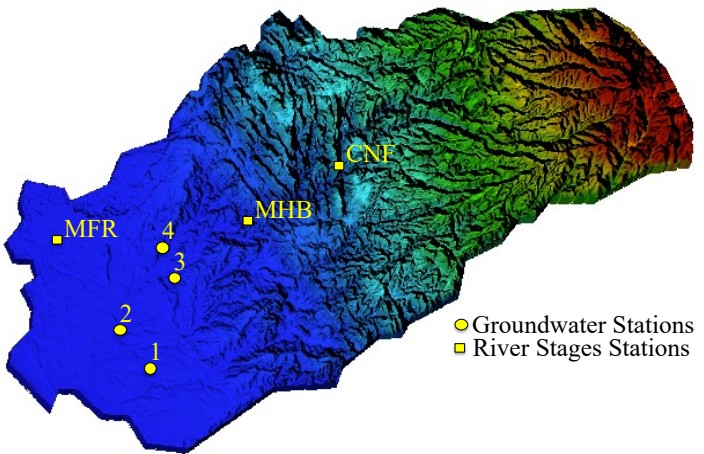


Figure C1: The locations of the 3 streamflow gauges (CNF, MHB, and MFR) and 4
groundwater wells (stars).

Figure C2a depicts the comparisons between simulated and measured river stages at the 3
stations indicated in figure C1. Absolute errors (L1) in m and relative errors (L2) are shown in
Table C1. Differences between simulated and measured streamflow vary between 0.4 and 0.8 m
(Table C1) indicating that the model is able to reproduce the river dynamics.
Absolute differences given by:
$$L_{1_{i,j}} = \left| X_{mes_{i,j}} - X_{sim_{i,j}} \right|$$ (C1)
Where $L_{1_{i,j}}$ is the absolute difference associated with cell i and time j, $X_{mes_{i,j}}$ is the
measured (or remotely sensed) data, and $X_{sim_{i,j}}$ the simulated value.
Relative differences $L_{2_{i,j}}$ are given by:
$$L_{2_{i,j}} = \frac{|X_{mes_{i,j}} - X_{sim_{i,j}}|}{X_{mes_{i,j}}}$$    (C2)

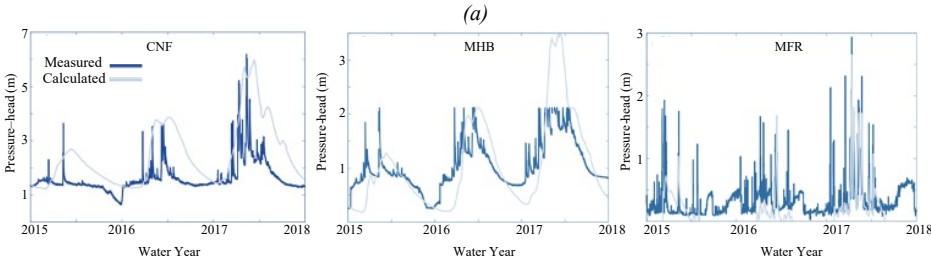

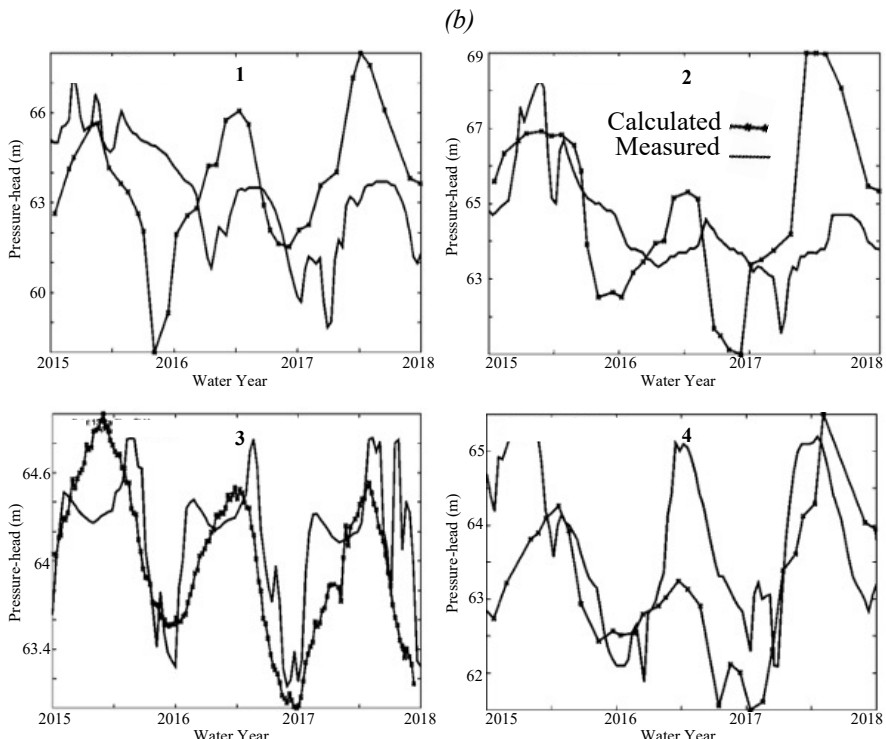

Figure C2: Comparisons between measured and calculated (a) river stages (i.e., pressure-
heads simulated by ParFlow-CLM) and (b) subsurface pressure-head. The location of the selected
points is indicated in Figure C1.

| Measurements | $L_1$ (m) | L2 (-) |
|---|---|---|
| River Stages (CNF) | 0.8 | 0.5 |
| River Stages (MHB) | 0.4 | 0.36 |
| River Stages (MFR) | 0.57 | 1.06 |
| Groundwater Levels (Well 1) | 3.73 | 0.05 |
| Groundwater Levels (Well 2) | 1.63 | 0.02 |
| Groundwater Levels (Well 3) | 0.476 | 0.0077 |
| Groundwater Levels (Well 4) | 1.08 | 0.016 |

Table C1: Differences between measured and calculated surface and groundwater levels. L1 is the
absolute error and R2 the relative error.

Comparisons between simulated and calculated groundwater levels (here referred to as the
pressure-heads at the bottom of the domain) shown in Figure C2b indicate that the model has
reasonable agreements with measurements. As shown in table C1, the error varies between 0.47 to
3.73 m depending on the station. Mismatches between simulated and observed groundwater levels
at wells 1 and 2 are likely due to an inaccurate estimation of pumping in these areas. The temporal
variations of the groundwater levels show an impact of withdrawals but because these withdrawals
are hard to estimate the model isn't correctly reproducing these trends.
ParFlow-CLM also solves the key land surface processes governing the transfer of water
and energy at the land-atmosphere-soil interface: evapotranspiration, snow dynamics, and soil
moisture. In Maina et al., (2020a), rigorous comparisons between the ParFlow-CLM simulated
land surface processes and remotely sensed estimates of these variables were conducted (Figure
C3). Table C2 shows the correlation coefficient between ParFlow-CLM results and the various
datasets compared.

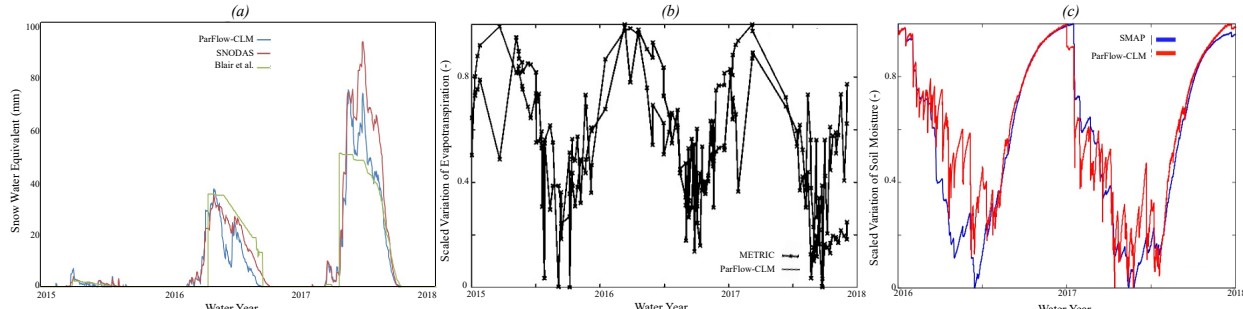


Figure C3: (a) Comparisons between domain-averaged total snow water equivalent obtained with
ParFlow-CLM, SNODAS and Bair et al., reconstruction, (b) Comparisons between actual
evapotranspiration obtained with ParFlow-CLM and METRIC (c) Relative variation of soil
moisture obtained with ParFlow-CLM and SMAP. Note that the x-axis of (c) is shorter because of
the availability of SMAP data

| Satellites based products | $L_1$ (m) | $L_2$ (-) | Pearson Correlation Coefficient |
|---|---|---|---|
| SWE SNODAS (mm) | 3.09 | 3.77 | 0.97 |
| SWE Bair et al., (mm) | 3.80 | 2.69 | 0.84 |
| Soil Moisture SMAP (-) | 0.217 | 3.07 | 0.94 |
| ET METRIC (mm/s) | 0.067 | 1.40 | 0.6 |

Table C2: differences between measured and remotely sensed evapotranspiration (METRIC), soil
moisture (SMAP), and snow water equivalent (SNODAS and Bair et al., 2016)

**Data availability**

Data    supporting    the    findings    of    this    study    can    be    found    here:
https://portal.nersc.gov/archive/home/a/arhoades/Shared/www/Hyperion/

**Author contribution**

The authors contribute equally to this work.

**Competing interests**

The authors declare that they have no conflict of interest.

**Acknowledgements**

Fadji Zaouna Maina and Erica Siirila-Woodburn were supported by LDRD funding from Berkeley
Lab, provided by the Director, Office of Science, of the U.S. Department of Energy under Contract
No. DE-AC02-05CH11231.
Author Alan M. Rhoades was funded by the Department of Energy, Office of Science Office of
Biological and Environmental Research program under Award Number DE-SC0016605 "A
framework for improving analysis and modeling of Earth system and intersectoral dynamics at
regional scales" and Award Number DE-AC02-05CH11231 "The Calibrated and Systematic
Characterization, Attribution, and Detection of Extremes - Science Focus Area".
This research used computing resources from the National Energy Research Scientific
Computing Center, a DOE Office of Science User Facility supported by the http://
dx.doi.org/10.13039/100006132 of the U.S. Department of Energy under Contract No. DE-
AC02-05CH11231.

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
