# Peer review of "Projecting end of century climate extremes and their impacts on the"

_Hydrology and Earth System Sciences, 2021_

## Referee Comment (RC2)

In this work, the authors present global climatic and hydrologic models to simulate the extremes and their impacts on the water-energy balance over California. The paper is well written and with high relevance to the hess journal. Please see some suggestions I kindly ask the authors to address:

1) The title of the paper is "Projecting the impacts of end of century climate extremes on the hydrology in California.". The title of the paper is a bit strong since it recommends that the whole hydrological-cycle has been modeled for the State of California and also for a time-window reaching the end of the century. Many authors struggle to simulate only one part of the hydrological-cycle of California (e.g., rainfall-runoff model, as for example in Yin et al., 2021; while many similar studies exist in literature). For such a promising title, a strong literature review should be performed to include similar studies for all hydrological-cycle variables and to show how the proposed model is more advanced.

2) There is a lack of calibration, validation, and verification of the proposed model. When a forecast is performed, one should use a part of the timeseries to calibrate-validate-verify their model, and then perform a forecast for the near future. I suggest the authors see/discuss this procedure concerning their own model. Also, the End of Century (EoC) forecast for such a large area is very optimistic in my opinion. Since climate dynamics is highly complex, I imagine that a forecast of only a few steps ahead is possible. If one is studying, for example, runoff on an annual scale, then after a couple of years, the variability of the forecast would be very wide, thus, reducing the credibility of the result (e.g., see Han et al., 2021). Also, the credibility of the outcome should depend on the available length of records. Here, the authors perform a forecast of 80 years ahead, which is double the length of records the authors use to construct the climatic and hydrologic model. I suggest to test/discuss how the variability/probability of the forecasts change as we move away from the present/historic data.

3) It is shown that due to long-range dependence effect to key hydrological-cycle processes (e.g., Dimitriadis et al., 2021) such as the ones the authors use, the variability of each climatic process would be even higher than, for example, under the assumption of zero auto- and cross- correlation (i.e., white noise). Please show/discuss whether the proposed model assumes a correlation function for the input variables. I also suggest the authors see/discuss whether their model forecasts also capture (and verify) the stochastic characteristics of the historical timeseries including the effects from climate change (such as marginal distribution function, auto-correlation function, etc.).

4) There are many equations in the text. Please consider creating a Table with all the inputs variables, output variables, boundary conditions, model assumptions, model limitations, simulation times, discretization method, etc., in order to help the readers identify the complexity/strength of the proposed model.

5) Please include more details on the water-energy balance equation and show whether is preserved in historical and forecasts. Also, have the authors included in the mass-energy balance analysis groundwater depletion in California (e.g., Badiuzzaman et al., 2017) and effects from sea level rise and ocean dynamics (e.g., Katsman et al., 2008)?

References

Badiuzzaman, P., E. McLauglin, and D. McCauley, Substituting freshwater: Can ocean desalination and water recycling capacities substitute for groundwater depletion in California?, J. Environ. Manag., 203, 123–135, 2017.

Dimitriadis, P., D. Koutsoyiannis, T. Iliopoulou, and P. Papanicolaou, A global-scale investigation of stochastic similarities in marginal distribution and dependence structure of key hydrological-cycle processes, Hydrology, 8 (2), 59, 2021.

Han, H., C. Choi, J. Jung, and H.S. Kim, Deep learning with long short term memory based sequence-to-sequence model for rainfall-runoff simulation, Water, 13 (4), p. 437, 10.3390/w13040437, 2021.

Katsman, C., W. Hazeleger; S. Drijfhout; G.J. van Oldenborgh, and G. Burgers, Climate scenarios of sea level rise for the northeast atlantic ocean: A study including the effects of ocean dynamics and gravity changes induced by ice melt, Climatic Change, 91, 351–374, 2008.

Yin, H., X. Zhang, F. Wang, Z. Yanning, R. Xia, and J. Jin, Rainfall-runoff modeling using LSTM-based multi-state-vector sequence-to-sequence model, J. Hydrol., 10.1016/j.jhydrol.2021.126378, 2021.

---

## Author Comment (AC1)

Below is an itemized list of all comments in plain text and our responses in blue.

**Anonymous Referee #1**

The manuscript simulates End of Century (EOC) extremes and their effects on the water-energy balance in the Cosumnes river basin, using cutting-edge global climate and integrated hydrologic models (ParFlow-CLM). I really like the way the authors used to analyze the hydroclimatic changes by median WY, dry WY and wet WY (e.g., Figures 3-5). The manuscript is overall clearly written, and the results are well discussed.

We thank the reviewers for their positive comments and feedback and for acknowledging the quality and the significance of our work.

My first concern is the insufficient validation of the models' simulations in the historical period. Besides temperature and precipitation outputs, other watershed-integrated fluxes, and storages (e.g., ET, soil moisture, TWS and streamflow) should also be validated as much as possible using the observations, remote sensing data and reanalysis, to ensure the models' simulations reasonable. Only then will we believe the further analysis between future and historical periods is valid. In my opinion, the historical simulation of VR-CESM is not so good because the simulated dry, median, and wet water years are distinct from the PRISM (Figure A2).

The developed hydrologic model was previously compared to measurements: simulated ET was compared to remotely sensed ET derived from METRIC, soil moisture was compared to SMAP, snow water equivalent to SNODAS and a reanalysis by Bair et al., streamflow and groundwater levels variations were compared to ground measurements (4 stations were used to compare streamflow and 3 wells for groundwater levels comparisons). Comparisons with GRACE TWS are not meaningful given the size of this watershed (~7000 km$^2$) which is far smaller than the footprint of GRACE TWS (200,000 km$^2$). An appendix containing these comparisons will be added to the revised manuscript.
Because the hydrologic model was only run for a certain period of time and specific years, the comparisons were only performed for these years on the contrary to the climate model which has been compared throughout the entire historical period.

**Validation of the hydrologic model**
*We compared temporal variations of streamflow at 3 stations located in the Sierra (uplands), the intersection between the Sierra and the Central Valley, and the outskirts of Sacramento (see Figure R1). Four wells in the watershed (see Figure R1a) have reasonable, publicly-available records of groundwater levels and were used to check the ability of the model to reproduce water table depth variations.*

[Figure]

Download : Download high-res image (403KB)
Download : Download full-size image

*Figure R1a: The Cosumnes watershed geology and the locations of the 3 streamflow gauges (CNF, MHB, and MFR) and 4 groundwater wells (stars).*

*Figure R1b depicts the comparisons between simulated and measured river stages at the 3 stations indicated in figure R1a. Absolute errors (L1) in m and relative errors (L2) are shown in Table R1a. Differences between simulated and measured streamflow vary between 0.4 and 0.8 m (Table R1a) indicating that the model is able to reproduce the river dynamics.*

[Figure]

*Figure R1b: Comparisons between measured and calculated river stages (i.e., pressure-heads simulated by ParFlow-CLM). Measurements locations are indicated in Figure R1a.*

**Measurements**

| | $L_1$(m) | $L_2$ |
|---|---|---|
| River stages (CNF) | 0.8 | 0.5 |
| River stages (MHB) | 0.4 | 0.36 |
| River stages (MFR) | 0.57 | 1.09 |
| Groundwater Levels (Well 1) | 3.73 | 0.0553 |
| Groundwater Levels (Well 2) | 1.63 | 0.025 |
| Groundwater Levels (Well 3) | 0.476 | 0.0077 |
| Groundwater Levels (Well 4) | 1.08 | 0.016 |

*Table R1a: Differences between measured and calculated surface and groundwater levels. L1 is the absolute error and R2 the relative error.*

*Comparisons between simulated and calculated groundwater levels (here referred to as the pressure-heads at the bottom of the domain) shown in figure R1c indicate that the model has reasonable agreements with measurements. As shown in table R1a, the error varies between 0.47 to 3.73 m depending on the station. Mismatches between simulated and observed groundwater levels at wells 1 and 2 are likely due to an inaccurate estimation of pumping in these areas. The temporal variations of the groundwater levels show an impact of withdrawals but because these withdrawals are hard to estimate the model isn't correctly reproducing these trends.*

[Figure]

*Figure R1c: Comparisons between measured and calculated pressure-heads at the bottom of the domain. Measurements locations are indicated in Fig. R1.*

*ParFlow-CLM also solves the key land surface processes governing the transfer of water and energy at the land-atmosphere-soil interface: evapotranspiration, snow dynamics, and soil*

*moisture. In Maina et al., (2020a), rigorous comparisons between the ParFlow-CLM simulated land surface processes and remotely sensed estimates of these variables was conducted. Table R1b shows the correlation coefficient between ParFlow-CLM results and the various datasets compared.*

[Figure]

*Figure R1d: (a) Comparisons between domain-averaged total snow water equivalent obtained with ParFlow-CLM, SNODAS and Bair et al., reconstruction, (b) Comparisons between actual evapotranspiration obtained with ParFlow-CLM and METRIC (c) Relative variation of soil moisture obtained with ParFlow-CLM and SMAP. Note that the x-axis of (c) is shorter because of the availability of SMAP data*

Satellites-based products

|  | $L_1$ | $L_2$ | Pearson correlation coefficient |
|---|---|---|---|
| SNODAS (mm) | 3.09 | 3.77 | 0.97 |
| Bair et al., 2016 (mm) | 3.80 | 2.69 | 0.84 |
| SMAP (-) | 0.217 | 3.07 | 0.94 |
| METRIC (mm/s) | 0.0367 | 1.40 | 0.6 |

*Table R1b: differences between measured and remotely sensed evapotranspiration (METRIC), soil moisture (SMAP), and snow water equivalent (SNODAS and Bair et al., 2016)*

VR-CESM is simulated under AMIP-protocols (bounded by monthly observed sea-surface temperatures and sea-ice extents), and therefore we do not expect VR-CESM to exactly recreate past historical water years. However, we do expect that our 30-year simulation can reasonably recreate the range of water year types over California and the Cosumnes, which is why we utilize the broader range of PRISM water years that are available. While the water years are different, the magnitudes of the precipitation are similar.

The authors may argue the historical simulations are acceptable, because a global climate and integrated hydrologic models are used (more complex and larger simulation domain). However, one can use a finer-resolution hydrological model (e.g., VIC, SWAT, and many others) driven by statistically or dynamically downscaled regional climate model outputs to obtain more reasonable (maybe more accurate from the perspective of validation) simulations in this river basin (7000 km2), and to do further analysis like the authors did in this study. Please explain why the global climate and integrated hydrologic models are more suitable for this case study?

We set up our modeling framework by taking into account the:
- Californian atmospheric dynamics.
- Impacts of groundwater dynamics and lateral flow on the hydrology and the land surface processes of the region.
- Dependance of the groundwater dynamics in the valley to the snow dynamics in the Sierra Nevada

These considerations are critical for a better understanding of the impacts of a changing climate on Californian hydrology.

ParFlow-CLM is an integrated hydrologic model that solves the transfer of water and energy from the bedrock to the canopy. Parflow uses the Richards equation a physics-based equation that solves the subsurface flow in three dimensions and therefore accounts for deeper and lateral flow. Previous studies have demonstrated that the lateral flow is very important to the surface and land energy dynamics (Maxwell and Condon, 2016). On the contrary to VIC, ParFlow accounts for the lateral flow. It also employs a series of physics-based equations contrary to SWAT. In addition to the subsurface flow, ParFlow also solves the overland (i.e., surface) flow by using the kinematic wave equation contrary to VIC.

When simulating the evolution of climate in California, the interaction between dynamical and thermodynamical responses has important, and sometimes, offsetting effects on features such as atmospheric rivers. Payne et al. (2020) show that the thermodynamic response to climate change enhances atmospheric river characteristics (e.g., Clausius-Clapeyron relationship), whereas the dynamical response diminishes atmospheric river characteristics (e.g., changes in the jet stream and storm track landfall location). Therefore, it is important to employ a modeling framework that accounts for the dynamical and thermodynamical effects of climate change such as VR-CESM.

We also did not perform statistical downscaling as this is one of the "selling points" of leveraging variable-resolution Earth system model capabilities, namely that it enables dynamical downscaling internally within an Earth system model which has the benefits of limiting multiple model bias propagation, allows for more consistent teleconnection responses, enables upscale/downscale effects to influence the broader climate, etc. As a result, we think this study adds a "unique" data point to the literature regarding changes in end-century hydrology in California given that it is a slightly different methodology compared with traditional regional climate model based dynamical downscaling efforts and/or bias-corrected statistically downscaled global climate model efforts.

To capture both the particularities of Californian climate and its interactions with the hydrology from bedrock to the canopy, the approach we used is more adequate than the aforementioned approaches.

Below is a table with the most used hydrologic models and their advantages and limitations when simulating the hydrology of California. Only Hydrogeosphere and ATS have similar advantages as ParFlow-CLM and are suitable to model the Californian hydrology. Because the equations and

the coupling approaches used by these models are similar, we expect their results to be the same. Moreover, these models are also computationally expensive hence they also have to limitation of resolution.

| Hydrologic Model | Land Surface | Surface | Subsurface | Limitations when simulating Californian hydrology |
|---|---|---|---|---|
| MODFLOW (Harbaugh, 2005)/FELFOW (Trefry and Muffels, 2007) | No | No | Yes (diffusivity equation) | These models do not integrate land surface processes (such as snow dynamics) and their interactions with the subsurface critical to the Californian hydrology. |
| SWAT (Soil and Water Assessment Tool) (Neitsch et al., 2000) | Yes | Yes | Yes | The model is based on HRU (hydrologic response units). The model isn't physics-based, therefore, it doesn't account for the two-way interaction between the land surface and the subsurface processes. |
| SAC-MA (Sacramento Soil Moisture Accounting Model) | No | Yes (Rainfall-Runoff) | Yes (Water Budget) | The model doesn't simulate snow dynamics and evapotranspiration. A water budget equation is used to simulate the groundwater dynamics which doesn't account for the lateral flow and unsaturated zone flow. |
| Noah-MP (Niu et al., 2011) | Yes (water and energy balance) | Yes (a routing scheme can be used to derive surface flow) | Yes (percolation) | Although this model physically solves the land surface processes including evapotranspiration and snow dynamics, it doesn't account for the two-way interaction between the land surface processes and the subsurface. Lateral and unsaturated zone flows are not represented. |
| VIC (Variable Infiltration Capacity Model Macroscale | Yes | Yes (Rainfall-Runoff) | Yes (percolation and water budget) | Although this model physically solves the land surface processes including evapotranspiration and snow dynamics, it doesn't account |

| | | | | |
|---|---|---|---|---|
| Hydrologic Model) (Liang et al., 1994) | | | | for the two-way interaction between the land surface processes and the subsurface. Lateral and unsaturated zone flows are not represented. |
| Hydrogeosphere (Aquanty, 2015) | Yes (water and energy balance) | Yes (2D diffusive wave equation) | Yes (3D Richards equation) | This model has similar advantages as ParFlow-CLM and could be used to model the hydrology of California. |
| CATHY (Catchment Hydrology) (Bixio et al., 2002) | Yes (there is a version coupled to Noah-MP) | Yes (1D Saint Venant Equation) | Yes (Mass balance equation) | The mass balance equation is not as robust as the Richards equation for describing the variably saturated flow in the subsurface and recharge processes. In addition, the original model doesn't solve land surface processes. |
| MIKE-SHE (Abbott et al., 1986) | No | Yes (diffusivity equation) | Yes (Darcy equation and a 1D Richards equation) | The main limitation of this model is the lack of land surface processes and the Darcy equation used to describe subsurface flow doesn't account for the unsaturated flow. |
| ATS (Advanced Terrestrial Simulator) (Coon et al., 2016) | Yes (water and energy balance) | Yes (2D diffusivity equation) | Yes (3D Richards equation) | This model has similar advantages as ParFlow-CLM and could be used to model the hydrology of California. |
| ParFlow-CLM (Kollet and Maxwell, 2006) | Yes (water and energy balance) | Yes (2D diffusivity equation) | Yes (3D Richards equation) | |

Table R1c: Advantages and limitations of the most used hydrological models

*Additional references*
*Maina, Fadji Z., Siirila-Woodburn, E. R., Newcomer, M., Xu, Z., & Steefel, C. (2020a).*
*Determining the impact of a severe dry to wet transition on watershed hydrodynamics in*
*California, USA with an integrated hydrologic model. Journal of Hydrology, 580, 124358.*
*https://doi.org/10.1016/j.jhydrol.2019.124358*

*Maina, F. Z., Siirila-Woodburn, E. R., & Vahmani, P. (2020b). Sensitivity of meteorological-forcing resolution on hydrologic variables. Hydrology and Earth System Sciences, 24(7), 3451–3474. https://doi.org/10.5194/hess-24-3451-2020*

*Maina, Fadji Zaouna, & Siirila-Woodburn, E. R. (2020c). Watersheds dynamics following wildfires: Nonlinear feedbacks and implications on hydrologic responses. Hydrological Processes, 34(1), 33–50. https://doi.org/10.1002/hyp.13568*

*Harbaugh AW (2005) MODFLOW-2005, The U.S. Geological Survey modular ground-water model: the ground-water flow process. US Geol Surv Tech Methods 6-A16. http://pubs.usgs.gov/tm/2005/tm6A16/.*

*Trefry, M.G.; Muffels, C. (2007). "FEFLOW: a finite-element ground water flow and transport modeling tool". Ground Water. 45 (5): 525–528. doi:10.1111/j.1745-6584.2007.00358.x.*

*Neitsch, S. L., Arnold, J. G., Kiniry, J. R., & Williams, J. R. (2001). Soil and Water Assessment tool (SWAT) user's manual version 2000. Grassland Soil and Water Research Laboratory. Temple, TX: ARS.*

*Niu, G.-Y., et al. (2011), The community Noah land surface model with multiparameterization options (Noah-MP): 1. Model description and evaluation with local-scale measurements. J. Geophys. Res., 116, D12109, doi: 10.1029/2010JD015139.*

*Liang, X., D. P. Lettenmaier, E. F. Wood, and S. J. Burges (1994), A simple hydrologically based model of land surface water and energy fluxes for general circulation models, J. Geophys. Res., 99(D7), 14415–14428, doi:10.1029/94JD00483.*

*Aquanty, I. (2015), HydroGeoSphere User Manual, 435 pp., Waterloo, Ont.*

*Bixio, A. C., G. Gambolati, C. Paniconi, M. Putti, V. M. Shestopalov, V. N. Bublias, A. S. Bohuslavsky, N. B. Kasteltseva, and Y. F. Rudenko (2002), Modeling groundwater-surface water interactions including effects of morphogenetic depressions in the Chernobyl exclusion zone, Environ. Geol., 42(2-3) 162-177*

*Abbott, M. B., J. C. Bathurst, J. A. Cunge, P. E. Oconnell, and J. Rasmussen (1986), An introduction to the european hydrological system: Sys- teme hydrologique Europeen, She .2. Structure of a physically-based, distributed modeling system, J. Hydrol., 87(1–2), 61–77.*

*Coon, E. T., J. D. Moulton, and S. L. Painter (2016), Managing complexity in simulations of land surface and near-surface processes, Environ. Modell Software, 78, 134-149*

*Kollet, S. J., & Maxwell, R. M. (2006). Integrated surface–groundwater flow modeling: A free surface overland flow boundary condition in a parallel groundwater flow model. Advances in Water Resources, 29(7), 945–958. https://doi.org/10.1016/j.advwatres.2005.08.006*

*Maxwell, R. M., & Condon, L. E. (2016). Connections between groundwater flow and transpiration partitioning. Science, 353(6297), 377–380. https://doi.org/10.1126/science.aaf7891*

---

## Author Comment (AC2)

In this work, the authors present global climatic and hydrologic models to simulate the extremes and their impacts on the water-energy balance over California. The paper is well written and with high relevance to the hess journal. Please see some suggestions I kindly ask the authors to address:

We thank the reviewer for their positive comments and feedback and for acknowledging the quality and the significance of our work.

1) The title of the paper is "Projecting the impacts of end of century climate extremes on the hydrology in California.". The title of the paper is a bit strong since it recommends that the whole hydrological-cycle has been modeled for the State of California and also for a time-window reaching the end of the century. Many authors struggle to simulate only one part of the hydrological-cycle of California (e.g., rainfall-runoff model, as for example in Yin et al., 2021; while many similar studies exist in literature). For such a promising title, a strong literature review should be performed to include similar studies for all hydrological-cycle variables and to show how the proposed model is more advanced.

We acknowledge that the title could be misleading since we are only simulating a watershed in California although the watershed is representative of the state's hydrology.

We propose to change the title to "Projecting end of century climate extremes and their impacts on the hydrology of a representative California watershed"

While we didn't simulate the hydrology through the end of the century, we selected particular years of interest by analyzing the end of century hydroclimate (from 2070 to 2100).

In the revised manuscript we will add the listed references although these studies are different from ours as they simulated the hydrology using rainfall-runoff and machine learning models, therefore, targeting single/individual components of the water cycle and/or not accounting for the physical characteristics of the area. To better understand how the hydrology will evolve in response to climate change it is important to represent the transfer of water and energy from the bedrock to the canopy especially in California where the subsurface hydrology downstream (i.e., groundwater dynamics) strongly depends on the land surface processes occurring upstream (i.e., snowmelt). To capture such behaviors, ParFlow-CLM is adequate.

Below is a table with the most used hydrologic models and their advantages and limitations when simulating the hydrology of California, highlighting the strong advantages of ParFlow-CLM. Only Hydrogeosphere and ATS have similar advantages as ParFlow-CLM and are suitable to model the Californian hydrology. Because the equations and the coupling approaches used by these models are similar, we expect their results to be the same.

| Hydrologic Model                      | Land
Surface | Surface | Subsurface                       | Limitations when
simulating Californian
hydrology                  |
|---------------------------------------|-----------------|---------|----------------------------------|--------------------------------------------------------------------------|
| MODFLOW
(Harbaugh,
2005)/FELFOW | No              | No      | Yes
(diffusivity
equation) | These models do not
integrate land surface
processes (such as snow |
|                                       |                 |         | 1 )                              | dynamics) and their                                                      |

| (Trefry and          |          |            |               | interactions with the        |
|----------------------|----------|------------|---------------|------------------------------|
| Muffels, 2007)       |          |            |               | subsurface critical to the   |
|                      |          |            |               | Californian hydrology.       |
| SWAT (Soil and       | Yes      | Yes        | Yes           | The model is based on HRU    |
| Water Assessment     |          |            |               | (hydrologic response units). |
| Tool) (Neitsch et    |          |            |               | The model isn't physics-     |
| al., 2000)           |          |            |               | based, therefore, it doesn't |
|                      |          |            |               | account for the two-way      |
|                      |          |            |               | interaction between the land |
|                      |          |            |               | surface and the subsurface   |
|                      |          |            |               | processes.                   |
| SAC-MA               | No       | Yes        | Yes (Water    | The model doesn't simulate   |
| (Sacramento Soil     |          | (Rainfall- | Budget)       | snow dynamics and            |
| Moisture             |          | Runoff)    | U ,           | evapotranspiration. A water  |
| Accounting           |          |            |               | budget equation is used to   |
| Model)               |          |            |               | simulate the groundwater     |
|                      |          |            |               | dynamics which doesn't       |
|                      |          |            |               | account for the lateral flow |
|                      |          |            |               | and unsaturated zone flow.   |
| Noah-MP (Niu et      | Yes      | Yes (a     | Yes           | Although this model          |
| al., 2011)           | (water   | routing    | (percolation) | physically solves the land   |
|                      | and      | scheme can | , a ,         | surface processes including  |
|                      | energy   | be used to |               | evapotranspiration and snow  |
|                      | balance) | derive     |               | dynamics, it doesn't account |
|                      |          | surface    |               | for the two-way interaction  |
|                      |          | flow)      |               | between the land surface     |
|                      |          |            |               | processes and the            |
|                      |          |            |               | subsurface. Lateral and      |
|                      |          |            |               | unsaturated zone flows are   |
|                      |          |            |               | not represented.             |
| VIC (Variable        | Yes      | Yes        | Yes           | Although this model          |
| Infiltration         |          | (Rainfall- | (percolation  | physically solves the land   |
| Capacity Model       |          | Runoff)    | and water     | surface processes including  |
| Macroscale           |          |            | budget)       | evapotranspiration and snow  |
| Hydrologic Model)    |          |            |               | dynamics, it doesn't account |
| (Liang et al., 1994) |          |            |               | for the two-way interaction  |
|                      |          |            |               | between the land surface     |
|                      |          |            |               | processes and the            |
|                      |          |            |               | subsurface. Lateral and      |
|                      |          |            |               | unsaturated zone flows are   |
|                      |          |            |               | not represented.             |
| Hydrogeosphere       | Yes      | Yes (2D    | Yes (3D       | This model has similar       |
| (Aquanty, 2015)      | (water   | diffusive  | Richards      | advantages as ParFlow-       |
|                      | and      | wave       | equation)     | CLM and could be used to     |
|                      | energy   | equation)  |               | model the hydrology of       |
|                      | balance) |            |               | California.                  |

| CATHY             | Yes         | Yes (1D      | Yes (Mass     | The mass balance equation    |
|-------------------|-------------|--------------|---------------|------------------------------|
| (Catchment        | (there is a | Saint Venant | balance       | is not as robust as the      |
| Hydrology) (Bixio | version     | Equation)    | equation)     | Richards equation for        |
| et al., 2002)     | coupled     |              |               | describing the variably      |
| í.                | to Noah-    |              |               | saturated flow in the        |
|                   | MP)         |              |               | subsurface and recharge      |
|                   |             |              |               | processes. In addition, the  |
|                   |             |              |               | original model doesn't solve |
|                   |             |              |               | land surface processes.      |
| MIKE-SHE          | No          | Yes          | Yes (Darcy    | The main limitation of this  |
| (Abbott et al.,   |             | (diffusivity | equation and  | model is the lack of land    |
| 1986)             |             | equation)    | a 1D Richards | surface processes and the    |
|                   |             |              | equation)     | Darcy equation used to       |
|                   |             |              |               | describe subsurface flow     |
|                   |             |              |               | doesn't account for the      |
|                   |             |              |               | unsaturated flow.            |
| ATS (Advanced     | Yes         | Yes (2D      | Yes (3D       | This model has similar       |
| Terrestrial       | (water      | diffusivity  | Richards      | advantages as ParFlow-       |
| Simulator) (Coon  | and         | equation)    | equation)     | CLM and could be used to     |
| et al., 2016)     | energy      |              |               | model the hydrology of       |
| í.                | balance)    |              |               | California.                  |
| ParFlow-CLM       | Yes         | Yes (2D      | Yes (3D       |                              |
| (Kollet and       | (water      | diffusivity  | Richards      |                              |
| Maxwell, 2006)    | and         | equation)    | equation)     |                              |
|                   | energy      |              |               |                              |
|                   | balance)    |              |               |                              |

Table R1c: Advantages and limitations of the most used hydrological models

2) There is a lack of calibration, validation, and verification of the proposed model. When a forecast is performed, one should use a part of the timeseries to calibratevalidateverify their model, and then perform a forecast for the near future. I suggest the authors see/discuss this procedure concerning their own model.

We didn't employ a time-series based comparison for the climate model due to the uncertainties of the model to capture individual events throughout the year. VR-CESM is simulated under AMIP-protocols (bounded by monthly observed sea-surface temperatures and sea-ice extents), and therefore we do not expect VR-CESM to exactly recreate past historical WYs. However, we do expect that our 30-year simulation can reasonably recreate the range of WY types over California and the Cosumnes, which is why we utilize the broader range of PRISM WYs that are available. The VR-CESM simulations are not forecasts or predictions, but rather projections. There is a subtle but important difference in a prediction, which aims to exactly recreate an event or time period, versus a projection, which aims to encapsulate the envelope of plausible future scenarios given socioeconomic development/greenhouse gas emissions, etc. The end-century projections performed with VR-CESM allow the atmosphere and land-surface model to interact under assumptions of the "upper end" emissions scenario (RCP8.5), land-surface cover changes, and

increases in sea-surface temperatures and decreases in sea-ice. Therefore, the 30-year period (2070-2100) encapsulated by these VR-CESM projections should be thought of as "what might happen to the middle and end member years (i.e., driest and wettest) if the world warms by  $+4 - 5\circ$ C?"

We calibrated and validated the hydrologic model using remotely sensed and ground measurements of streamflow, groundwater levels, snow water equivalent, soil moisture, and evapotranspiration.

Below are the details of the comparisons which have been published in a previous paper and will be added to the appendix of the revised manuscript.

**Model validation procedure (also added to the response to reviewer 1)**

We compared temporal variations of streamflow at 3 stations located in the Sierra (uplands), the intersection between the Sierra and the Central Valley, and the outskirts of Sacramento (see Figure R1). Four wells in the watershed (see Figure R1a) have reasonable, publicly-available records of groundwater levels and were used to check the ability of the model to reproduce water table depth variations.

Download : Download high-res image (403KB) Download : Download full-size image

*Figure R1a: The Cosumnes watershed geology and the locations of the 3 streamflow gauges (CNF, MHB, and MFR) and 4 groundwater wells (stars).*

Figure R1b depicts the comparisons between simulated and measured river stages at the 3 stations indicated in figure R1a. Absolute errors (L1) in m and relative errors (L2) are shown in Table

*R1a.* Differences between simulated and measured streamflow vary between 0.4 and 0.8 m (Table *R1a*) indicating that the model is able to reproduce the river dynamics.

*Figure R1b: Comparisons between measured and calculated river stages (i.e., pressure-heads simulated by ParFlow-CLM). Measurements locations are indicated in Figure R1a.*

| Measurements                |           |        |  |
|-----------------------------|-----------|--------|--|
|                             | $L_1$ (m) | $L_2$  |  |
| River stages (CNF)          | 0.8       | 0.5    |  |
| River stages (MHB)          | 0.4       | 0.36   |  |
| River stages (MFR)          | 0.57      | 1.09   |  |
| Groundwater Levels (Well 1) | 3.73      | 0.0553 |  |
| Groundwater Levels (Well 2) | 1.63      | 0.025  |  |
| Groundwater Levels (Well 3) | 0.476     | 0.0077 |  |
| Groundwater Levels (Well 4) | 1.08      | 0.016  |  |

Table R1a: Differences between measured and calculated surface and groundwater levels. L1 is the absolute error and R2 the relative error.

Comparisons between simulated and calculated groundwater levels (here referred to as the pressure-heads at the bottom of the domain) shown in figure R1c indicate that the model has reasonable agreements with measurements. As shown in table R1a, the error varies between 0.47 to 3.73 m depending on the station. Mismatches between simulated and observed groundwater levels at wells 1 and 2 are likely due to an inaccurate estimation of pumping in these areas. The temporal variations of the groundwater levels show an impact of withdrawals but because these withdrawals are hard to estimate the model isn't correctly reproducing these trends.

---

## Author Response (AR1)

Below is an itemized list of all comments in plain text and our responses in blue.

**Anonymous Referee #1**

The manuscript simulates End of Century (EOC) extremes and their effects on the water-energy balance in the Cosumnes river basin, using cutting-edge global climate and integrated hydrologic models (ParFlow-CLM). I really like the way the authors used to analyze the hydroclimatic changes by median WY, dry WY and wet WY (e.g., Figures 3-5). The manuscript is overall clearly written, and the results are well discussed.

We thank the reviewers for their positive comments and feedback and for acknowledging the quality and the significance of our work.

My first concern is the insufficient validation of the models' simulations in the historical period. Besides temperature and precipitation outputs, other watershed-integrated fluxes, and storages (e.g., ET, soil moisture, TWS and streamflow) should also be validated as much as possible using the observations, remote sensing data and reanalysis, to ensure the models' simulations reasonable. Only then will we believe the further analysis between future and historical periods is valid. In my opinion, the historical simulation of VR-CESM is not so good because the simulated dry, median, and wet water years are distinct from the PRISM (Figure A2).

The developed hydrologic model was previously compared to measurements in Maina et al. (2020): simulated ET was compared to remotely sensed ET derived from METRIC, soil moisture was compared to SMAP, snow water equivalent to SNODAS and a reanalysis by Bair et al., streamflow and groundwater levels variations were compared to ground measurements (4 stations were used to compare streamflow and 3 wells for groundwater levels comparisons). Comparisons with GRACE TWS are not meaningful given the size of this watershed (~7000 km$^2$) which is far smaller than the footprint of GRACE TWS (200,000 km$^2$). We have added details of the model validation below (Validation of the hydrologic model) and to appendix C of the revised manuscript.

We have also added the following lines to the revised manuscript, please refer to lines 341-365 (see below the text in italic).

*"We specifically compared simulated and measured river stages at three stations located in the Sierra Nevada headwater, foothill, and the Central Valley. The annual averages absolute differences between measurements and simulations were between 0.4 and 0.8 m. We selected four wells in the Cosumnes watershed based on their availability of data to compare measured and simulated groundwater levels. These wells are sparsely distributed in the Central Valley. The absolute differences observed and simulated groundwater levels vary between 0.47 to 3.73 m. The highest absolute differences were attributed to the lack of a best estimation of groundwater pumping rates in the region. Nonetheless, the reasonable agreement between observations and simulated variables has allowed us to conclude that the model can capture these extreme dynamics. We rely on remote sensing data to assess the ability of our model to simulate key land surface processes (evapotranspiration, soil moisture, and snow dynamics). We compared the simulated SWE to SNODAS (The National Weather Service's Snow Data Assimilation, National Operational Hydrologic Remote Sensing Center, 2004) and a SWE reanalysis by Bair et al., (2016). Our comparisons indicated that the absolute differences between our SWE values and these data were equal to 3 mm on average. Moreover, the simulated key parameters controlling*

*the snow dynamics such as peak snow and timing of snow ablation were also in agreement with remotely sensed data for both dry and wet years (Appendix C). Absolute differences between the simulated ET and the remotely sensed ET from METRIC (Mapping Evapotranspiration at High Resolution with Internalized Calibration, Allen et al., 2007) were equal to 0.036 mm/s while the differences between the simulated soil moisture and the SMAP (Soil Moisture Active Passive, SMAP, 2015) soil moisture were 0.2."*

This hydrologic validation was based on WY 2015-2017 in Maina et al. (2020), which used meteorological forcings specific to those years. In contrast, the climate model simulations were compared throughout the entire historical period because they represent 30 plausible realizations of the historical climate, since the simulations are only bounded by observed sea-surface temperatures and sea ice extents (a common practice in the climate modeling community known as AMIP protocols, https://pcmdi.llnl.gov/mips/amip/home/overview.html). Therefore, these simulations would not be expected to exactly recreate specific water years, due to internal variability in the atmosphere, but would be expected to recreate the distribution of water year types. We have clarified it in the revised manuscript, please refer to lines 190-201 (see below the text in italic).

*"The atmospheric model used for these simulations is the Community Atmosphere Model (CAM) version 5.4 with the spectral element dynamical core, with an atmospheric dynamics time step of 75 seconds, an atmospheric physics time step of 450 seconds, a prognostic treatment of rainfall and snowfall in the microphysics scheme (Gettelman and Morrison, 2015) and run under Atmosphere Model Intercomparison Project (AMIP) protocols (Gates, 1992). Under the AMIP protocols, the atmosphere and land-surface components of the Earth system model are coupled and periodically bounded by monthly observed sea-surface temperatures and sea-ice extents. Although this configuration does not exactly recreate historical water years and events, it is expected to reasonably simulate the distribution of water year types. Also, it should be noted that the model only projects future conditions, within the envelope of plausible future conditions of the RCP8.5 scenario and its assumptions of greenhouse gas emissions, sea-surface temperatures, and sea ice extents and would not be expected to exactly forecast individual water years."*

**Validation of the hydrologic model**
*We compared temporal variations of streamflow at 3 stations, one each located in the Sierra (uplands), at the intersection between the Sierra and the Central Valley, and in the outskirts of Sacramento (see Figure R1). Four wells in the watershed (see Figure R1a) have reasonable, publicly-available records of groundwater levels and were used to check the ability of the model to reproduce water table depth variations.*

[Figure]

*Figure R1a: The Cosumnes watershed geology and the locations of the 3 streamflow gauges (CNF, MHB, and MFR) and 4 groundwater wells (stars).*

*Figure R1b depicts the comparisons between simulated and measured river stages at the 3 stations indicated in figure R1a. Absolute errors (L1) in m and relative errors (L2) are shown in Table R1a. Differences between simulated and measured streamflow vary between 0.4 and 0.8 m (Table R1a) indicating that the model is able to reproduce the river dynamics.*

*(a)*

[Figure]

*Figure R1b: Comparisons between measured and calculated river stages (i.e., pressure-heads simulated by ParFlow-CLM). Measurements' locations are indicated in Figure R1a.*

| Measurements | $L_1$ (m) | $L_2$ (-) |
|---|---|---|
| River Stages (CNF) | 0.8 | 0.5 |
| River Stages (MHB) | 0.4 | 0.36 |
| River Stages (MFR) | 0.57 | 1.06 |
| Groundwater Levels (Well 1) | 3.73 | 0.05 |
| Groundwater Levels (Well 2) | 1.63 | 0.02 |
| Groundwater Levels (Well 3) | 0.476 | 0.0077 |
| Groundwater Levels (Well 4) | 1.08 | 0.016 |

*Table R1a: Differences between measured and calculated surface and groundwater levels. L1 is the absolute error and R2 the relative error.*

*Comparisons between simulated and calculated groundwater levels (here referred to as the pressure-heads at the bottom of the domain) shown in figure R1c indicate that the model has reasonable agreements with measurements. As shown in table R1a, the error varies between 0.47 to 3.73 m depending on the station. Mismatches between simulated and observed groundwater levels at wells 1 and 2 are likely due to an inaccurate estimation of pumping in these areas. The temporal variations of the groundwater levels show an impact of withdrawals but because these withdrawals are hard to estimate the model does not correctly reproduce these trends.*

[Figure]

*Figure R1c: Comparisons between measured and calculated pressure-heads at the bottom of the domain. Measurement locations are indicated in Fig. R1.*

*ParFlow-CLM also simulates the key land surface processes governing the transfer of water and energy at the land-atmosphere-soil interface: evapotranspiration, snow dynamics, and soil moisture. In Maina et al., (2020a), we conducted rigorous comparisons between the ParFlow-CLM simulated land surface processes and remotely sensed estimates of these variables. Table R1b shows the correlation coefficient between ParFlow-CLM results and the various datasets compared.*

[Figure]

*Figure R1d: (a) Comparisons between domain-averaged total snow water equivalent obtained with ParFlow-CLM, SNODAS and Bair et al., reconstruction, (b) Comparisons between actual evapotranspiration obtained with ParFlow-CLM and METRIC (c) Relative variation of soil moisture obtained with ParFlow-CLM and SMAP. Note that the time series of (c) is shorter because of the availability of SMAP data*

| Satellites based products | L1 (m) | L2 (-) | Pearson Correlation Coefficient |
|---|---|---|---|
| SWE SNODAS (mm) | 3.09 | 3.77 | 0.97 |
| SWE Bair et al., (mm) | 3.80 | 2.69 | 0.84 |
| Soil Moisture SMAP (-) | 0.217 | 3.07 | 0.94 |
| ET METRIC (mm/s) | 0.067 | 1.40 | 0.6 |

*Table R1b: differences between measured and remotely sensed evapotranspiration (METRIC), soil moisture (SMAP), and snow water equivalent (SNODAS and Bair et al., 2016)*

The authors may argue the historical simulations are acceptable, because a global climate and integrated hydrologic models are used (more complex and larger simulation domain). However, one can use a finer-resolution hydrological model (e.g., VIC, SWAT, and many others) driven by statistically or dynamically downscaled regional climate model outputs to obtain more reasonable (maybe more accurate from the perspective of validation) simulations in this river basin (7000 km2), and to do further analysis like the authors did in this study. Please explain why the global climate and integrated hydrologic models are more suitable for this case study?

We set up our modeling framework by taking into account the:
- Californian atmospheric dynamics.
- Impacts of groundwater dynamics and lateral flow on the hydrology and the land surface processes of the region.
- Dependance of the groundwater dynamics in the valley to the snow dynamics in the Sierra Nevada.

These considerations are critical for a better understanding of the impacts of a changing climate on Californian hydrology.

ParFlow-CLM is an integrated hydrologic model that solves the transfer of water and energy from the bedrock to the canopy. Parflow uses the Richards equation a physics-based equation that solves the subsurface flow in three dimensions and therefore accounts for deeper and lateral flow. Previous studies have demonstrated that the lateral flow is very important to the surface and land energy dynamics (Maxwell and Condon, 2016). On the contrary, VIC does not simulate this lateral

subsurface flow and only solves overland flow based on an additional routing model. It also employs a series of physics-based equations contrary to SWAT.

When simulating the evolution of California's climate, the interaction between dynamical and thermodynamic responses has important, and sometimes, offsetting effects on critical storms that drive annual precipitation and snowpack totals in the Sierra Nevada, such as atmospheric rivers. Payne et al. (2020) show that thermodynamic responses to climate change enhance atmospheric river characteristics (e.g., Clausius-Clapeyron relationship), whereas dynamical responses diminish atmospheric river characteristics (e.g., changes in the jet stream and storm track landfall location). Therefore, we argue that it is critical to account for both the dynamical and thermodynamical effects of climate change, which we do through the use of VR-CESM.

We did not perform statistical downscaling because leveraging variable-resolution Earth system model capabilities, such as VR-CESM, enables dynamical downscaling internally within an Earth system model which limits traditional multiple model bias propagation (e.g., bias from a global climate model forcing imposed on a regional climate model simulation that in turn would also generate biases), allows for more consistent teleconnection responses, enables upscale/downscale effects to influence the broader climate, etc. As a result, we think this study adds a "unique" data point to the literature regarding changes in end-century hydrology in California as it is a distinctly different methodology than previously explored regional climate model based dynamical downscaling efforts and/or bias-corrected statistically downscaled global climate model efforts.

Below is a table (Table R1c) with a number of commonly-used hydrologic models and their advantages and limitations when simulating the hydrology of California. Only Hydrogeosphere and ATS have similar advantages as ParFlow-CLM and represent best the Californian hydrology of interest to this study. Because these models use similar equations and the coupling approaches use, we expect their results to be the same. Moreover, these models all share the resolution limits imposed by high computational expense. We have added this information to the revised manuscript, please refer to lines 280-296 and the text below in italic.

*"ParFlow has many advantages in comparisons to other hydrologic models. Compared to other hydrologic models (MODFLOW (Harbaugh, 2005), FELFOW (Trefry and Muffels, 2007), SWAT (Soil and Water Assessment Tool) (Neitsch et al., 2000), SAC-MA (Sacramento Soil Moisture Accounting Model)), ParFlow has the advantages of accounting for land surface processes such as snow dynamics and evapotranspiration and their interactions with the subsurface which are crucial for studying the hydrology of California. ParFlow also solved the subsurface flow by accounting for variably saturated conditions, an important feature for calculating groundwater recharge and the connection between the groundwater and the land surface processes, which is not the case for the aforementioned models. While some hydrologic models have a better representation of the land surface processes (Noah-MP (Niu et al., 2011), VIC (Variable Infiltration Capacity Model Macroscale Hydrologic Model) (Liang et al., 1994)), these models do not have a detailed representation of the subsurface flows. Because the surface flow is important in the region and it establishes the connection between the headwaters and the valleys, its good representation is essential for projecting changes in hydrology. Compared to other integrated hydrologic models (CATHY (Catchment Hydrology) (Bixio et al., 2002), MIKE-SHE (Abbott et al.,*

*1986)), ParFlow has the advantages of solving a two-dimensional kinematic flow equation that is fully coupled to the Richards equation."*

| Hydrologic Model | Land Surface | Surface | Subsurface | Limitations when simulating Californian hydrology |
|---|---|---|---|---|
| MODFLOW (Harbaugh, 2005)/FELFOW (Trefry and Muffels, 2007) | No | No | Yes (diffusivity equation) | These models do not integrate land surface processes (such as snow dynamics) and their interactions with the subsurface critical to the Californian hydrology. |
| SWAT (Soil and Water Assessment Tool) (Neitsch et al., 2000) | Yes | Yes | Yes | The model is based on HRU (hydrologic response units). The model isn't physics-based, therefore, it doesn't account for the two-way interaction between the land surface and the subsurface processes. |
| SAC-MA (Sacramento Soil Moisture Accounting Model) | No | Yes (Rainfall-Runoff) | Yes (Water Budget) | The model doesn't simulate snow dynamics and evapotranspiration. A water budget equation is used to simulate the groundwater dynamics which doesn't account for the lateral flow and unsaturated zone flow. |
| Noah-MP (Niu et al., 2011) | Yes (water and energy balance) | Yes (a routing scheme can be used to derive surface flow) | Yes (percolation) | Although this model physically solves the land surface processes including evapotranspiration and snow dynamics, it doesn't account for the two-way interaction between the land surface processes and the subsurface. Lateral and unsaturated zone flows are not represented. |

| | | | | |
|---|---|---|---|---|
| VIC (Variable Infiltration Capacity Model Macroscale Hydrologic Model) (Liang et al., 1994) | Yes | Yes (Rainfall-Runoff) | Yes (percolation and water budget) | Although this model physically solves the land surface processes including evapotranspiration and snow dynamics, it doesn't account for the two-way interaction between the land surface processes and the subsurface. Lateral and unsaturated zone flows are not represented. |
| Hydrogeosphere (Aquanty, 2015) | Yes (water and energy balance) | Yes (2D diffusive wave equation) | Yes (3D Richards equation) | This model has similar advantages as ParFlow-CLM and could be used to model the hydrology of California. |
| CATHY (Catchment Hydrology) (Bixio et al., 2002) | Yes (there is a version coupled to Noah-MP) | Yes (1D Saint Venant Equation) | Yes (Mass balance equation) | The mass balance equation is not as robust as the Richards equation for describing the variably saturated flow in the subsurface and recharge processes. In addition, the original model doesn't solve land surface processes. |
| MIKE-SHE (Abbott et al., 1986) | No | Yes (diffusivity equation) | Yes (Darcy equation and a 1D Richards equation) | The main limitation of this model is the lack of land surface processes and the Darcy equation used to describe subsurface flow doesn't account for the unsaturated flow. |
| ATS (Advanced Terrestrial Simulator) (Coon et al., 2016) | Yes (water and energy balance) | Yes (2D diffusivity equation) | Yes (3D Richards equation) | This model has similar advantages as ParFlow-CLM and could be used to model the hydrology of California. |
| ParFlow-CLM (Kollet and Maxwell, 2006) | Yes (water and energy balance) | Yes (2D diffusivity equation) | Yes (3D Richards equation) | |

Table R1c: Advantages and limitations of the most used hydrological models

*Additional references*

*Maina, Fadji Z., Siirila-Woodburn, E. R., Newcomer, M., Xu, Z., & Steefel, C. (2020a). Determining the impact of a severe dry to wet transition on watershed hydrodynamics in California, USA with an integrated hydrologic model. Journal of Hydrology, 580, 124358. https://doi.org/10.1016/j.jhydrol.2019.124358*

*Maina, F. Z., Siirila-Woodburn, E. R., & Vahmani, P. (2020b). Sensitivity of meteorological-forcing resolution on hydrologic variables. Hydrology and Earth System Sciences, 24(7), 3451–3474. https://doi.org/10.5194/hess-24-3451-2020*

*Maina, Fadji Zaouna, & Siirila-Woodburn, E. R. (2020c). Watersheds dynamics following wildfires: Nonlinear feedbacks and implications on hydrologic responses. Hydrological Processes, 34(1), 33–50. https://doi.org/10.1002/hyp.13568*

*Harbaugh AW (2005) MODFLOW-2005, The U.S. Geological Survey modular ground-water model: the ground-water flow process. US Geol Surv Tech Methods 6-A16. http://pubs.usgs.gov/tm/2005/tm6A16/.*

*Trefry, M.G.; Muffels, C. (2007). "FEFLOW: a finite-element ground water flow and transport modeling tool". Ground Water. 45 (5): 525–528. doi:10.1111/j.1745-6584.2007.00358.x.*

*Neitsch, S. L., Arnold, J. G., Kiniry, J. R., & Williams, J. R. (2001). Soil and Water Assessment tool (SWAT) user's manual version 2000. Grassland Soil and Water Research Laboratory. Temple, TX: ARS.*

*Niu, G.-Y., et al. (2011), The community Noah land surface model with multiparameterization options (Noah-MP): 1. Model description and evaluation with local-scale measurements. J. Geophys. Res., 116, D12109, doi: 10.1029/2010JD015139.*

*Liang, X., D. P. Lettenmaier, E. F. Wood, and S. J. Burges (1994), A simple hydrologically based model of land surface water and energy fluxes for general circulation models, J. Geophys. Res., 99(D7), 14415–14428, doi:10.1029/94JD00483.*

*Aquanty, I. (2015), HydroGeoSphere User Manual, 435 pp., Waterloo, Ont.*

*Bixio, A. C., G. Gambolati, C. Paniconi, M. Putti, V. M. Shestopalov, V. N. Bublias, A. S. Bohuslavsky, N. B. Kasteltseva, and Y. F. Rudenko (2002), Modeling groundwater-surface water interactions including effects of morphogenetic depressions in the Chernobyl exclusion zone, Environ. Geol., 42(2-3) 162-177*

*Abbott, M. B., J. C. Bathurst, J. A. Cunge, P. E. Oconnell, and J. Rasmussen (1986), An introduction to the european hydrological system: Sys- teme hydrologique Europeen, She .2. Structure of a physically-based, distributed modeling system, J. Hydrol., 87(1–2), 61–77.*

*Coon, E. T., J. D. Moulton, and S. L. Painter (2016), Managing complexity in simulations of land surface and near-surface processes, Environ. Modell Software, 78, 134-149*

*Kollet, S. J., & Maxwell, R. M. (2006). Integrated surface–groundwater flow modeling: A free surface overland flow boundary condition in a parallel groundwater flow model. Advances in Water Resources, 29(7), 945–958. https://doi.org/10.1016/j.advwatres.2005.08.006*

*Maxwell, R. M., & Condon, L. E. (2016). Connections between groundwater flow and transpiration partitioning. Science, 353(6297), 377–380. https://doi.org/10.1126/science.aaf7891*

**Anonymous Referee #2**

In this work, the authors present global climatic and hydrologic models to simulate the extremes and their impacts on the water-energy balance over California. The paper is well written and with high relevance to the hess journal. Please see some suggestions I kindly ask the authors to address:

We thank the reviewer for their positive comments and feedback and for acknowledging the quality and the significance of our work.

1) The title of the paper is "Projecting the impacts of end of century climate extremes on the hydrology in California.". The title of the paper is a bit strong since it recommends that the whole hydrological-cycle has been modeled for the State of California and also for a time-window reaching the end of the century. Many authors struggle to simulate only one part of the hydrological-cycle of California (e.g., rainfall-runoff model, as for example in Yin et al., 2021; while many similar studies exist in literature). For such a promising title, a strong literature review should be performed to include similar studies for all hydrological-cycle variables and to show how the proposed model is more advanced.

We acknowledge that the title could be misleading since we are only simulating a watershed in California although the watershed is representative of the state's hydrology. We propose to change the title to "Projecting end of century climate extremes and their impacts on the hydrology of a representative California watershed".

While we didn't simulate the hydrology over the entire 30-years at the end of the century (2070-2100), we selected three years that represent the spread of hydroclimatic conditions in this end-of-century period by choosing the driest, median, and wettest years from the climate simulations. We believe that the study mentioned by the reviewer has a different scope from ours as it seeks to forecast discharge a week ahead of time for use in rainfall-runoff and machine learning models. To better understand how the hydrology will evolve over long timescales in response to climate change it is important to represent the transfer of water and energy from the bedrock to the canopy. This is especially important in California where the subsurface hydrology downstream (i.e., groundwater dynamics) strongly depends on the land surface processes occurring upstream (i.e., snowmelt). ParFlow-CLM has been shown to capture these critical processes in many sites.

To justify how our hydrologic model differs from others, we provide a table below (Table R1c) with the most used hydrologic models and their advantages and limitations when simulating the hydrology of California. Only Hydrogeosphere and ATS have similar advantages as ParFlow-CLM. We also argue that Parflow-CLM can best represent the Californian hydrology of interest to this study due to these unique advantages over other models. Because the equations and the coupling approaches used by these models are similar, we expect their results to be the same.

We have added this justification to the revised manuscript, please refer to lines 280-296 and the text in italic below.

*"ParFlow has many advantages in comparisons to other hydrologic models. Compared to other hydrologic models (MODFLOW (Harbaugh, 2005), FELFOW (Trefry and Muffels, 2007), SWAT (Soil and Water Assessment Tool) (Neitsch et al., 2000), SAC-MA (Sacramento Soil Moisture Accounting Model)), ParFlow has the advantages of accounting for land surface processes such as snow dynamics and evapotranspiration and their interactions with the subsurface which are crucial for studying the hydrology of California. ParFlow also solved the subsurface flow by accounting for variably saturated conditions, an important feature for calculating groundwater recharge and the connection between the groundwater and the land surface processes, which is not the case for the aforementioned models. While some hydrologic models have a better representation of the land surface processes (Noah-MP (Niu et al., 2011), VIC (Variable Infiltration Capacity Model Macroscale Hydrologic Model) (Liang et al., 1994)), these models do not have a detailed representation of the subsurface flows. Because the surface flow is important in the region and it establishes the connection between the headwaters and the valleys, its good representation is essential for projecting changes in hydrology. Compared to other integrated hydrologic models (CATHY (Catchment Hydrology) (Bixio et al., 2002), MIKE-SHE (Abbott et al., 1986)), ParFlow has the advantages of solving a two-dimensional kinematic flow equation that is fully coupled to the Richards equation."*

| Hydrologic Model | Land Surface | Surface | Subsurface | Limitations when simulating Californian hydrology |
|---|---|---|---|---|
| MODFLOW (Harbaugh, 2005)/FELFOW (Trefry and Muffels, 2007) | No | No | Yes (diffusivity equation) | These models do not integrate land surface processes (such as snow dynamics) and their interactions with the subsurface critical to the Californian hydrology. |
| SWAT (Soil and Water Assessment Tool) (Neitsch et al., 2000) | Yes | Yes | Yes | The model is based on HRU (hydrologic response units). The model isn't physics-based, therefore, it doesn't account for the two-way interaction between the land surface and the subsurface processes. |
| SAC-MA (Sacramento Soil Moisture Accounting Model) | No | Yes (Rainfall-Runoff) | Yes (Water Budget) | The model doesn't simulate snow dynamics and evapotranspiration. A water budget equation is used to simulate the groundwater dynamics which doesn't account |

| | | | | |
|---|---|---|---|---|
| | | | | for the lateral flow and unsaturated zone flow. |
| Noah-MP (Niu et al., 2011) | Yes (water and energy balance) | Yes (a routing scheme can be used to derive surface flow) | Yes (percolation) | Although this model physically solves the land surface processes including evapotranspiration and snow dynamics, it doesn't account for the two-way interaction between the land surface processes and the subsurface. Lateral and unsaturated zone flows are not represented. |
| VIC (Variable Infiltration Capacity Model Macroscale Hydrologic Model) (Liang et al., 1994) | Yes | Yes (Rainfall-Runoff) | Yes (percolation and water budget) | Although this model physically solves the land surface processes including evapotranspiration and snow dynamics, it doesn't account for the two-way interaction between the land surface processes and the subsurface. Lateral and unsaturated zone flows are not represented. |
| Hydrogeosphere (Aquanty, 2015) | Yes (water and energy balance) | Yes (2D diffusive wave equation) | Yes (3D Richards equation) | This model has similar advantages as ParFlow-CLM and could be used to model the hydrology of California. |
| CATHY (Catchment Hydrology) (Bixio et al., 2002) | Yes (there is a version coupled to Noah-MP) | Yes (1D Saint Venant Equation) | Yes (Mass balance equation) | The mass balance equation is not as robust as the Richards equation for describing the variably saturated flow in the subsurface and recharge processes. In addition, the original model doesn't solve land surface processes. |
| MIKE-SHE (Abbott et al., 1986) | No | Yes (diffusivity equation) | Yes (Darcy equation and a | The main limitation of this model is the lack of land surface processes |

| | | | 1D Richards equation) | and the Darcy equation used to describe subsurface flow doesn't account for the unsaturated flow. |
|---|---|---|---|---|
| ATS (Advanced Terrestrial Simulator) (Coon et al., 2016) | Yes (water and energy balance) | Yes (2D diffusivity equation) | Yes (3D Richards equation) | This model has similar advantages as ParFlow-CLM and could be used to model the hydrology of California. |
| ParFlow-CLM (Kollet and Maxwell, 2006) | Yes (water and energy balance) | Yes (2D diffusivity equation) | Yes (3D Richards equation) | |

Table R1c: Advantages and limitations of the most used hydrological models

2) There is a lack of calibration, validation, and verification of the proposed model.
When a forecast is performed, one should use a part of the timeseries to calibrate-validate-verify their model, and then perform a forecast for the near future. I suggest
the authors see/discuss this procedure concerning their own model.

We didn't employ a time-series based comparison for the climate model simulations because they are climate projections and not weather forecasts. VR-CESM is simulated under AMIP-protocols, meaning the atmosphere and land-surface components of the Earth system model are coupled and allowed to solve prognostic and diagnostic equations that describe the interactions between the atmosphere and land-surface while being prescribed new lower boundary conditions every month via observed sea-surface temperatures and sea-ice extents. Therefore, we do not expect VR-CESM to exactly recreate past historical water years and do not consider that these projections would exactly forecast the weather in a given future year. However, we do expect that our 30-year simulation can reasonably recreate the range of water year types over California and the Cosumnes, which is why we utilize the broader range of PRISM water years that are available to compare with our 30-year simulation.

To clarify, the VR-CESM simulations are not forecasts or predictions, but rather projections. There is a subtle but important difference in a prediction, which aims to exactly recreate an event or time period, versus a projection, which aims to encapsulate the envelope of plausible future scenarios given greenhouse gas emissions, sea-surface temperatures, sea ice extents, land-surface cover changes, etc. The end-century projections performed with VR-CESM allow the atmosphere and land-surface model to interact under assumptions of the high emissions scenario (RCP8.5), account for land-surface cover changes, and increases in sea-surface temperatures and decreases in sea-ice extent. Therefore, the 30-year period (2070-2100) encapsulated by these VR-CESM projections should be thought of as "what might happen to the middle and end member years (i.e., driest and wettest) if the world warms by +4 - 5∘C?".

We have clarified it in the revised manuscript, please refer to lines 190-201 (see below the text in italic).

*"The atmospheric model used for these simulations is the Community Atmosphere Model (CAM) version 5.4 with the spectral element dynamical core, with an atmospheric dynamics time step of 75 seconds, an atmospheric physics time step of 450 seconds, a prognostic treatment of rainfall and snowfall in the microphysics scheme (Gettelman and Morrison, 2015) and run under Atmosphere Model Intercomparison Project (AMIP) protocols (Gates, 1992). Under the AMIP protocols, the atmosphere and land-surface components of the Earth system model are coupled and periodically bounded by monthly observed sea-surface temperatures and sea-ice extents. Although this configuration does not exactly recreate historical water years and events, it is expected to reasonably simulate the distribution of water year types. Also, it should be noted that the model only projects future conditions, within the envelope of plausible future conditions of the RCP8.5 scenario and its assumptions of greenhouse gas emissions, sea-surface temperatures, and sea ice extents and would not be expected to exactly forecast individual water years."*

We calibrated and validated the hydrologic model using remotely sensed and ground measurements of streamflow, groundwater levels, snow water equivalent, soil moisture, and evapotranspiration. Below are the details of the comparisons which were published in Maina et al. (2020) and added to appendix C of the revised manuscript.
We have also added the following lines to the revised manuscript, please refer to lines 341-365 (see below the text in italic).

*"We specifically compared simulated and measured river stages at three stations located in the Sierra Nevada headwater, foothill, and the Central Valley. The annual averages absolute differences between measurements and simulations were between 0.4 and 0.8 m. We selected four wells in the Cosumnes watershed based on their availability of data to compare measured and simulated groundwater levels. These wells are sparsely distributed in the Central Valley. The absolute differences observed and simulated groundwater levels vary between 0.47 to 3.73 m. The highest absolute differences were attributed to the lack of a best estimation of groundwater pumping rates in the region. Nonetheless, the reasonable agreement between observations and simulated variables has allowed us to conclude that the model can capture these extreme dynamics. We rely on remote sensing data to assess the ability of our model to simulate key land surface processes (evapotranspiration, soil moisture, and snow dynamics). We compared the simulated SWE to SNODAS (The National Weather Service's Snow Data Assimilation, National Operational Hydrologic Remote Sensing Center, 2004) and a SWE reanalysis by Bair et al., (2016). Our comparisons indicated that the absolute differences between our SWE values and these data were equal to 3 mm on average. Moreover, the simulated key parameters controlling the snow dynamics such as peak snow and timing of snow ablation were also in agreement with remotely sensed data for both dry and wet years (Appendix C). Absolute differences between the simulated ET and the remotely sensed ET from METRIC (Mapping Evapotranspiration at High Resolution with Internalized Calibration, Allen et al., 2007) were equal to 0.036 mm/s while the differences between the simulated soil moisture and the SMAP (Soil Moisture Active Passive, SMAP, 2015) soil moisture were 0.2."*

**Model validation procedure (also added to the response to reviewer 1)**

*We compared temporal variations of streamflow at 3 stations, one each located in the Sierra (uplands), at the intersection between the Sierra and the Central Valley, and in the outskirts of Sacramento (see Figure R1). Four wells in the watershed (see Figure R1a) have reasonable, publicly-available records of groundwater levels and were used to check the ability of the model to reproduce water table depth variations.*

[Figure]

*Figure R1a: The Cosumnes watershed geology and the locations of the 3 streamflow gauges (CNF, MHB, and MFR) and 4 groundwater wells (stars).*

*Figure R1b depicts the comparisons between simulated and measured river stages at the 3 stations indicated in figure R1a. Absolute errors (L1) in m and relative errors (L2) are shown in Table R1a. Differences between simulated and measured streamflow vary between 0.4 and 0.8 m (Table R1a) indicating that the model is able to reproduce the river dynamics.*

*(a)*

[Figure]

*Figure R1b: Comparisons between measured and calculated river stages (i.e., pressure-heads simulated by ParFlow-CLM). Measurements locations are indicated in Figure R1a.*

| Measurements | L1 (m) | L2 (-) |
|---|---|---|
| River Stages (CNF) | 0.8 | 0.5 |
| River Stages (MHB) | 0.4 | 0.36 |

| River Stages (MFR) | 0.57 | 1.06 |
|---|---|---|
| Groundwater Levels (Well 1) | 3.73 | 0.05 |
| Groundwater Levels (Well 2) | 1.63 | 0.02 |
| Groundwater Levels (Well 3) | 0.476 | 0.0077 |
| Groundwater Levels (Well 4) | 1.08 | 0.016 |

*Table R1a: Differences between measured and calculated surface and groundwater levels. L1 is the absolute error and R2 the relative error.*

*Comparisons between simulated and calculated groundwater levels (here referred to as the pressure-heads at the bottom of the domain) shown in figure R1c indicate that the model has reasonable agreements with measurements. As shown in table R1a, the error varies between 0.47 to 3.73 m depending on the station. Mismatches between simulated and observed groundwater levels at wells 1 and 2 are likely due to an inaccurate estimation of pumping in these areas. The temporal variations of the groundwater levels show an impact of withdrawals but because these withdrawals are hard to estimate the model does not correctly reproduce these trends.*

[Figure]

*Figure R1c: Comparisons between measured and calculated pressure-heads at the bottom of the domain. Measurement locations are indicated in Fig. R1.*

*ParFlow-CLM also simulates the key land surface processes governing the transfer of water and energy at the land-atmosphere-soil interface: evapotranspiration, snow dynamics, and soil moisture. In Maina et al., (2020a), we conducted rigorous comparisons between the ParFlow-CLM simulated land surface processes and remotely sensed estimates of these variables. Table R1b shows the correlation coefficient between ParFlow-CLM results and the various datasets compared.*

[Figure]

*Figure R1d: (a) Comparisons between domain-averaged total snow water equivalent obtained with ParFlow-CLM, SNODAS and Bair et al., reconstruction, (b) Comparisons between actual evapotranspiration obtained with ParFlow-CLM and METRIC (c) Relative variation of soil moisture obtained with ParFlow-CLM and SMAP. Note that the time series of (c) is shorter because of the availability of SMAP data*

| Satellites based products | L1 (m) | L2 (-) | Pearson Correlation Coefficient |
|---|---|---|---|
| SWE SNODAS (mm) | 3.09 | 3.77 | 0.97 |
| SWE Bair et al., (mm) | 3.80 | 2.69 | 0.84 |
| Soil Moisture SMAP (-) | 0.217 | 3.07 | 0.94 |
| ET METRIC (mm/s) | 0.067 | 1.40 | 0.6 |

*Table R1b: differences between measured and remotely sensed evapotranspiration (METRIC), soil moisture (SMAP), and snow water equivalent (SNODAS and Bair et al., 2016)*

Also, the End of Century (EoC) forecast for such a large area is very optimistic in my opinion. Since climate dynamics is highly complex, I imagine that a forecast of only a few steps ahead is possible. If one is studying, for example, runoff on an annual scale, then after a couple of years, the variability of the forecast would be very wide, thus, reducing the credibility of the result (e.g., see Han et al., 2021). Also, the credibility of the outcome should depend on the available length of records. Here, the authors perform a forecast of 80 years ahead, which is double the length of records the authors use to construct the climatic and hydrologic model. I suggest to test/discuss how the variability/probability of the forecasts change as we move away from the present/historic data.

The study mentioned by the reviewer (Han et al., 2021) uses a deep learning approach. The deep learning approach is different from the model we employed in this study, which solves physical equations both prognostically and diagnostically. Although physics-based models depend on the initial conditions, the impact of those initial conditions decreases with time (Maina et al, 2017). While the geology dictating the hydrodynamic parameters such as hydraulic conductivity, porosity, and specific storage could change with time, this change usually occurs on geological timescales (thousands or millions of years). As acknowledged in the manuscript, the land cover may change by the end of the century. However, this change is uncertain and difficult to predict hence we didn't incorporate it in this study–in this sense our results are a sensitivity analysis focused on the shifts in meteorological forcing rather than a fully-integrated assessment of changes. We specifically used the physics-based integrated hydrologic models because these

models do not strongly rely on the historical/initial conditions and rather are controlled by the representations of watershed processes and physical characteristics of the area. Likewise, because the climate model is also solving the fundamental physics of fluid flow, thermodynamics, etc. in the atmosphere, it doesn't rely on historical and past observations to bound the simulations. Moreover, the memory of physics-based climate models is shorter than that of the integrated hydrologic models. The uncertainties that could arise from the long forecast is the trajectory of $CO_2$ emissions that could potentially change by the end of the century. We also perform long-term, 30-year simulations because we are not trying to forecast the exact conditions at end-of-century rather looking to investigate the envelope of possibilities based on atmospheric dynamics and thermodynamics and their impacts on hydrologic processes.

3) It is shown that due to long-range dependence effect to key hydrological-cycle processes (e.g., Dimitriadis et al., 2021) such as the ones the authors use, the variability of each climatic process would be even higher than, for example, under the assumption of zero auto- and cross- correlation (i.e., white noise). Please show/discuss whether the proposed model assumes a correlation function for the input variables. I also suggest the authors see/discuss whether their model forecasts also capture (and verify) the stochastic characteristics of the historical timeseries including the effects from climate change (such as marginal distribution function, autocorrelation function, etc.).

As mentioned in the previous answer, we used a physics-based model not a machine learning model that is based on the previous observations to perform prediction and is strongly dependent on the previous conditions and the period used to do the training and make the predictions. Also, because these models are based on physics there is no need to account for a longer historical period that captures the statistical distribution of the event. Nonetheless, we validate our model by testing its ability to simulate dry and wet years in California. The comparisons (please refer to the previous answer) have shown that the developed model captures such extremes.

4) There are many equations in the text. Please consider creating a Table with all the inputs variables, output variables, boundary conditions, model assumptions, model limitations, simulation times, discretization method, etc., in order to help the readers identify the complexity/strength of the proposed model.

We have added the following paragraphs to Appendix B of the revised manuscript.

1. **Input Variables**

[Figure]

Figure R2a: Geological map of the Cosumnes watershed (source: USGS, Jennings et al., 1977)

| Hydrodynamic properties based on the geology | | | | |
|---|---|---|---|---|
| Geological Formation | Porosity (-) | Specific Storage (m-1) | Van Genuchten α (m-1) | Van Genuchten n (-) |
| Bedrock (Consolidated, Plutonic and Volcanic Rocks) | 0.02 | 10-6 | 3.0 | 3.0 |
| Alluvial aquifers | 0.2 | 10-4 | 3.0 | 3.0 |

Table R2b: Assigned values of hydrodynamic parameters (porosity, specific storage and Van Genuchten parameters). Values are based on literature review (Faunt et al., 2010; Faunt and Geological Survey (U.S.), 2009; Flint et al., 2013; Gilbert and Maxwell, 2017; Welch and Allen, 2014).

[Figure]

Figure R2b: Cosumnes watershed characteristics: land use and land cover (source: Homer et al., 2015), and model boundaries.

| Surface roughness based on land use | | | |
|---|---|---|---|
| **Land Use** | **Manning Coefficient (h.m-1/3)** | | |
| Forest | 5x10-2 | | |
| Shrub land and agricultural area | 5x10-3 | | |
| Urban areas | 5x10-5 | | |
| **Crop properties** | | | |
| **Crop Type and Reference** | **Height (m)** | **Maximum Leaf Area Index (-)** | **Minimum Leaf Area Index (-)** |
| Alfalfa (Evett et al., 2000; Orloff, 1995; Robison et al., 1969) | 0.6 | 6.0 | 2.0 |
| Pasture (Buermann et al., 2002; King et al., 1986; Rahman and Lamb, 2017) | 0.12 | 6.0 | 1.0 |
| Vineyards (Johnson and Pierce, 2004; Vanino et al., 2015) | 0.9 | 3.0 | 0.6 |

Table R2b: Manning coefficients and crop properties

| Boundary conditions | Value |
|---|---|
| Mokelumne and American river | Weekly-varying Dirchlet boundary conditions. These values are based on the measured river stages. |
| Sierra Nevada limit | No flow Neumann boundary condition |
| Bottom of the model | No flow Neumann boundary condition |

Table R2c: boundary conditions

**2. Numerical model set-up**

| Domain size | ~7000 km2 |
|---|---|
| Spatial discretization | 200 m horizontal from 0.1 m to 30 m in the vertical direction

 **Vertical Resolution**
 {table below} |
| Simulation time | Model validation (from water year 2012 to water year 2017), then future water years. |
| Temporal discretization | hourly |

**Vertical Resolution**

| Layer | 1 | 2 | 3 | 4 | 5 | 6 | 7 | 8 |
|---|---|---|---|---|---|---|---|---|
| $\Delta z$ (m) | 0.1 | 0.3 | 0.6 | 1.0 | 8.0 | 15.0 | 25.0 | 30.0 |

Table R2d: Numerical model discretization

**3. Output variables**

| Selected output variables | Temporal scale | Spatial scale |
|---|---|---|
| Snow Water Equivalent | Yearly, monthly, and hourly | Domain-average and point scale |
| Evapotranspiration | Yearly, monthly, and hourly | Domain-average and point scale |
| Soil Moisture | Yearly, monthly, and hourly | Domain-average and point scale |
| River Stages (also surface water storages) | Yearly, monthly, and hourly | Domain-average and point scale |
| Groundwater levels variations (also subsurface storages) | Yearly, monthly, and hourly | Domain-average and point scale |

Table R2e: Selected output variables

5) Please include more details on the water-energy balance equation and show whether is preserved in historical and forecasts. Also, have the authors included in the mass-energy balance analysis groundwater depletion in California (e.g., Badiuzzaman et al., 2017) and effects from sea level rise and ocean dynamics (e.g., Katsman et al., 2008)?

Mass balance is preserved when solving the mixed form of the Richards equation shown in equation (1) (Celia, et al., 1990). ParFlow-CLM numerically solves this equation by using the New-Krylow linearization scheme, this scheme iteratively solves the equation at each time step until the mass balance criteria set (equal to $10^{-3}$) is satisfied. Any large errors in the mass balance will automatically stop the resolution of the equation.
We have added the mass conservative properties of the mixed of the Richards equation in the revised manuscript, please refer to lines 265-266, see below the text in italic.

*"ParFlow solves the mixed form of the Richards equation which has the advantage of conserving the mass (Celia et al., 1990)."*

The Richards equation as shown in (1) accounts for groundwater depletion which is included in the term qs. While groundwater depletion plays an important role in the hydrodynamics of California we didn't account for this effect in this study (stated in lines 377-382 of the revised

manuscript) because the current pumping rates are difficult to estimate and their prediction by the end of the century is highly uncertain as it depends on many factors including policy and management. We, however, tested the impacts of this assumption of excluding pumping in our simulation (please see below the discussion: Discussion on the potential impacts of groundwater depletion on hydrologic projection in California). We found that the simulations without pumping do not significantly change the observed dynamics of the system, but they could overestimate the depletion of aquifer by evapotranspiration by 5 to 10%.

The watershed is not located near the coastal region; therefore, the effects of sea level rise are negligible.

**Discussion on the potential impacts of groundwater depletion on hydrologic projection in California**

*Because pumping rates may substantially change in the future due to new demands, policies/ regulations, and changes in land cover and land use, a model which includes a projection (or an envelope of these projections) is a work in itself. Therefore, we did not include them in this work, although the ParFlow-CLM model of this basin was developed to account for an approximation of the pumping and irrigation practices (to date) in the Central Valley. In the simulations originally shown here, we chose to simulate the natural system, given the constraints and uncertainty around the aforementioned projections in water and land management practices. However, we have taken the reviewer's comment seriously and have performed additional simulations comparing the EoC simulations with pumping and irrigation as a type of "numerical experiment". Specifically, we performed two additional simulations for both historical and EoC median water years with pumping and irrigation. The two simulations are as follow:*

- *Baseline without any pumping and irrigation*

- *Pumping and irrigation, around 700 pumping wells operating from April to November have been placed in the Central Valley aquifers. The number of wells, timing, and rates of pumping were determined by discussion with stakeholders in the areas and an estimation technique, which accounts for the water required by each crop for its optimal growth. More details about the estimation technique can be found in Maina et al., (2020a).*

*Figure R6 illustrates the temporal variations of surface water and groundwater storages obtained with the four simulations. As expected, the pumping scenarios have lower storages than the baselines. We notice that both pumping and baseline EoC scenarios are characterized by an earlier and higher increase in groundwater and surface water storage compared to the historical conditions (similar to the main conclusions of our study). These storages decrease by the end of the water year to become nearly equal to the historical baseflow conditions. In the baseline scenarios, the EoC groundwater storage is lower than the historical groundwater storage into August, though the historical baseline storage dips below the EoC baseline in September. In contrast, in the historical pumping scenario the groundwater storage remains lower throughout*

*the summer, showing distinct behavior in the pumping scenarios. We attribute this difference to reduced evapotranspiration in the pumping scenarios because of the deep water tables.*

[Figure]

*Figure R2c: Temporal variation of groundwater and surface water storages associated with EoC and historical baseline and pumping scenarios. The dashed green lines indicate the beginning and end of the pumping.*

*An analysis of the spatial differences between the baseline and the pumping differences (not shown here) has shown that these differences are mostly located in areas close to the pumping wells. Figure R7 depicts the temporal variation of water table depth and recharge associated with EoC and historical baseline and pumping scenarios at a selected point (located close to the pumping wells) in the Central Valley.*

*In the pumping scenarios, the water table decreases in the first two months whereas the water table is constant during this period in the baseline simulations. As the water table becomes deeper, the recharge also decreases. In the EoC, there is an early rise of the water table and an increase in recharge in both pumping and baseline scenarios due to the meteorological conditions (high and early precipitation). The water table rises earlier in the baseline compared to the pumping scenario. This rise is much earlier in the EoC than the historical conditions because the high precipitation of the EoC quickly compensates for the depressions created by pumping and increases the water table and therefore increases the recharge as explained in the schematic figure R8.*

[Figure]

*Figure R2d: Temporal variation of water table depth (WTD) and recharge associated with EoC and historical baseline and pumping at a selected point in the Central Valley. The dashed green lines indicate the beginning and end of the pumping.*

[Figure]

*Figure R2e: Schematic representation of the influence (on recharge) of pumping in historical and EoC conditions. At a local point, early and high precipitation of the EoC leads the water table to rise earlier and the recharge to increase because the unsaturated zone (UZ) becomes less thick, and the effective permeability k becomes higher.*

*While the pumping simulations have lower storages than the baselines, the mechanisms (early and high increases in storages and depletion in spring and summer) in both EoC and historical conditions remain the same. This is because we applied the same rate of pumping in both EoC and historical conditions and the timing of the pumping is assumed to be the same in both simulations. However, we note that the simulations without pumping could overestimate the depletion of aquifer by evapotranspiration by 5 to 10%.*

References
Badiuzzaman, P., E. McLauglin, and D. McCauley, Substituting freshwater: Can ocean desalination and water recycling capacities substitute for groundwater depletion in California?, J. Environ. Manag., 203, 123–135, 2017.
Dimitriadis, P., D. Koutsoyiannis, T. Iliopoulou, and P. Papanicolaou, A global-scale investigation of stochastic similarities in marginal distribution and dependence structure of key hydrological-cycle processes, Hydrology, 8 (2), 59, 2021.
Han, H., C. Choi, J. Jung, and H.S. Kim, Deep learning with long short term memory based sequence-to-sequence model for rainfall-runoff simulation, Water, 13 (4), p. 437, 10.3390/w13040437, 2021.
Katsman, C., W. Hazeleger; S. Drijfhout; G.J. van Oldenborgh, and G. Burgers, Climate scenarios of sea level rise for the northeast atlantic ocean: A study including the effects of ocean dynamics and gravity changes induced by ice melt, Climatic Change, 91, 351–374, 2008.

Yin, H., X. Zhang, F. Wang, Z. Yanning, R. Xia, and J. Jin, Rainfall-runoff modeling using LSTM-based multi-state-vector sequence-to-sequence model, J. Hydrol., 10.1016/j.jhydrol.2021.126378, 2021.

*Additional references*

*Maina, Fadji Z., Siirila-Woodburn, E. R., Newcomer, M., Xu, Z., & Steefel, C. (2020a). Determining the impact of a severe dry to wet transition on watershed hydrodynamics in California, USA with an integrated hydrologic model. Journal of Hydrology, 580, 124358. https://doi.org/10.1016/j.jhydrol.2019.124358*

*Maina, F. Z., Siirila-Woodburn, E. R., & Vahmani, P. (2020b). Sensitivity of meteorological-forcing resolution on hydrologic variables. Hydrology and Earth System Sciences, 24(7), 3451–3474. https://doi.org/10.5194/hess-24-3451-2020*

*Maina, Fadji Zaouna, & Siirila-Woodburn, E. R. (2020c). Watersheds dynamics following wildfires: Nonlinear feedbacks and implications on hydrologic responses. Hydrological Processes, 34(1), 33–50. https://doi.org/10.1002/hyp.13568*

*Harbaugh AW (2005) MODFLOW-2005, The U.S. Geological Survey modular ground-water model: the ground-water flow process. US Geol Surv Tech Methods 6-A16. http://pubs.usgs.gov/tm/2005/tm6A16/.*

*Trefry, M.G.; Muffels, C. (2007). "FEFLOW: a finite-element ground water flow and transport modeling tool". Ground Water. 45 (5): 525–528. doi:10.1111/j.1745-6584.2007.00358.x.*

*Neitsch, S. L., Arnold, J. G., Kiniry, J. R., & Williams, J. R. (2001). Soil and Water Assessment tool (SWAT) user's manual version 2000. Grassland Soil and Water Research Laboratory. Temple, TX: ARS.*

*Niu, G.-Y., et al. (2011), The community Noah land surface model with multiparameterization options (Noah-MP): 1. Model description and evaluation with local-scale measurements. J. Geophys. Res., 116, D12109, doi: 10.1029/2010JD015139.*

*Liang, X., D. P. Lettenmaier, E. F. Wood, and S. J. Burges (1994), A simple hydrologically based model of land surface water and energy fluxes for general circulation models, J. Geophys. Res., 99(D7), 14415–14428, doi:10.1029/94JD00483.*

*Aquanty, I. (2015), HydroGeoSphere User Manual, 435 pp., Waterloo, Ont.*

*Bixio, A. C., G. Gambolati, C. Paniconi, M. Putti, V. M. Shestopalov, V. N. Bublias, A. S. Bohuslavsky, N. B. Kasteltseva, and Y. F. Rudenko (2002), Modeling groundwater-surface water interactions including effects of morphogenetic depressions in the Chernobyl exclusion zone, Environ. Geol., 42(2-3) 162-177*

*Abbott, M. B., J. C. Bathurst, J. A. Cunge, P. E. Oconnell, and J. Rasmussen (1986), An introduction to the european hydrological system: Sys- teme hydrologique Europeen, She .2. Structure of a physically-based, distributed modeling system, J. Hydrol., 87(1–2), 61–77.*

*Coon, E. T., J. D. Moulton, and S. L. Painter (2016), Managing complexity in simulations of land surface and near-surface processes, Environ. Modell Software, 78, 134-149*

*Kollet, S. J., & Maxwell, R. M. (2006). Integrated surface–groundwater flow modeling: A free surface overland flow boundary condition in a parallel groundwater flow model. Advances in Water Resources, 29(7), 945–958. https://doi.org/10.1016/j.advwatres.2005.08.006*

*Maina F. Z., Delay F., Ackerer P. (2017). Estimating initial conditions for groundwater flow modeling using an adaptive inverse method. J. Hydrol. 552, 52–61. doi:10.1016/j.jhydrol.2017.06.041*

---

## Author Response (AR2)

The Authors have addressed all comments from the previous review round and adopted some of the suggestions. I kindly ask them to address the comments below, most of which are based on the replies to the previous review round:

*We thank the reviewer for their comments. We have added a discussion to the "study limitations" section, please refer to lines 875-901 of the revised manuscript. The reviewer's comments are in black, our responses are in blue italic fonts, and the added manuscript text are in blue plain text.*

1) Thank you for addressing this comment.

2) The authors have included in their reply of this comment the difference between forecast and projection. However, both weather-forecast and climate-projections should be accompanied by a probability of occurrence, since they both include expectations (or in other words, a statistical estimation of the expected/mean value of an envelope of events). In high-complex systems (such as climate dynamics), this can be achieved by a sensitivity analysis of the input parameters in order to create the envelope of different possible projections to see "what might happen if the world warms by +4-5 oC", as for example, for the RCP8.5 scenario that the authors use (e.g., see, for example, recent discussions in Schwalm et al., 2020; Hausfather and Peters, 2020). Therefore, a sensitivity analysis of the input parameters is required to create such an envelope of events. For example, the comparison between the observed and simulated river stages, groundwater levels and snow cover at the 3 stations indicated in Figures R1a-d, illustrate that (with the exception of snow cover) there is a large difference between them. However, if the Authors perform a sensitivity analysis on the several input parameters of their model, they would see that the cloud of simulations is expected to cover the observed records and to also assign a probability of occurrence to the end of century expectations.

Moreover, the high-dependence of the simulations on the initial conditions is a known issue in atmospheric dynamics and thermodynamics. One of the first studies (there are many since then) is by Lorenz (1963), who discovered the existence of the phenomenon of chaos in the atmospheric dynamics and the high-dependence on the initial conditions, which increases with time. Please note that only the statistical estimations, as for example the expectation/mean of an envelope of events, is expected to be independent to initial conditions, but this requires a sensitivity analysis and not the application of a deterministic model with fixed values of the input parameters and initial conditions.

I understand it is difficult for the authors to perform such a huge sensitivity analysis at this stage, and so, this is why I kindly ask to at least discuss the issue of uncertainty and variability on the input parameters and initial conditions, and thus, on the simulated output.

*We agree with the reviewer that due to the uncertainties associated with the numerous input parameters required by both the climate and hydrological models, a sensitivity analysis is ideal. However, these models are highly computationally expensive which makes the sensitivity analysis difficult to undertake. Indeed, it takes more than 24 clock hours to perform a single year simulation with the integrated hydrologic model using 320 cores on a supercomputer. Because the model requires a lot of parameters, a full exploration of the parameter space to perform a rigorous sensitivity analysis may require thousands runs of the model making such analysis*

*infeasible with the current available computational resources. We acknowledge this limitation in the revised manuscript by adding the following statement:*

"In this study, we relied on deterministic models to represent both the atmospheric (VR-CESM) and hydrologic (ParFlow-CLM) dynamics. These models are very sensitive to the initial conditions and input parameters (La Follette et al., 2021; Lehner et al., 2020; Song et al., 2015) which are uncertain given the lack of data characterizing the above and below-ground environment, including its hydrological response. Thus, while it is important to assess the sensitivity of the model outputs to these uncertain parameters, these models are computationally expensive and require many parameters. For example, a complete sensitivity analysis of the hydrologic model requires running it thousands of times to explore the full parameter space (which has a dimension of over 29). Such an approach is not feasible with the currently available computational resources because it takes longer than one wall-clock day to simulate a single water year for a single model parameterization, even in a high-performance computing environment. Future work could employ reduced order models based on a subset of the physics-based model runs to explore parameter space further (e.g., Maina et al., 2022)."

*In addition, the sensitivity of the model outputs to the initial conditions could be reduced by performing a spin-up which consists of running the model multiple times repetitively until an equilibrium is reached. We have used this approach to develop our model.*

3) The so-called Long-Range Dependence (LRD; or else known as long-term persistence or the Hurst phenomenon; Hurst, 1951) was the first study that gave a justification of why the so-called clustering of events (i.e., the tendency of wet/dry years to occur together in a non-predictive manner forming clusters) may occur in natural processes. Earlier, and independently, Kolmogorov (1940) introduced the so-called fractional-Gaussian-noise (fGn) model that was able to simulate this clustering behaviour. Since then, there are many developments towards the simulation of the LRD in natural processes and atmospheric dynamics (the literature is huge; see, for example, studies by Mandelbrot Wallis, 1968; Klemes, 1974; Tsonis, 1999; Dimitriadis et al., 2021, etc., which all include many references on this issue and identify the LRD in key hydrological-cycle processes - like streamflow, precipitation, air temperature, specific humidity, atmospheric pressure, wind speed, solar radiation, evapotranspiration, etc.- from analysis of thousands of stations and billions of records). The impact of the LRD behaviour on the atmospheric dynamics extends to the climatic scale (over 30-years), and so, it is not enough to show a validation of a 5 years with the model but rather to validate the model in an over-30-years window. I understand that it may be difficult to perform such analyses at this stage, however, I kindly ask the authors to at least discuss the impact and need for preservation of the LRD behaviour in the atmospheric dynamics model. This is in line with the previous comment, since the LRD is a stochastic attribute that arises from the intrinsic uncertainty of the chaotic behaviour apparent in atmospheric dynamics and thermodynamics models.

*We have now included in the "study limitations" section, these limitations of our approach based on deterministic models. Such approaches are embedded with uncertainties that must be assessed; however, given the computational demand of these models this is not currently feasible. Please see below the added paragraph.*

"In addition, because of the behavior of hydrological processes, the climate variability, and the

uncertainties of deterministic models, model validation should ideally be performed over a long period to account for different changes and variabilities. In this study, model validation was limited to a period of 5 years due to computational constraints. Although this period encompasses the wettest and driest years on record in the region, we acknowledge that it may not be sufficient to capture the full range of hydrological variability. Another limitation of using deterministic models is that the temporal variations of hydrological processes tend to follow a stochastic behavior in accordance with the so-called Hurst phenomenon (Hurst, 1951; Koutsoyiannis, 2003). As a result, the use of deterministic models such as the ones employed in this study could intensify the impacts of hydrological extremes and climate change."

4) Thank you for addressing this comment. From the provided Tables, it can be observed that there are in total 29 input parameters (8 for the hydrodynamic properties based on the geology, 12 for the surface roughness and crop properties based on land use, and 9 for the numerical model set-up) and 5 output variables.

*Yes. Because the model requires a lot of parameters, exploring the parameter space with 29 dimensions is not feasible with the current available computational resources. We now describe this in the study limitations section of the revised manuscript.*

5) Please consider further discussing the effect of groundwater and present both scientific opinions. It is mentioned by the Authors that the simulations without pumping do not significantly change the observed dynamics of the system; however, the overexploitation of groundwater is considered an important anthropogenic impact at the local and global climate and the hydrological cycle with a visible effect on sea level rise (see, for example, a recent analysis by Koutsoyiannis, 2020).

*We have added the following statement in the revised manuscript to discuss the effects of pumping:*
"Finally, it has also been demonstrated that while the changes in water balance exhibit greater variability on climatic scales, the most important changes in hydrologic processes remain the overexploitation of groundwater (Ferguson and Maxwell, 2010) which has an impact on the rise in sea level (Koutsoyiannis, 2020). In addition to projecting the use of groundwater by the end of the century, future studies could compare the two approaches (deterministic and stochastic) to better assess the limitations and the uncertainties associated with them.".

References
Dimitriadis, P., D. Koutsoyiannis, T. Iliopoulou, and P. Papanicolaou, A global-scale investigation of stochastic similarities in marginal distribution and dependence structure of key hydrological-cycle processes, Hydrology, 8 (2), 59, doi:10.3390/hydrology8020059, 2021.

Hausfather, Z., and G.P. Peters, RCP8.5 is a problematic scenario for near-term emissions, Proc. Natl. Acad. Sci. USA, 117, 27791–27792, 2020.

Hurst, Long-Term Storage Capacity of Reservoirs, Trans. Am. Soc. Civ. Eng., 116, 770–799, 1951.

Klemes, V., The Hurst phenomenon: A puzzle?, Water Resour. Res., 10 (4) 675-688, 1974. Kolmogorov, A.N., Wiener spirals and some other interesting curves in a Hilbert space, Dokl. Akad. Nauk SSSR, 26, 115–118, 1940.

Koutsoyiannis, D., Revisiting the global hydrological cycle: is it intensifying?, Hydrology and Earth System Sciences, 24, 3899–3932, doi:10.5194/hess-24-3899-2020, 2020. Lorenz, E.N., Deterministic nonperiodic flow. Journal of the Atmospheric Sciences, 20 (2), 130–141, 1963.

Mandelbrot, B.B. and J.R. Wallis, Noah, Joseph and operational hydrology, Water Resour. Res., 4, 909–918, 1968.

Schwalm, C.R., S. Glendon, and P.B. Duffy, RCP8.5 tracks cumulative $CO_2$ emissions, Proc. Natl. Acad. Sci. U.S.A., 117, 19656–19657, 2020. Tsonis A.A., P.J. Roebber, and J.B. Elsner, Long-range correlations in the extratropical atmospheric circulation: origins and implications, J. Clim, 12, 1534–41, 1999.

Dimitriadis, P., D. Koutsoyiannis, T. Iliopoulou, and P. Papanicolaou, A global-scale investigation of stochastic similarities in marginal distribution and dependence structure of key hydrological-cycle processes, Hydrology, 8 (2), 59, doi:10.3390/hydrology8020059, 2021.